



# The topography contribution to the influence of the atmospheric boundary layer at high altitude stations

Martine Collaud Coen[1], Elisabeth Andrews[2,3], Diego Aliaga[4], Marcos Andrade[4], Hristo Angelov[5], Nicolas Bukowiecki[6], Marina Ealo[7], Paulo Fialho[8], Harald Flentje[9], A. Gannet Hallar[10,11], Rakesh Hooda[12,13], Ivo Kalapov[5], Radovan Krejci[14], Neng-Huei Lin[15], Angela Marinoni[16], Jing Ming[17,18], Nhat Anh Nguyen[19], Marco Pandolfi[7], Véronique Pont[20], Ludwig Ries[21], Sergio Rodríguez[22], Gerhard Schauer[23], Karine Sellegri[24], Sangeeta Sharma[25], Junying Sun[26], Peter Tunved[14], Patricio Velasquez[27] and Dominique Ruffieux[1]

[1]Federal Office of Meteorology and Climatology, MeteoSwiss, 1530 Payerne, Switzerland

[2]University of Colorado, CIRES, Boulder, Colorado, 80305, USA

[3]National Oceanic and Atmospheric Administration, Earth System Research Laboratory, Boulder, Colorado, 80305, USA

[4]Laboratory for Atmospheric Physics, Institute for Physics Research, Universidad Mayor de San Andres, Campus Universitario Cota Cota calle 27, Edificio FCPN piso 3, La Paz, Bolivia

[5]Institute for Nuclear Research and Nuclear Energy, 1784 Sofia, Bulgaria

[6]Laboratory of Atmospheric Chemistry, Paul Scherrer Institute, 5232 Villigen PSI, Switzerland

[7] Institute of Environmental Assessment and Water Research, c/ Jordi-Girona 18-26, 08034 Barcelona, Spain

[8] Instituto de Investigação em Vulcanologia e Avaliação de Riscos - IVAR, Rua da Mãe de Deus, 9500-321 Ponta Delgada, Portugal

[9] Deutscher Wetterdienst, Met. Obs. Hohenpeissenberg, D-82383 Hohenpeissenberg, Germany

[10] Department of Atmospheric Science, University of Utah, Salt Lake City, UT, USA

[11] Storm Peak Laboratory, Desert Research Institute, Steamboat Springs, CO, USA

[12] Finnish Meteorological Institute, P.O. Box 503, 00101 Helsinki, Finland

[13] The Energy and Resources Institute, IHC, Lodhi Road, New Delhi -110003, India

[14] Department of Environmental Science and Analytical Chemistry (ACES), Atmospheric Science Unit, S 106 91 Stockholm, Sweden

[15] Department of Atmospheric Sciences, National Central University, Taoyuan, Taiwan

[16] Institute of Atmospheric Sciences and Climate, National Research Council of Italy, 40129, Bologna, Italy

[17] Max Planck Institute for Chemistry, Mainz 55128, Germany

[18] Guest at the State Key Laboratory of Cryospheric Sciences, Chinese Academy of Sciences, Lanzhou 730000, China





[19]Hydro-Meteorological and Environmental Station Network Center (HYMENET), National Hadry-Meteorological Service (NHMS), Hanoi, Vietnam

[20] Université Toulouse III - Laboratoire d'aérologie UMR 5560, 31400 Toulouse- France

[21] German Environment Agency, Platform Zugspitze, GAW-Global Observatory Zugspitze/Schneefernerhaus, Zugspitze 5, 82475 Zugspitze

[22] Izaña Atmospheric Research Centre, AEMET, Joint Research Unit to CSIC "Studies on Atmospheric Pollution",Santa Cruz de Tenerife, Spain.

[23] Sonnblick Observatory, Zentralanstalt für Meteorologie und Geodynamik (ZAMG), 5020 Salzburg, Austria

[24]Laboratoire de Météorologie Physique, UMR6016, Université Blaise Pascal, 63170 Aubière, France

[25] Climate Chemistry Measurements Research, Climate Research Division, Environment and Climate Change Canada, 4905 Dufferin Street, Toronto, M3H 5T4 Canada

[26]Key Laboratory of Atmospheric Chemistry of CMA, Chinese Academy of Meteorological Sciences, Beijing 100081, China

[27] Climate and Environmental Physics, Physics Institute, University of Bern, Bern, Switzerland

*Correspondence to*: Martine Collaud Coen (martine.collaudcoen@meteoswiss.ch)

**Abstract.** High altitude stations are often emphasized as free tropospheric measuring sites but they remain influenced by atmospheric boundary layer (ABL) air masses due to convective transport processes. The local and meso-scale topographical features around the station are involved in the convective boundary layer development and in the formation of thermally induced winds leading to ABL air lifting. The station altitude is not a sufficient parameter to characterize the ABL influence. Topography data from the global digital elevation model GTopo30 were used to calculate 5 parameters for 46 high altitude stations situated in five continents. The geometric mean of these 5 parameters determines a topography based index called ABL-TopoIndex which can be used to rank the high altitude stations as a function of the ABL influence. To construct the ABL-TopoIndex, we rely on the criteria that the ABL influence will be low if the station is one of the highest points in the mountainous massif, if there is a large altitude difference between the station and the valleys or plateaus, if the slopes around the station are steep, and finally if the drainage basin for air convection is small. All stations on volcanic islands exhibit a low ABL-TopoIndex whereas stations in the Himalaya and the Tibetan Plateau have high ABL-TopoIndex values. Spearman's rank correlation between aerosol optical properties and number concentration from 28 stations and the ABL-TopoIndex, the altitude and the latitude are used to validate this topographical approach. Statistically significant (s.s.) correlations are found between the 5 and 50 percentiles of all aerosol parameters and the ABL-TopoIndex whereas no s.s. correlation is found with the station altitude. The diurnal cycles of aerosol parameters seem to be best explained by the station latitude although a s.s. correlation is found between the amplitude of the diurnal cycles of the absorption coefficient and the ABL-TopoIndex. Finally, the main flow paths for air convection were calculated for various ABL heights.



# 1. Introduction

Climate monitoring programs aim to measure climatically relevant parameters at remote sites and to monitor rural, arctic, coastal and mountainous environments. The majority of these programs consist of in-situ instruments probing the Atmospheric Boundary Layer (ABL). The high altitude stations provide a unique opportunity to make long-term, continuous in-situ observations of the free troposphere (FT) with high time and space resolution. It is however well-known that, even if located at high altitudes, the stations designed to measure the FT may be influenced by the transport of boundary layer air masses. Remote sensing instruments can be used to complement in-situ measurements in order to provide more information about the FT. For example, sun photometers measure aerosol optical depth of the integrated atmospheric column including the FT although they don't provide vertical information to enable separation of FT and ABL conditions. Light Detection and Ranging (LIDAR) type instruments measure the profile of various atmospheric parameters (meteorological, aerosol, gas-phase) and thus can provide information not only on the ABL but also on the FT. They can be used to detect the ABL and Residual Layer (RL) heights at high altitude stations from a convenient site at lower elevation (Haeffelin et al., 2012; Ketterer et al., 2014; Poltera et al., 2017). These instruments are however limited in the presence of fog and low clouds and they don't measure above the cloud cover. Further, the use of LIDAR to attribute the various aerosol gradients to ABL layers remains a delicate problem. Lastly, few LIDAR instruments are currently installed in regions of complex topography. Instrumented airplanes can make detailed measurements of the vertical and spatial distribution of atmospheric constituents, but, because of the limited temporal scope of most measurement campaigns, cannot provide long-term, continuous context for the measurements. Ideally, to make FT measurements, a combination of these techniques would be used, but due to limited resources that is rarely possible. Thus, it is important to evaluate the constraints of each technique. The high altitude time series from surface measurements remain the most numerous and the longest data sets to characterize the FT and its evolution during the last decades. Here we focus on identifying factors controlling the influence of ABL air on high altitude surface stations hoping to sample FT air.

The ABL is the lowest part of the atmosphere that directly interacts with the Earth's surface and is most of the time structured into several sub layers. In the case of fair-weather days, the ABL has a well-defined structure and diurnal cycle leading to the development of a Convective Boundary Layer (CBL), also called a mixing or mixed layer, during the day and a Stable Boundary Layer (SBL) which is capped by a Residual Layer (RL) during the night (Stull, 1988). The aerosol concentration is maximum in the CBL during daytime and remains high in the RL. In the case of cloudy or rainy conditions as well as in the case of advective weather situations, free convection is no longer driven primarily by solar heating, but by ground thermal inertia, cold air advection, forced mechanical convection and/or cloud top radiative cooling. In those cloudy cases the CBL development remains weaker than in the case of clear sky conditions. Long range or RL advection can



however lead to a high aerosol concentration above the CBL during daytime, leading to high altitude aerosol layers (AL) that can be decoupled from the CBL and the SBL.

There are several rather complex mechanisms able to bring ABL air up to high altitude (Rotach et al., 2015; Stull, 1988; De Wekker and Kossmann, 2015). An important factor in many of these mechanisms is how the CBL develops over

mountainous massifs. In their extensive review of concepts, De Wekker and Kossmann (2015) studied the CBL development over slope, valley, basin, plateau as well as over complex mountainous massifs and concluded that the CBL height behavior can be categorized into four distinct patterns describing their spatial extent as a function of the surface topography: the hyper-terrain following, the terrain following, the level and the contra-terrain following. The type of CBL height behavior depends on several factors such as the atmospheric stability, synoptic wind speed and vertical and horizontal scale of the

orography. Stull (1992) concluded that the CBL height tends to become more horizontal (level behavior) at the end of the day, that deeper CBLs are less terrain following than shallower ones, and that the CBL top is less level over orographic features with a large horizontal extent. Even if the CBL height remains lower than the mountainous ridges, thermally driven winds develop along slopes, or in valleys or basins and these winds are able to bring ABL air masses up to mountainous ridges and summits. These phenomena were extensively modeled (Gantner et al., 2003; Zardi and Whiteman, 2012) and also

measured (Gantner et al., 2003; Rotach and Zardi, 2007; Rucker et al., 2008; Venzac et al., 2008; Whiteman et al., 2009) and are part of the active mountainous effects allowing a vertical transport of polluted air masses to the FT. Finally, ABL air masses can also be advected from mesoscale or wider regions and influence high altitude measurements. For example, a Continuous Aerosol Layer (CAL) is often measured above the CBL during dry, clear-sky and convective synoptic situations (Poltera et al., 2017).

The ABL influence of the mesoscale regions at high altitude sites were directly shown by airborne LIDAR measurements over the Alps and the Apennine (Nyeki et al., 2000, 2002; De Wekker et al., 2003) and more indirectly by the seasonal and diurnal cycles of aerosol parameters at high altitude stations (Andrews et al., 2011). Many methods have been used to separate FT from ABL influenced measurements, including those based on time of day and time of year approach (Baltensperger et al., 1997; Gallagher et al., 2011), wind sectors (Bodhaine et al., 1980), the vertical component of the wind

(García et al., 2014), wind variability (Rose et al., 2016), $CO_2$ (Zellweger et al., 2002), NOy/CO or radon concentrations (Griffiths et al., 2014; Herrmann et al., 2015) and water vapor concentrations (Ambrose et al., 2011; Obrist et al., 2008), although none of these methods leads to an absolute screening procedure to ensure the measurement of pure FT atmosphere.

The altitude range of stations which claim they sample in the FT (at least some of the time) spans from about 1000 to more than 5320 m a.s.l., but a simple analysis of the aerosol parameters (for example, the black carbon concentration) as a

function of altitude suggests that higher altitude stations are not necessarily less influenced by anthropogenic pollution. While station altitude may not be the main parameter explaining the ABL influence, topographical features around the station are nevertheless involved in the CBL development, and in the formation of thermally induced winds leading to ABL air lifting (Andrews et al., 2011; Kleissl et al., 2007). In addition to topography there are other important parameters determining the ABL influence at mountainous stations such as the wind velocity and direction, soil moisture and albedo,



synoptic weather conditions, pollution sources and sea surface temperature for islands, but none of these parameters will be considered in this study.

The aim of this paper is twofold: (i) to define a topography based index called ABL-TopoIndex that can be utilized to rank the high altitude stations as a function of the ABL influence and (ii) to compare the potential ABL influence of several

locations in a mountainous range in order, for example, to choose the best sampling location. Several tools used in this study are taken from the hydrology analysis field, since both air and water flow along defined flow paths. The ABL air masses flow towards high altitudes, in contrast to the downward flow of water. However, similar to hydrological concepts, the ABL "lakes" are found in the plains and valleys.

## 2.    Experimental

### 2.1.  Stations

Forty-six high altitude stations (Table 1 and Fig. 1) were selected based on various criteria, such as the presence of aerosol or gaseous measurements, their representativeness of the mountainous massif and/or the possibility to compare several stations from the same mountainous massif. They are representative of 5 continents and their altitudes range between 1074 (SHN) and 5352 m a.s.l (CHC). Even if clearly situated within the ABL, some stations like HPB, MSY or ZEP were added to this

analysis to verify the results of the ABL-TopoIndex at lower altitude sites. Several mountainous massifs such as the Alps, the Himalaya, the Rocky Mountains and the Andes Cordillera are well represented with three to five stations. Some other stations such as BEO in the Balkan Peninsula, HAC in the Peloponnese, WLG in China, MKN in Kenya and the high plateau of ASK in the Hoggar Mountains of southern Algeria are the only representative of their massif. The volcanic islands form a category in themselves, despite being located in different oceans and at various latitudes.

### 2.2.  Topography data and analysis

The topography data were taken from the global digital elevation model GTopo30 (https://lta.cr.usgs.gov/GTOPO30). GTopo30 has a horizontal grid spacing of 30 arc seconds corresponding to a spatial resolution between 928 m in the East/West direction at the equator, 598 m at WHI (50° N) and 373 m at the SUM polar station (72.6° N). In the North/South direction, 30 arc seconds are almost constant with latitude and correspond to 921 m at the equator and 931 m at the poles.

The geographical coordinate system WGS84 (World Geodetic System revised in 1984) from GTopo30 was projected in the Universal Transverse Mercator (UTM) conformal projection to ensure homogeneity in vertical and horizontal coordinates. The various algorithms were then tested to several domain sizes ranging from 50 to 1000 km². A domain size of 750 km x 750 km was finally chosen to calculate the ABL-TopoIndex. If not specified, the given altitudes correspond to altitude above sea level (a.s.l.).

The TopoToolbox-master version of the free shareware TopoToolbox (https://topotoolbox.wordpress.com/), a MATLAB-based software for topographical analysis (Schwanghart and Scherler, 2014), was used for this analysis. The water flow



paths were calculated with the single flow direction representation and the Digital Elevation models (DEM) were preprocessed by filling holes with a carving process prior to calculate the flow directions.

## 2.3. ABL-TopoIndex

To construct the ABL TopoIndex, we rely on the following four criteria to indicate that the ABL influence will be low if:

5       1) the station is one of the highest points in the mountainous massif,

      2) there is a large altitude difference between the station and the valleys, plateaus or the average domain elevation,

      3) the slopes around the station are steep, and

      4) the «drainage basin» for air convection is small.

Based on these criteria, the red station on Fig. 2 will be less influenced by the ABL than the blue station, despite being situated at lower altitude. These criteria are quantitatively represented using five parameters:

1. **Parameter 1 – hypso%:** A hypsometric curve is the cumulative distribution function of elevation on the considered domain – 750 km x 750 km for the ABL-TopoIndex. The frequency percentage of the hypsometric curve at the station altitude (hypso%) provides a representation of criterion 1 for a large spatial scale. Figure 3

presents some normalized hypsometric curves with dots indicating the station hypsometric value. While most of the high altitude stations have hypso% values less than 5%, PYR and NCOS are situated respectively at 21% and 58% on wide inflection points of the hypsometric curve. BEO and FWS are found at less than 0.001% of the curve indicating they are located at one of the highest points of their respective mountainous massifs. The ABL influence should increase with increasing value of hypso%.

2. **Parameter 2 – hypsoD50:** The second parameter (hypsoD50) is the difference between the station altitude and the altitude at 50% of the hypsometric curve for the 750 km x 750 km domain. The parameter hypsoD50 corresponds to criterion 2 for a large spatial scale. In some rare cases (e.g., MUK), the station is situated under the 50% of the hypsocurve leading to a negative hypsoD50. For these sites the hypsoD50 is set to the very small value of 10 to allow the geometric mean to be applied (see equation 1). The ABL influence decreases

with increasing values of hypsoD50.

3. **Parameter 3 – LocSlope:** The altitude difference between the station and the minima in a circular domain centered at the station is plotted as a function of the domain radius on Fig. 3b. The slope of this curve between 1km and 10 km is then calculated (LocSlope) and corresponds to criteria 2 and 3 for a small spatial scale. The steepness of the slopes (criterion 3) around the station is only evaluated from the station toward the lowest

elevations. Figure 3b shows that the change in the altitude difference as a function of domain radius can be very different from station to station. For example, there is a rapid decrease of the elevation minima with increasing distance that gradually levels off for radius greater than 7 km for JFJ and for radius greater than 4 km for MBO; there is a continuous decrease of the minima elevation for PYR and ASK up to radius larger than 30 km; and





there are some steps for CHC and BEO. NCOS appears very different than the other sites plotted since the NCOS station is near the vast Nam lake of 1950 km$^2$ situated at 4718 m. The ABL influence should increase with decreasing LocSlope.

4. **Parameter 4 – G8:** The mean gradient in elevation in the eight directions (N, NE, E, SE, S, SW, W, NW) at the station is called G8. This parameter takes into account the slopes towards lower and higher elevations over a local scale (2-4 km, with the size of the local scale depending on latitude) and corresponds to criterion 3. The ABL influence should decrease for increasing G8 gradient.

5. **Parameter 5 – DBinv:** Since the air masses have to "flow" from the plain towards the summit to influence the station measurements, the size of the drainage basin (DBinv) for convection can be calculated with hydrology tools using an inverse topography, where the altitude Z is changed to –Z allowing the summit to become a hole. To do this calculation, all grid points with an altitude higher than the station altitude are set equal to the station elevation to minimize artifacts. Figures 4d and 5d are examples of the DBinv calculation for BEO and PYR. The DBinv is related to criterion 4. The ABL influence should increase with increasing size of the convection drainage basin.

To summarize, the ABL influence should increase with decreasing values of hypsoD50, LocSlope and G8 and with increasing values of hypso% and DBinv. Thus, to determine the ABL-TopoIndex, the geometric mean is calculated on the inverse of hypsoD50, LocSlope and G8 along with the values of hypso% and DBinv. To avoid any particularities of the station site and due to the fact that the ABL influence is a regional factor, the mean of the values at the grid point containing the station and at the eight neighboring grid points (recall that grid spacing is 30 arc seconds) are used to calculate the ABL-TopoIndex. The ABL-TopoIndex is then taken as the geometric mean of the five parameters:

$$ABL\text{-}TopoIndex = \sqrt[5]{hypso\% \times \frac{1}{hypsoD50} \times \frac{1}{LocSlope} \times \frac{1}{G8} \times DBinv} \qquad (1)$$

The geometric mean is used here on strictly positive parameters that have widely different numeric ranges (e.g., Table 2). The geometric mean is used instead of the arithmetic mean because it effectively "normalizes" the various parameter ranges, so that no parameter dominates the weighting. Further, a given percentage change in any of the parameters will yield an identical change in the calculated geometric mean value. Because of these properties, the geometric mean is the recommended method to determine a meaningful indices from multiple parameters (Ebert and Welsch, 2004). The extrema, median and mean of the parameters constituting the ABL-TopoIndex are reported in Table 2. The value of the ABL-TopoIndex has no significance in itself, so that the units are not important, but it allows ranking of the stations as a function of the ABL influence due to convection.





## 2.4 Aerosol parameters

Aerosol datasets from 25 high altitude and 3 mid-altitude stations (Table 1) were available for this study. The datasets comprise absorption coefficient, scattering coefficient and/or number concentration and cover time periods ranging from at least one year up to more than one decade of measurement (see supplement Table S1). Stations with time series shorter than

one year were not used, since they are not representative of a complete seasonal cycle. Due to the non-normal distribution of the aerosol parameters, the 5, 50 and 95 percentiles were taken as representative of the minimal, central position and maximal concentrations.

No correction for standard pressure and temperature were applied in order to use the measured aerosol properties and concentration at high altitude. For consistency, the measured hourly absorption and scattering coefficients were adjusted to a

wavelength of 550 nm if reported at a different wavelength. using an Ångström exponent of 1. Additionally, the scattering coefficients were corrected for truncation error. Because of their measurement technique and the low aerosol concentrations at many high altitudes, filter-based photometers regularly measure negative absorption coefficients at some of these sites. Some datasets contain up to 20-30% of negative absorption values. Depending on the data owner's policy, these negative values were either left in the dataset, set to zero (or to a minimal value) or considered as missing values. To ensure a similar

treatment for all datasets, negatives values, zeros or minimal values attributed to negatives were therefore set to missing values.

The diurnal and seasonal cycles were only analyzed on datasets longer than 2 years. To be able to statistically calculate the diurnal and seasonal cycles, the autocorrelation at one hour (first lag) were first removed from the dataset by a whitening procedure (Wang and Swail, 2001). The auto-correlations at each lag time were then calculated on the whitened dataset

taking into account missing data (see supplement for further explanations). Only auto-correlation values statistically significant at 95% confidence level were kept. Since the diurnal (24 h) and annual cycles (365 days) were not well defined due to variable meteorological conditions and some shorter datasets, the auto-correlation at lags 22 to 26 h and at lags 350 to 380 days were summed to obtain the strength (i.e., the cycle amplitude) of the diurnal and seasonal cycles, respectively. Noise in the aerosol measurements makes the strength of the cycle a somewhat qualitative value. The diurnal cycles were

calculated for each month of the year in order to observe the seasonal change of the diurnal cycles.

## 3. Results

### 3.1. Case studies

Mount Moussala (BEO) is the highest summit not only of Bulgaria but of the whole Balkan massif. The regional GAW station is located at the summit (2925 m). The topographic dominance of BEO can be visualized on the topography map

(Fig. 4a). Figure 4a also shows the main flow paths which follow the Iskar, Martisa and Metsa rivers. Figure 4b shows the flow accumulations, which are the accumulated flows of all cells flowing into each downslope cell in the output raster,



allowing visualization of the greatest features of the Bulgaria hydrographic network. BEO is at the junction of four drainage basins corresponding to the four main rivers (Fig. 4c). Figure 4d shows that when the convection drainage basin is calculated with the inverse topography, BEO is in the center of a large drainage basin that covers most of the plotted domain. Even though BEO's altitude is under 3000 m, BEO's ABL-TopoIndex of 0.22 is one of the lowest due to an almost zero hypso%

(2e-4 %), a high hypsoD50 of 2603 m and a small DBinv of $1.4*10^5$ (Table 2). HAC is a very similar case to BEO since it is situated almost at the top of Mount Helmos, the third highest mountain of the Peloponnese (Greece).

PYR (5079 m) is the second highest station considered here, but the station is located at the foot of Mount Everest (8848 m) at a confluence point of several valleys (Fig. 5a and b).  Figure 5c shows that PYR is situated in the middle of a very large hydrological drainage basin. The PYR ABL-TopoIndex is consequently quite high (3.89) and supports the observation of a

large ABL influence in the Himalaya region  (Bonasoni et al., 2008). The daily arrival of polluted air masses from the Indo-Gangetic plain is frequently reported in PYR data analyses (Bonasoni et al., 2010, 2012; Marinoni et al., 2010).

## 3.2.  Relation between ABL-TopoIndex and domain size

  The ABL-TopoIndex depends on the size of the chosen domain (Fig. 6a). The gradient G8 and the local slope LocSlope are

calculated on small fixed horizontal scales (3 and 10 km, respectively) and are consequently constant with domain size (Fig 6e,f), although there are small fluctuations in LocSlope and the G8 parameters due to some changes occurring during the projection of GTOPO30 in the UTM WGS84 coordinate system. The other three parameters do change with domain size which is the reason that the ABL-TopoIndex also is a function of the domain area. DBinv tends to increase with the domain size for all stations (Fig. 6b), since the low altitude area potentially contributing to the ABL influence increases with domain

size. The hypso% decreases continuously for stations situated in a dominant position in the whole mountainous massif such as JFJ, SBO or BEO (Fig. 6c). For stations located at a lower position in the massif (see for example HPB), the hypso% first increases before decreasing once the domain contains all the highest peaks of the massif. Finally, stations situated atop a high local mountain but surrounded by higher mountains such as MUK (not shown in Fig. 6) have a continuously increasing hypso% up to a domain size of $10^6$ km². HypsoD50, the difference between the station elevation and the minimum of

elevation in the domain, always increases (or at least stays constant) with domain size but changes more or less rapidly depending on the domain topography (Fig. 6d). In general, the ABL-TopoIndex usually increases with domain size (i.e., more ABL influence). The greatest increases are usually found for the stations with the highest ABL-TopoIndex at small domain sizes and are due to an increase in DBinv overcoming the decrease in hypso% and the increase in hypsoD50. Some stations (primarily sites located in the Himalayas and not pictured on Fig. 6) exhibit a decrease in ABL-TopoIndex for small

domain sizes (PYR) due to large variations of the hypso% and hypsoD50, or a decrease for large domain sizes (LAN, HLE or YEL) due to a large decrease of hypsoD50 when low altitude regions are taken into consideration.





### 3.3. Relation between ABL-TopoIndex and altitude

As stated in the introduction, the development of the ABL-TopoIndex relies on the assumption that the station position in the mountain massif is a better criterion for determining the ABL influence than the station altitude. To compare these two parameters, the ABL-TopoIndex is reported as a function of the altitude for all grid points in a 5km x 5km domain around some stations on Fig. 7. For the grid points at the highest altitudes, there is a clear dependence between the ABL-TopoIndex and the altitude, with ABL-TopoIndex increasing (more ABL influence) as altitude decreases; Fig. 7 shows that the OMP and PYR regions have the steepest and ASK the flattest ABL-TopoIndex decrease with altitude. At middle altitudes for each massif, the valleys, high plateaus, various mountainous slopes and networks lead to a wide range of the altitudes corresponding to the same ABL-TopoIndex value. For example, the altitude range corresponding to an ABL-TopoIndex of 2 varies between 3000 m and 6000 m at PYR, while at OMP an ABL-TopoIndex of 2 is achieved at an altitude range between 250-350 m. At SBO, for ABL-TopoIndex values smaller than 1, there are three discrete groupings of points likely corresponding to the basins of three different valleys around the site. The ABL-TopoIndex values of the stations shown in Fig. 7 are indicated by the squares which allows visualization of their relative situation in their respective mountain massifs: OMP, HAC and CHC and to some extent SBO were constructed at places with the lowest ABL-TopoIndex of their regions thus minimizing potential ABL influence. In contrast the region around PYR (and to a lesser extent ASK) shows locations with much lower ABL-TopoIndex (less ABL influence) at a similar altitudes to the stations.

### 3.4. Ranking of the stations by the ABL-TopoIndex

The ABL-TopoIndex values for the forty-six stations are grouped on Fig. 8 by continents and mountainous massifs or regions (see Table 1) that can correspond to various geomorphologies. The first obvious observation is that all islands have very low ABL-TopoIndex (note the logarithmic scale for the ABL-TopoIndex), whereas the stations in the Himalaya massif have the greatest ABL-TopoIndex. Further conclusions that can be derived include:

- **Islands**: the islands with sites included in this study have a small area, are delimited by the large flat ocean (though most of them are grouped in archipelagos) and their summits were formed by volcanic activities leading to steep slopes. All these factors lead to very low ABL-TopoIndex values. Pico Mountain Observatory (OMP) in the Portuguese Azores archipelago ranks as the monitoring station with the lowest ABL influence. The low ABL-TopoIndex of OMP is caused by the following reasons: 1) Pico Mountain is the only summit of the island and is not only the highest mountain (2350 m) of the Azores but also of the mid-Atlantic ridge, 2) OMP is the smallest island (surface area = 447 $km^2$) and 3) the research station is only 126 m below the mountain summit. The effect of the proximity to the summit can be clearly seen by the difference between TDE (ABL-TopoIndex of 0.1) and IZO (ABL-TopoIndex of 0.3). These two sites, both located on the island of Tenerife, are separated by only 15 km in horizontal distance but a vertical distance of 1165 m, with TDE being the higher station. Taiwan, where LLN is



located, has the greatest surface area (36193 km$^2$) of the islands considered and, additionally, is in close proximity to a continent (China is at 130 km to the West). Both of these facts explain LLN's high ABL-TopoIndex in the island category. MLO in Hawaii is the highest station located on an island (3397 m), but the island of Hawaii has a second summit, Mauna Kea (4205 m). Further, the MLO research station is 870 m beneath the volcano top, explaining why its ABL-TopoIndex is higher than that of OMP. This difference in ABL-TopoIndex between OMP and MLO is confirmed by an almost daily occurrence of buoyant upslope flow at MLO while such flow patterns are much less frequent (<20% of the time) at OMP (Kleissl et al., 2007).

- **Alps**: The European Alps consist of a broad mountainous massif with the highest summits between 4500 and 4800 m. The three high research stations (JFJ, SBO and ZUG) are located between 2900 m and 3600 m, ZSF being only some 300 m under ZUG. HPB (985 m) was added to this study as a low elevation station in the Alps. All the three high elevation stations have low ABL-TopoIndex: JFJ (0.49), SBO (0.69) and ZUG (0.96). Their ABL-TopoIndex values are generally a little higher than those determined for the islands. As expected, the ABL influence at HPB is much stronger (ABL-TopoIndex is 5.38) due to both its lower altitude and position near the bottom of the Zugspitze massif.

- **Pyrenees**: the Pyrenees are a natural border between France and Spain and peak at 3400 m. PDM is a high altitude station (2877 m) with an ABL-TopoIndex similar to the European alpine high altitude stations. MSA is located at a mid-altitude range of the massif and has a median ABL-TopoIndex, while the low altitude MSY station, added for comparison purposes, has a high ABL-TopoIndex.

- **Other European stations**: BEO and HAC are situated at the highest points of their massifs and have therefore very low ABL-TopoIndex values, comparable to those of the island high elevation sites. The lower altitudes of CMN and PUY, their middle position in mountainous massifs containing several higher summits and, to a lesser extent, their proximity other massifs such as the Alps and the Pyrenees result in higher ABL-TopoIndex values for these two sites.

- **Himalaya and Tibetan Plateau**: The Himalaya is the highest mountainous massif on Earth with 14 summits peaking at more than 8000 m. The altitude of the research stations between 2200 and 5100 m are therefore at relatively low elevation in comparison to the summits. This is clearly reflected in their high ABL-TopoIndex values (between 4 and 30). MUK and SZZ are both situated in the foothills of the Himalaya in India (Uttarakhand region) and in south China (Yunnan region), respectively, and both have an ABL-TopoIndex value in the 4-5 range. Although MUK is at a lower altitude than SZZ, it is located at a higher position than SZZ relative to the mean altitude of its meso-scale environment. The high ABL-TopoIndex values for HLE and NCOS are due to their position in a large valley and on the edge of a vast lake, respectively, that largely decreases all the parameters related to criteria 1, 2 and 3 (see Sect. 2.3). WLG is constructed within some ten's of meter of Mount Waliguan's summit at the northeastern part of the Tibetan plateau, so that its dominant position in its meso-scale domain leads to a middle range ABL-TopoIndex value.



- **Japan** : Mount Fuji is the highest peak of Japan and the research station is located at the top of the symmetric volcano located near the coast. The second highest peak in Japan is some 500 m lower than Mount Fuji. This particular topography leads to an ABL-TopoIndex similar to the volcanic islands. The two other Japanese stations are at much lower altitudes and have mid-range ABL-TopoIndex values.

- **North America**: Four stations (MZW, NWR, SPL and YEL) are situated in the Rockies, whose summits peak at 4400 m. The three stations higher than 3000 m have lower ABL-TopoIndex values similar to some of the European mountains, whereas YEL is situated on the large Yellowstone plateau at an average elevation of 2400 m resulting in a high ABL-TopoIndex (8.1) that is similar to the values for NCOS and HLE. Mount Bachelor (MBO) is located near the top of an isolated volcano from the Cascade volcanic arc that dominates the plain surrounding it, explaining

its low ABL-TopoIndex. WHI is located in the Pacific Coast Mountains, the mountain range name referring to the vicinity between the high altitude massif and the ocean coast. The highest peaks in the Pacific Coast Mountains have summits between 3000 and 4000 m (WHI is at 2182 m). WHI has middle range ABL-TopoIndex (1.25) despite its low altitude due to the proximity of the ocean and to the rather narrow width of the massif (300 km). Both APP and SHN are situated at the same altitude in the Blue Ridge mountains of the Appalachian range. At the

latitude of SHN, the Blue Ridge mountain is much narrower than at APP's latitude. Moreover SHN is almost on the top of the ridge whereas APP is on a plateau. SHN therefore has a higher G8 and LocSlope and a lower hyps% and DBinv leading to much lower ABL-TopoIndex than found for APP.

- **Andes**: CHC (5320 m), the highest station in this study, is located in the Cordillera Oriental, itself a sub-range of the Bolivian Andes massif, and is part of the mountain bell surrounding the Altiplano (literal translation high plain)

with an average height of 3750 m. This position explains its mid-range ABL-TopoIndex of about 1 due to relatively high hypso% (0.14%) and low hypsoD50 (2688 m). PEV (4765 m), the South America station with the lowest ABL-TopoIndex, is located at the extreme northeastern extension of the South America's Andes mountain range that peaks at about 5000 m. Its high position in its mountain range is characterized by a very low hypso% (0.013 %) and the highest hypsoD50 of 4633 m. TLL is situated in the foothills of the Andes in Chile near to the Pacific ocean

and has a similar ABL-TopoIndex to MUK due similarities in topography. LQO is at higher altitude than TLL but located in the middle of the Altiplano leading to an ABL-TopoIndex larger than 10.

- **Africa**: Mount Kenya (5199 m), the second highest peak of Africa and the highest in Kenya, is an isolated volcanic massif with several peaks. MKN observatory is located some 1500 m under Mount Kenya's summit resulting in a mid-range ABL-TopoIndex of about 1. Assekrem (ASK, 2710 m) is located on a small (about 2.5 km$^2$) high plateau

in the Hoggar Mountains located in central Sahara. The highest summit in the Hoggar range peaks at 2908 m. Despite being situated on a flat area, ASK has quite low ABL-TopoIndex value because of its relatively high elevation in the Hoggar Mountains.

- **Arctic**: SUM is located high atop the Greenland ice sheet in the central Arctic. The ice sheet has a smooth topography due to its build up by glaciation and precipitation. While SUM has a high hypso%, its hypsoD50 and



LocSLope are very low leading to mid-range ABL-TopoIndex. The Zeppelin Observatory in Svalbard is located near the top of Zeppelinfjellet above Ny-Ålesund. While its altitude of 475 m is very low, its ABL-TopoIndex is similar to those of HPB, MSY, LQO and YEL though they are situated at much higher altitudes than ZEP.

## 3.5. Correlation between aerosol parameters and the ABL-TopoIndex

While Fig. 8 shows that there are some clear patterns in the ABL-TopoIndex, it is also instructive to see how the ABL-TopoIndex relates to measurements at mountain sites and to compare those relationships with other indicators of ABL influence. In order to have a robust estimate of the correlation between the aerosol measurements and the topographical parameters the Spearman's rank correlation was calculated. The Spearman's rank correlation measures the strength and direction between two ranked variables without the requirement that the variables be normally distributed. Here it is also used to verify that the assumed relationships between topographical and aerosol parameters correspond to those proposed in section 2.3 (e.g., that a positive correlation with aerosol loading as a surrogate for ABL influence is found for the ABL-TopoIndex, hypso%, DBinv and station altitude and an anti-correlation for hypsoD50, LocSlope and G8). Happily, that is the case for all topographical parameters. The Spearman's rank correlation coefficients of the 5th, 50th and 95th percentiles of the measured aerosol parameters with the altitude, the latitude, the ABL-TopoIndex as well as all the individual parameters constituting the ABL-TopoIndex are presented on Fig. 9.

The ABL-TopoIndex has statistically significant (s.s.) correlation for the 5 and 50 percentiles of all aerosol parameters and for the 95 percentile of the absorption coefficient. The highest correlation and s.s. are found for the absorption and the number concentration 5 and 50 percentiles. The correlation with the maxima of the aerosol parameters (95th percentile) is always lower and s.s. only for the absorption coefficient. The minima of the aerosol parameters - particularly of the absorption coefficient- correspond to the measurement of air masses with the lowest aerosol concentration, namely FT air masses with the lowest ABL influence and no advection of polluted air masses. In contrast, the maxima correspond to the advection or convection of air masses with high aerosol loads and can, to some extent, be caused by special events such as dust or biomass burning events. In contrast to the absorption coefficient, the particle number concentration (and, to a far lower extent, the scattering coefficient) depend not only on the ABL influence but also on the new particle formation (NPF) that can be enhanced at high altitudes (Boulon et al., 2011; Rose et al., 2015). Thus, the high correlations of the ABL-TopoIndex with the minima of the aerosol absorption coefficient as well as its lower correlation with the absorption coefficient maxima and with the scattering coefficient suggest the ABL-TopoIndex is indeed a promising indicator for ABL influence.

Amongst all the parameters constituting the ABL-TopoIndex, the hypso% has clearly the greatest correlation with the aerosol optical properties with s.s. coefficient of determination between 0.46 and 0.73 for all aerosol parameters. The LocSlope and G8 anti-correlations are also s.s. for all number concentration percentiles and for the 5 and 50 percentiles of





the absorption coefficient. The hypsoD50 is s.s. for the 5% of the absorption coefficient and for the 50% and 95% of the number concentration. Only the DBinv exhibits no s.s. correlation with any of the aerosol parameters.

There are no s.s. correlations among the station altitude and the percentiles of any of the aerosol parameters. The station elevation is therefore not a good predictor of the ABL influence (at least as it relates to particle concentration and aerosol optical properties). The latitude has s.s. anti-correlation with 50% and 95% of the scattering and absorption coefficients. These anti-correlations may be explained by the more intense insolation at low latitudes leading to higher surface temperature and greater convection.

The correlations of the topographical parameters with the diurnal and seasonal cycles of the aerosol measurements exhibit a completely different pattern. Fig. 10 shows the Spearman's rank correlation coefficients of the topographical parameters with the minimum (Dmin) and maximum (Dmax) of the monthly diurnal cycle strength as well as with the seasonal cycle (Season) of the aerosol parameters. The greatest correlation is found between the amplitudes of the diurnal cycles and the latitude for all three aerosol parameters. This anti-correlation is particularly marked for the number concentration diurnal cycles. At low latitudes, the stronger insolation enhances the surface temperature and the thermal convection leading to stronger diurnal cycles, particularly in summer, and the convective flow is less likely to be inhibited during the winter due to longer daylight hours. Together these effects result in a greater ABL influence year round and explain the high correlations with the diurnal cycle amplitude. The high correlation with number concentration can also be explained by the coupling between stronger insolation and a greater ABL influence at high altitudes that promotes NPF (Bianchi et al., 2016).

The ABL-TopoIndex is s.s. correlated with the diurnal cycle minimal and maximal strengths of the absorption coefficient. This correlation is once again principally due to the hypso% and, to a lower extent, the LocSlope and G8. The only s.s. correlation with station altitude is found for the scattering coefficient seasonal cycle. Similar to the correlation with the percentiles, there is a high anti-correlation between the particle number concentration diurnal cycles and G8 suggesting that the slope steepness in the vicinity of the stations inhibited both the transport of polluted air masses and NPF. The lack of s.s. correlations with the seasonal cycles can probably be attributed to the relatively small time period (2-3 years) covered by most of the datasets leading to difficulties in the statistical determination of a yearly periodicity due to inter-annual variability.

The NCOS station has a very high ABL-TopoIndex due to its situation on a high altitude plain near a vast lake and was therefore excluded from this analysis due to its clear outlier status. The mid-altitude stations (HPB, MSY and ZEP all situated under 1000 m) have ABL-TopoIndex values higher than 4 and were also not included in Fig. 9 and 10 since their seasonal and diurnal cycles exhibit different features than the high altitude stations (see Sect. 4.1). However, if one were to include these three lower altitude sites in the correlation with percentiles (i.e., Fig. 9) the results would be similar albeit with lower s.s. and correlation coefficients than when the analysis only includes the high altitude stations. However, the s.s. correlations of the diurnal cycle amplitudes with the ABL-Topoindex (i.e., Fig. 10) is lost when the low altitude stations are included. The Kendall's tau correlation analysis leads to the same conclusions (see Table S2).



### 3.6. Flow paths as a function of ABL heights

Figures 4a and 5a present the main hydrological flow paths calculated for the BEO and PYR stations. To study the possible water flow paths for an ABL covering part of the terrain, the topography of the lowest altitude grid points were modified by
imposing an added ABL as a sea over the topography and decreasing exponentially with altitude up to 400 m under the station altitude (see the schematic of added ABL on Fig. 2). To avoid instability in the flow path calculation for domains with constant height, a random roughness smaller than 40 m was added to the ABL height. Due to some edge effects for finite domains, the flow paths calculated for different domain sizes are not always similar. The flow paths were therefore calculated for 3 domain sizes (300x300 km$^2$, 500x500 km$^2$ and 10000x10000 km$^2$) and for assumed ABL height values from
200 m a.s.l. to a maximum of 400 m under the station altitude in 100 m vertical intervals. Here again, no wind components were taken into account.

Figure 11 shows these main flow paths as a function of the ABL height for the Alps, the Pyrenees and the western North American mountain sites. For the stations in the Alps, HPB has main flow paths mostly from the North for all ABL heights corresponding to flow from central Germany. ZUG and ZSF have additionally main flow paths from West and East at low
ABL heights and a flow path coming from the Adriatic Sea from the southeast for ABL heights greater than 2000 m. This flow path is frequently observed at ZSF during summer time and is related to convective upwind systems during daytime that directly depends on the duration of radiation (Birmili et al., 2009). SBO has flow paths from almost all directions: central Germany, east Austria and the Adriatic Sea, but not from the alpine domain westwards. The JFJ station has main flow paths from the whole Swiss plateau for all ABL heights and also from the Po valley for high altitude ABL. This is
confirmed by several analysis leading to measurement of highly polluted air masses advected from southerly regions to the JFJ (Lugauer et al., 1998). The example of the three stations in the Pyrenees (Fig. 10b) shows that PDM has flow paths coming the north of the Pyrenees for all ABL heights whereas MSA primarily has flow paths from the south of the Pyrenees. In North America, WHI and MBO both have flow paths mostly from the Pacific coast. SPL and MZW in the Rocky Mountains have flow paths mostly from the low lands west of the Rocky Mountains, whereas NWR flow paths reach the low
lands east of the Rockies.

CHC's main flow paths as a function of ABL height are plotted on Fig. 12a. The results show primary flow paths going towards the north for low ABL heights (cyan lines) and towards the NE and E for higher ABL heights (fine red and magenta lines). For ABL heights greater than 2500 m (red lines), some flow paths from the Altiplano and parallel to the mountainous range are also found. Fig. 12b shows a 2% sample of all the 96 h back-trajectories modeled at CHC between 2012 and 2014.
During the dry season (May to July), most of the trajectories come from the NW Altiplano whereas during the wet season (December to April), the greatest number of back-trajectories come from the Yungas (i.e. from the tempered valleys) in north and northeast from CHC. While the trajectories circulating around the mountain range from S of the Illimani and going over the pass on each side of the Charquini are also depicted by the flow path analysis (Fig. 12a), the trajectories getting





around the Huayna Potosi from the N are not found. Moreover winds from the Altiplano are often measured at CHC, leading to the back-trajectories from W and SW. The similarities and differences between the calculated main flow paths with an added PBL and the back-trajectories analysis reveals both the potential and the limitations of the topographical analysis that does not take into account the wind and the main synoptic systems.

## 4.    Discussion

In this section the assessments, improvements and applications of the ABL-TopoIndex are discussed. First the possible species and phenomena enabling the estimation of the ABL influence are summarized and the occurrence of diurnal and seasonal cycles as a function of the station elevation are discussed. Second, the significance of the correlations between the topographical and the aerosol parameters are further interpreted. Finally, possible additional parameters that could increase the significance and the application of the ABL-TopoIndex are mentioned, in addition to the criteria relevant for choosing future sites to sample FT air masses.

### 4.1.   Using measurements to assess the ABL influence

In order to test the relevance of the ABL-TopoIndex, it is first necessary to find a parameter commonly measured at high altitude stations that can be used as an ABL tracer. Pollutants emitted at the Earth's surface and having a (typically) minimal concentration in the FT could act as potential tracers of the ABL influence. Our results showed that of the three aerosol parameters tested in this study (number concentration, absorption coefficient and scattering coefficient), absorption coefficient provided the most robust indicator of ABL-influence based on the ABL-TopoIndex values.  Other possible candidates for testing the ABL-TopoIndex include the aerosol mass concentration, size distribution and chemical composition, the water vapor and the trace gases concentrations (e.g., $CO_2$, PAN, $NO_x$, $NO_y$, $O_3$, $SO_2$, isotopologue ratio of water vapor) and the radon[222] concentration. These parameters have been used in different studies to provide information about the seasonal and diurnal cycles (e.g., Collaud Coen et al., 2011; Griffiths et al., 2014; Marinoni et al., 2010; McClure et al., 2016; Okamoto and Tanimoto, 2016; Pandolfi et al., 2014; Ripoll et al., 2015; Zellweger et al., 2009), the sources and transport of aerosol to the site ( e.g., Cuevas et al., 2013; García et al., 2017; Pandey Deolal et al., 2014; Ripoll et al., 2014), the local orographic flows and the effect of the synoptic- and meso-scale weather types ( e.g., Bonasoni et al., 2010; Gallagher et al., 2011; González et al., 2016; Henne et al., 2005; Kleissl et al., 2007; Tsamalis et al., 2014; Zellweger et al., 2002). All of the extensive aerosol parameters, the radon[222], water vapor concentration, $NO_3$ and organics have been shown to be correlated with ABL transport whereas $CO/NO_y$ and $NO_x/NO_y$ ratios are anti-correlated (Legreid et al., 2008; Zellweger et al., 2002). Because there are variable pollution levels in the vicinity of the stations, a single absolute value of a pollutant cannot be used to evaluate the ABL-TopoIndex (or ABL influence in general) when considering multiple high altitude stations. An inventory of the proximate pollution sources would be required before directly using absolute pollutant





concentrations as indicators of ABL influence at high altitude sites. This problem can be avoided by instead considering dynamical parameters such as the various temporal cycles.

At most of the high altitude stations, a seasonal cycle in ABL-indicator species is observed. The maximum values of the seasonal cycles are correlated with ABL transport and typically occur in summer or in the pre-monsoon season while the minimum of the seasonal cycle occurs in winter or monsoon seasons. Usually the spring leads to higher concentration of ABL species than the autumn. These seasonal cycles are explained by the stronger thermal heating of the soil which induces convection and buoyancy in summer and by the atmospheric cleaning effect of precipitation during the monsoon. It would be expected that stations continuously situated in the ABL throughout the year could exhibit different seasonal cycles than FT sites due to the seasonal modification of the sources and/or of the synoptic and meso-scale meteorological conditions (see for example the difference between HPB and JFJ on Fig. S2). In contrast, a station located such that it stayed continuously in the FT would have a seasonal cycle that depends only on long-range, high altitude transport climatology (e.g., long-range transport of Asian dust and pollution at MLO in spring (Collaud Coen et al., 2013), North-America ABL transport to IZO through westerlies in spring (García et al., 2017), and dust events in EU spring and autumn (Collaud Coen et al., 2004)). Since seasonally changing parameters (e.g., temperatures, cloud cover, solar radiation, wind speeds, surface albedo) were not studied and since the length of most of the time series are too short to smooth these effects, the ABL-TopoIndex will probably not represent an overall picture of ABL influence except at seasonally invariant sites (e.g., very low latitude sites).

The typical diurnal cycle of ABL pollutants at high altitude stations that are partially influenced by the ABL consists of a minimum in the early morning (4h-6h LTC) followed by an increase of the compound with a maximum in the late afternoon (15-17h LTC) and a decrease during the night. If the ABL influence is mostly due to orographic winds, upslope/valley winds begin to flow some hours after sunrise and downslope/mountain winds initiate after the occurrence of negative vertical heat flux. Stations always situated in the FT should exhibit no systematic diurnal cycles whereas the stations always situated in the ABL often show various diurnal cycles that can be explained by the behavior of local sources, the diurnal cycle of the ABL height and/or local meteorological conditions. At high elevation, high latitude stations the diurnal cycle typically vanishes during winter but is clearly present during summer, spring and, to a lesser extent, autumn. For stations at lower altitude that stay in the ABL (or CBL, SBL or RL) during the whole day in summer (e.g., MSA (Pandolfi et al., 2013), HPB and PUY (Hervo et al., 2014)), the diurnal cycle may also vanish during that period.

Testing the ABL-TopoIndex using pollutant diurnal cycles is further complicated by the presence of the residual layer (RL) that keeps the pollutants brought to high altitudes during the previous days at those elevated levels during the nighttime. The climatology of the RL height usually exhibits a similar seasonality as the ABL height, with a maxima in summer (or pre-monsoon) season and a minima in winter (Birmili et al., 2009, 2010; Collaud Coen et al., 2014; Wang et al., 2016). Further, the RL also has similar dependency as the ABL as a function of latitude. The RL's maximum height also depends, therefore, on the duration of the incoming radiation. The RL pollutant concentrations are much higher than nighttime FT



concentrations, leading to less marked diurnal cycles in summer than in spring (Blay-Carreras et al., 2014; Collaud Coen et al., 2011; Hallar et al., 2016; Hervo et al., 2014).

Recently, the influences of the local and of the more regional or meso-scale ABL at the JFJ were separated by differentiating the Local Convective Boundary Layer (LCBL) height from the high altitude aerosol layer (Poltera et al., 2017). The LCBL was found to rarely influence the JFJ research station (never in winter, 4% of the time which corresponds to 22% of the days in summer), whereas the continuous aerosol layer has a large influence on the JFJ pollutant concentrations (21% of the time in winter and 41% of the time corresponding to 77% of the days in summer). This suggests that the mechanisms explaining the heights of the LCBL and the more horizontally extended aerosol layer have different causes and do not follow the same diurnal pattern. This phenomenon will be more pronounced at continental high altitude stations than at marine isolated island stations since the marine ABL is less prone to strong diurnal cycles.

## 4.2 Correlation between the topography and the aerosol parameters

The correlations between topographical and aerosol parameters presented under Sect. 3.5 can now be further discussed in light of the pollutant temporal cycles. Among the aerosol parameters studied here, the absorption coefficient, which is primarily due to the presence of black carbon emitted from combustion processes, is the best tracer for anthropogenic pollution and biomass burning and consequently of ABL influence. It is therefore expected that a better correlation will be obtained between the topography parameters increasing the ABL influence and the absorption coefficient. The ABL-TopoIndex reflects this correspondence, particularly through the contribution of the hypso% parameter (recall that hypso% represents the relative altitude of a station in its mountain range). The best correlation for both ABL-TopoIndex and hypso% is found for the $5^{th}$ percentile of the absorption coefficient, since the minima of the aerosol loading is a better tracer of the lowest ABL influence, whereas the maxima is much more dependent on source intensity and special events. Similar to this result, a clear correlation was also found between the continuous aerosol layer maximum height and the absorption coefficient measured in-situ at the JFJ (Fig. 8 in (Poltera et al., 2017)). The absorption coefficient amplitudes of the diurnal cycle are also the only aerosol parameter having a s.s. correlation with the ABL-TopoIndex.

It is more difficult to directly tie scattering and number concentration to the ABL incursions. This is because the formation of new particles and their subsequent growth are well-known to be very efficient processes at high altitudes due to the high insolation and the low temperature. Moreover, the NPF is also enhanced by local thermal winds and forced convection due to favorable changes in thermodynamic conditions (Boulon et al., 2011; Rose et al., 2015). It was found at the JFJ and confirmed at other stations that new particle formation, and particularly strong nucleation events, occur mostly when the air masses were in contact with the ABL within 2 days before arriving at high altitudes (Bianchi et al., 2016). New particle formation and subsequent growth of the particles have a large impact on the number concentration and its temporal cycles and a smaller influence on the scattering coefficient. The parameters describing the local topography (G8 and LocSlope) have the greatest correlation with the number concentration and are probably more relevant to the local CBL transport than




to the longer range continuous aerosol layer as defined in Poltera et al. (2017). The number concentration and, to a lesser extent, the absorption coefficient percentiles and diurnal cycles are anti-correlated with the local (G8: 2-4 km) and regional (LocSlope and hypsD50: 10 km) slopes, suggesting there is an increase of particle number concentration when there are small altitude differences and gentle slopes around the station. This dependence on the ease of local transport can be

explained by larger scale transport to the station not only of aerosol, but also of gaseous precursors for new particle formation and of newly formed particles at lower elevations. The greater correlation of slope with the number concentration rather than with the absorption coefficient can be explained both by the very scarce sources of black carbon in the near vicinity of most of the high altitude stations and by the smoother pressure decrease leading to more condensation processes and nucleation.

The aerosol diurnal cycles are influenced by numerous phenomena (see Sect. 4.1) leading to a non-trivial relationship with the ABL influence. There are consequently few correlations between topography parameters and the diurnal cycles. The clearest correlation is the influence of the insolation on the aerosol diurnal cycles amplitudes. This dependence between the latitude and the aerosol concentration was already mentioned by Kleissl et al. (Kleissl et al., 2007) and is easily understandable, the convection and the new particle formation being directly dependent on the solar radiation intensity.

**4.3 Improving and applying the ABL-TopoIndex**

The choice of the 5 parameters included in the ABL-TopoIndex was initially based on several assumptions relating the topography to the ABL influence (see Sect. 2.3). Several other parameters such as the topographical wetness index, the catchment area, the accumulation, dispersion and transit percentages, the hypsometric index and the prominence were tested but were finally eliminated as being not relevant for various reasons. Indeed, most of the parameters comprising the ABL-

TopoIndex exhibit some correlations with aerosol parameters. The hypso% values derived from the hypsometric curve clearly explain the greatest variance in the aerosol optical properties and particularly in the absorption coefficient (e.g., the primary marker of anthropogenic ABL pollution). Thus, if a single topographical parameter should be chosen to describe the ABL influence, the hypso% is the best candidate. It also seems evident that the topographical parameters linked to the steepness and the altitude differences are clear indicators for NPF. The DBinv seems to be the least explanatory parameter in

terms of ABL influence and this large scale parameter should probably be bounded with a source inventory to increase its relevance for identifying boundary layer influence. However, the aerosol parameters and, particularly, the absorption coefficient cannot be considered as unique tracers of the ABL. Analysis of other ABL marker (gaseous species, radon, wind turbulences, etc.) can provide information on additional transport mechanisms which would allow for refinement of this topographic analysis.

The East-oriented slopes are heated early during the day and have therefore a greater contribution to the thermal convection and the associated valley winds. A parameter weighting the east slope area could therefore be added to the ABL-TopoIndex. The various geomorphologies of the mountainous ranges included in this study also raise the question of whether the stations should all be combined together for analysis as was done here, or if a morphological parameter should instead be found for





each massif. The mountain steepness (at a larger scale than LocSlope and G8) and at all altitudes also determines the necessary velocity for the wind to cross the mountains and could be an additional parameter. Finally, future studies should attempt to build a direction dependent ABL-TopoIndex that also takes into account the topography of each valley up to the meso-scale range.

It's important to understand the FT versus ABL influence on historical data sets from established high altitude observatories. The ABL-TopoIndex is one tool that can help elucidate the different influences. A further improvement could include an angular dependency of the ABL-TopoIndex allowing quantifying the potential direction of the maximal ABL influence. The ABL-TopoIndex may also be useful *a priori* in locating measurements for a field campaign or identifying potential sites for long-term observatories if FT measurements are the goal. For example, in-situ aerosol measurements are done at IZO at an

altitude of 2373 m whereas aerosol optical depth and water vapor isotopologues measurements are done at TDE at 3538 m on the same volcano. TDE has a much lower ABL-TopoIndex than IZO and consequently TDE's measurements are more likely to represent the FT. The topography around the PYR station suffers from several inconveniences (see Fig. 7 and Sect. 3.1) leading to a high ABL influence. Even if the choice of the actual site is driven by compelling and practical logistical arguments, other emplacements at similar altitudes would have ensured lower pollution impact. Finally, some stations such

as BEO (2925 m), HAC (2314 m) and MWO (1916 m) are not situated at very high altitudes but present excellent locations for FT sampling. Obviously there are other issues to consider when deploying instruments as well (e.g., ease of access, power availability, presence of local pollution sources, etc.), but the ABL-TopoIndex is one factor that could be considered to maximize the potential for FT sampling.

## Conclusion

The ABL-TopoIndex is a topographical index based on the hypsometric curve, the slope of the terrain around the station and the drainage basin for convection. It allows one to rank the high altitude stations as a function of their ABL influence or to optimize choice of site location for FT sampling. High altitude stations situated on volcanic islands, the highest stations in the Alps, in the Andes and in the Pyrenees have low ABL-TopoIndex values. Stations situated at or near the summit of their mountainous ranges such as BEO and HAC also have low ABL-TopoIndex values. Stations situated at altitudes between

4000 and 5500 m in the Himalaya and the Tibetan Plateau have high ABL-TopoIndex values due to their relatively low position compared to the Himalayan summits. Statistically significant correlations between the ABL-TopoIndex and the aerosol parameters measured at high altitude sites allow validation of the methodological approach. The greatest correlations are found with the minima of the aerosol parameters that represent the minimal ABL influence or, in other words, the most likely FT air masses. The maxima of aerosol parameters are more representative of the intensity of aerosol sources and of

advection of air masses with high aerosol concentrations. If high altitude stations undergo daytime ABL air influence due to convection, a pronounced diurnal cycle of aerosol parameters is usually measured. The variance of the diurnal cycles is however mostly explained by the latitude of the station, leading to the conclusion that the sun radiation intensity and duration





drive the aerosol diurnal cycle. Finally, the determination of the main convective flow paths with a constant ABL height superimposed on the topography allows one to find the regions possibly influencing the high altitude sites.

**Acknowledgments**

The authors greatly acknowledge:

- W. Schwanghart, the programmer of the TopoToolBox for putting his codes as a freeware and for all the kind and always rapid support he offers.
  - CHC: the Chacaltaya consortium and all Laboratory for Atmospheric Physics at UMSA for taking care of the station and collecting the data. Swedish participation was supported by The Swedish Foundation for International Cooperation in Research and Higher Education (STINT) and The Swedish research council FORMAS.
- CMN: the European Commission funded ACTRIS, ACTRIS-2 EU project and NEXTDATA National project funded by MIUR.
  - IZO: Global Atmospheric Watch program funded by AEMET and by the project AEROATLAN (CGL2015-66299-P, funded by the Ministry of Economy and Competitiveness of Spain and the European Regional Development Fund - ERDF).
- JFJ: International Foundation High Altitude Research Stations Jungfraujoch and Gornergrat (HFSJG) and the Federal Office of Meteorology and Climatolofy, MeteoSwiss, within the Swiss program of the Global Atmosphere Watch (GAW) of the World Meteorological Organization, funding from the European Union's Horizon 2020 research and innovation programme under grant agreement No 654109 (ACTRIS2), and the Swiss State Secretariat for Education, Research and Innovation (SERI) under contract number 15.0159-1. The opinions expressed and
arguments employed herein do not necessarily reflect the official views of the Swiss Government.
  - LLN: by Taiwan Environmental Protection Administration.
  - MSA and MSY: MINECO (Spanish Ministry of Economy and Competitiveness), the MAGRAMA (Spanish Ministry of Agriculture, Food and Environment), the Generalitat de Catalunya (AGAUR 2014 SGR33 and the DGQA) and FEDER funds under the PRISMA project (CGL2012- 39623-C02/00). This work has received funding
from the European Union's Horizon 2020 research and innovation programme under grant agreement No 654109. Marco Pandolfi is funded by a Ramón y Cajal Fellowship (RYC-2013-14036) awarded by the Spanish Ministry of Economy and Competitiveness
  - MLO, NWR and SUM: the NOAA observatory staff at each station and the NOAA Climate Program Office's Atmospheric Chemistry, Carbon Cycle and Climate (AC4) program
- MUK: Ministry of Foreign Affairs of Finland, Academy of Finland (Project number 264242) for the financial support and an esteemed collaboration of FMI and TERI.





- NCOS: Max Planck Institute for Chemistry and Chinese Academy of Sciences (Project SKLCS-ZZ-2017 and KJZD-EW-G03-03).

- PDI: Federal Office of Meteorology and Climatology MeteoSwiss through the project Capacity Building and Twinning for Climate Observing Systems (CATCOS) Phase 2, Contract no. 81025332 between the Swiss Agency
for Development and Cooperation and MeteoSwiss

- PEV: Support from Swedish Research Council (Vetenskaprådet) and Swedish International Development Cooperation Agency (SIDA)

- PDM: Pyrenean Platform for Observation of the Atmosphere P2OA (http://p2oa.aero.obs-mip.fr), University Paul Sabatier, Toulouse, France, and CNRS (Centre National de la Recherche Scientifique).

- OMP: United States National Oceanic and Atmospheric Administration (NOAA) grants NA16GP1658, NA86GP0325 and NA030AP430002; United States National Science Foundation (NSF) grants ATM-0215843 and INT-0110397; Portuguese Foundation for Science and Technology (FCT) grants POCTI-32649-CTA-2000; SFRH/BD/9049/2002; and Portuguese Regional Govern of Azores and especially Professor Richard Honrath (1961-2009).

- PYR: the framework of the UNEP — ABC (Atmospheric Brown Clouds) and EvK2CNR — SHARE (Stations at High Altitude for Research on the Environment) projects and the appreciated collaboration with the technical Nepalese staff.

- SBO: TU Wien, Commission of Climate and Air Quality - ÖAW, Local Government of Salzburg - Immissionsschutz, Umweltbundesamt GmbH and Aerosol d.o.o .

- SPL: Randolph Borys, Ian McCubbin, Douglas Lowenthal, Peter Atkins, and Joe Messina, the US National Science Foundation (grant AGS-0079486), the Steamboat Ski Resort and the Desert Research Institute, a permittee of the Medicine-Bow Routt National Forests

- TLL: Federal Office of Meteorology and Climatology MeteoSwiss through the project Capacity Building and Twinning for Climate Observing Systems (CATCOS) Phase 2, Contract no. 81025332 between the Swiss Agency
for Development and Cooperation and MeteoSwiss, G. Wehrle from Paul Scherrer institute, the site operator Luis Valle from DGAC and to the maintenance staff from MeteoChile.

- WLG: Natrional Scientific Foundation of China (41675129), National Key Project of MOST (2016YFC0203306) and the CMA Innovation Team for Haze-fog Observation and Forecast.

- ZEP: Support from Swedish Environmental Protection Agency (Naturvårdsverket),  The Swedish research council
FORMAS, Knut and Alice Wallenberg Foundation (KWA) and Norwegian Polar Institute (NPI)

- ZUG: H.E. Scheel for performing aerosol measurements at ZUG and W. Junkermann, Karlsruhe Institute of Technology, Garmisch-Partenkirchen, Germany, who give us access to this dataset.




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



**Tables**

Table 1: List of station names, acronyms, latitude [°], longitude [°], altitude [m], their mountain range or region and continent. If aerosol time series were used, the station name is given in bold. The references principally describe the station measurement program and, particularly, the aerosol parameters measured.

| Station | Latitude | Longitude | Altitude | Massif | Continent | References |
|---|---|---|---|---|---|---|
| **HPB** <br> **Hohenpeissenberg, Germany** | 47.8015 | 11.0096 | 985 | | | (Flentje et al., 2015) |
| **JFJ** <br> **Jungfraujoch, Switzerland** | 46.5477 | 7.985 | 3580 | | | (Bukowiecki et al., 2016) |
| **SBO** <br> **Sonnblick, Austria** | 47.0539 | 12.951 | 3106 | Alps | | (Schauer et al., 2016) |
| **ZSF** <br> **Schneefernhaus, Germany** | 47.4165 | 10.9796 | 2671 | | | (Birmili et al., 2009) |
| **ZUG** <br> **Zugspitze, Germany** | 47.4211 | 10.9859 | 2962 | | | -- |
| **MSA** <br> **Montsec, Spain** | 42.05 | 0.7333 | 1570 | | **Europa** | (Ealo et al., 2016; Pandolfi et al., 2014; Ripoll et al., 2014) |
| **MSY** <br> **Montseny, Spain** | 41.7795 | 2.3579 | 700 | Pyrenees | | (Pandolfi et al., 2011) |
| **PDM** <br> **Pic du Midi, France** | 42.9372 | 0.1411 | 2877 | | | (Gheusi et al., 2011, Hulin et al., 2017) |
| **BEO** <br> **Moussala, Bulgaria** | 42.1792 | 23.5856 | 2925 | Balkan | | (Angelov et al., 2016) |
| **CMN** <br> **Monte Cimone, Italy** | 44.1667 | 10.6833 | 2165 | Apennines | | (Cristofanelli et al., 2016; Marinoni et al., 2008) |
| HAC <br> Mount Helmos, Greece | 37.9843 | 22.1963 | 2314 | Peloponnese | | |

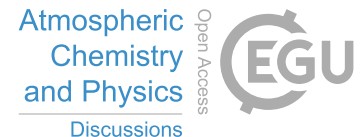



| | | | | | | |
|---|---|---|---|---|---|---|
| **PUY**<br>**Puy de Dôme, France** | 45.7723 | 2.9658 | 1465 | Central massif | | (Venzac et al., 2009) |
| **CHC**<br>**Chacaltaya, Bolivia** | -16.200 | -68.100 | 5320 | | | (Andrade et al., 2015) |
| LQO<br>La Quiaca Observatorio, Argentina | -22.100 | -65.599 | 3459 | | | |
| **PEV**<br>**Pico Espeje, Venezuela** | 8.5167 | -71.05 | 4765 | Andes | South America | (Hamburger et al., 2013; Schmeissner et al., 2011) |
| **TLL**<br>**Cerro Tololo, Chile** | -30.1725 | -70.7992 | 2220 | | | Velasquez, 2016 |
| MZW<br>Mount Zirkel Wildness, USA | 40.5433 | -106.6844 | 3243 | | | |
| **NWR**<br>**Niwot Ridge, USA** | 40.04 | -105.54 | 3035 | | | |
| **SPL**<br>**Steamboat, USA** | 40.455 | -106.744 | 3220 | Rocky | | (Hallar et al., 2015) |
| YEL<br>Yellowstone NP, USA | 44.5654 | -110.4003 | 2430 | | | YEL<br>Yellowstone NP, USA |
| APP<br>Appalachian State University, USA | 36.2130 | -81.6920 | 1076 | | North America | |
| SHN<br>Shenandoah National Park, USA | 38.5226 | -78.4358 | 1074 | Appalachian | | |
| MBO<br>Mount Bachelor, USA | 43.979 | -121.687 | 2743 | | | |
| MWO<br>Mount Washington | 44.2703 | -71.3033 | 1916 | | | |
| **WHI**<br>**Whistler, Canada** | 50.0593 | -122.9576 | 2182 | | | (Gallagher et al., 2011) |
| HLE<br>Henle, India | 32.7794 | 78.9642 | 4517 | Himalaya | Asia | |
| LAN<br>Langtang, Nepal | 28.2200 | 85.6200 | 3920 | | | |





| **MUK** **Mukteshwar, India** | 29.4371 | 79.6194 | 2180 | | (Hyvärinen et al., 2009; Panwar et al., 2013) |
|---|---|---|---|---|---|
| **NCOS** **Nam Co, China** | 30.7728 | 90.9621 | 4730 | | (Zhang et al., 2017) |
| **PYR** **ABC Pyramid, Nepal** | 27.9578 | 86.8149 | 5079 | | (Bonasoni et al., 2010; Marcq et al., 2010; Marinoni et al., 2010) |
| SZZ Shangrimla ZhuZhang, China | 27.9998 | 99.4266 | 3583 | | |
| **WLG** **Mount Waligan, China** | 36.2875 | 100.8963 | 3810 | Tibetan Plateau | (Andrews et al., 2011) |
| **PDI** **Pha Din, Vietnam** | 21.5728 | 103.5160 | 1466 | -- | -- |
| FWS Mount Fuji, Japan | 35.3606 | 138.7273 | 3776 | | |
| HPO Mount Happo, Japan | 36.6972 | 137.7989 | 1850 | Japan Alps | |
| MTA Mount Takayama, Japan | 36.1461 | 137.4230 | 1420 | | |
| **IZO** **Izaña, Spain** | 28.309 | -16.4994 | 2373 | Atlantic | (Rodríguez et al., 2012) |
| **LLN** **Mount Lulin, Taiwan** | 23.4686 | 120.8736 | 2862 | Pacific | (Hsiao et al., 2017) |
| **MLO** **Mauna Loa, USA** | 19.5362 | -155.576 | 3397 | Pacific | (Bodhaine, 1995) |
| **OMP (previously PICO-NARE)** **Pico Mountain, Azores, Portugal** | 38.4704 | -28.4039 | 2225 | Atlantic | (Fialho et al., 2004) |
| RUN Ile de la Réunion, France | -21.0795 | 55.3831 | 2160 | Indian | |
| TDE Izaña, Spain | 28.2702 | -16.6385 | 3538 | Atlantic | |



| | | | | | | |
|---|---|---|---|---|---|---|
| ASK<br>Assekrem, Algeria | 23.2667 | 5.6333 | 2710 | **Africa** | | |
| MKN<br>Mount Kenia | -0.0622 | 37.2972 | 3678 | | | |
| **SUM**<br>**Summit, Arctic** | 72.58 | -38.48 | 3238 | **Arctic** | (Backman et al., 2016) |
| **ZEP**<br>**Zeppelin Observatory, Norway** | 78.9067 | 11.8893 | 475 | | (Tunved et al., 2013) |

Table 2: Extrema, median and mean of the topographical parameters for the 46 stations studied.

| Parameter | min | median | mean | max |
|---|---|---|---|---|
| ABL-TopoIndex | 0.098 | 1.42 | 3.55 | 32.25 |
| Hypso% [%] | 0.0002 | 1.8 | 12.0 | 58.2 |
| HypsoD50 [m] | -154 | 1487 | 1549 | 4633 |
| LocSlope [Mm$^{-1}$] | 1.6 | 89 | 99 | 288 |
| G8 [tangent] | 0.0024 | 0.1797 | 0.2025 | 0.5253 |
| DBinv [km$^2$] | 424 | 201029 | 196210 | 535518 |
| Altitude [m] | 475 | 2771 | 2802 | 5320 |
| \|Latitude\| [°] | 0.06 | 37.3 | 36.2 | 78.9 |



**Figures:**

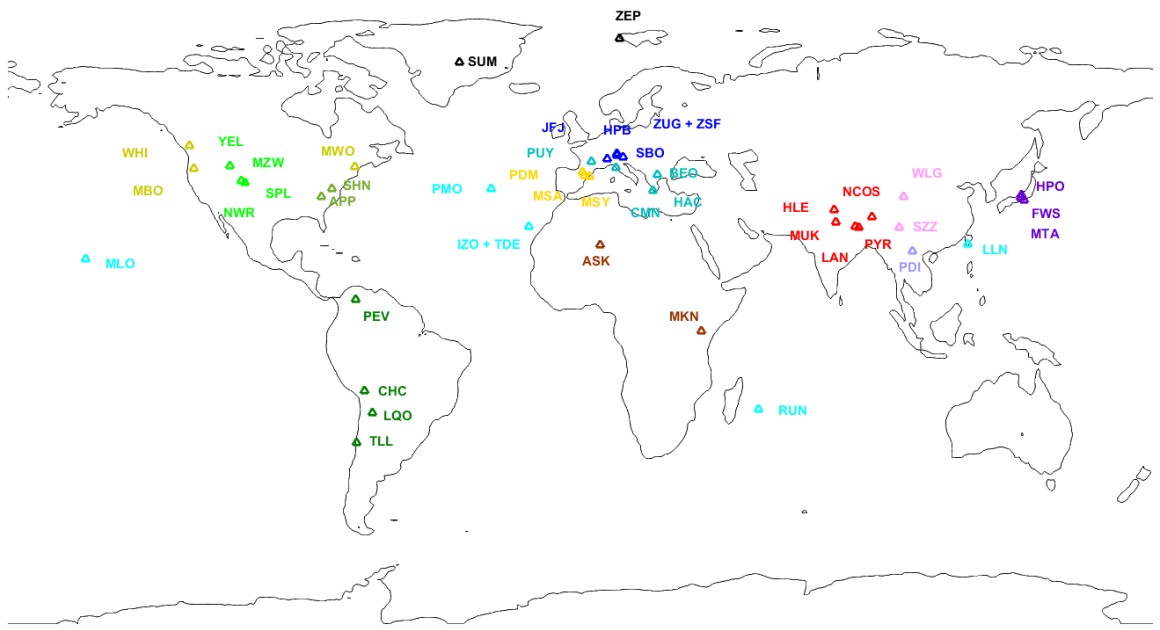

Figure 1: Map of the stations colored by their mountain ranges or region.

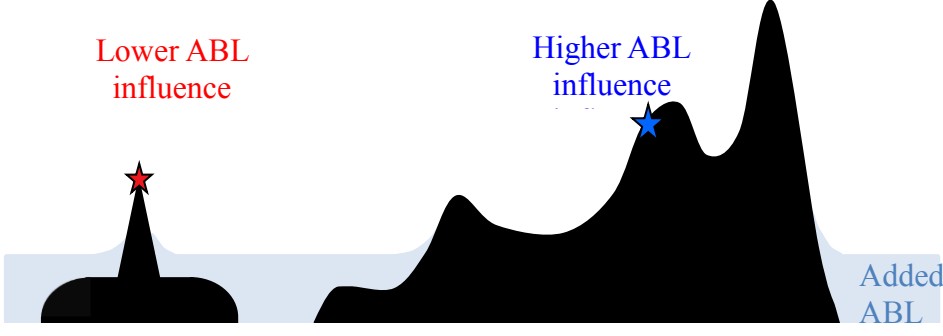

Figure 2: Schematic view of the topographical features underlying the ABL-TopoIndex. The added ABL corresponds schematically to the ABL overlaid over the real topography to calculate the main flow paths (see Sect. 3.6).





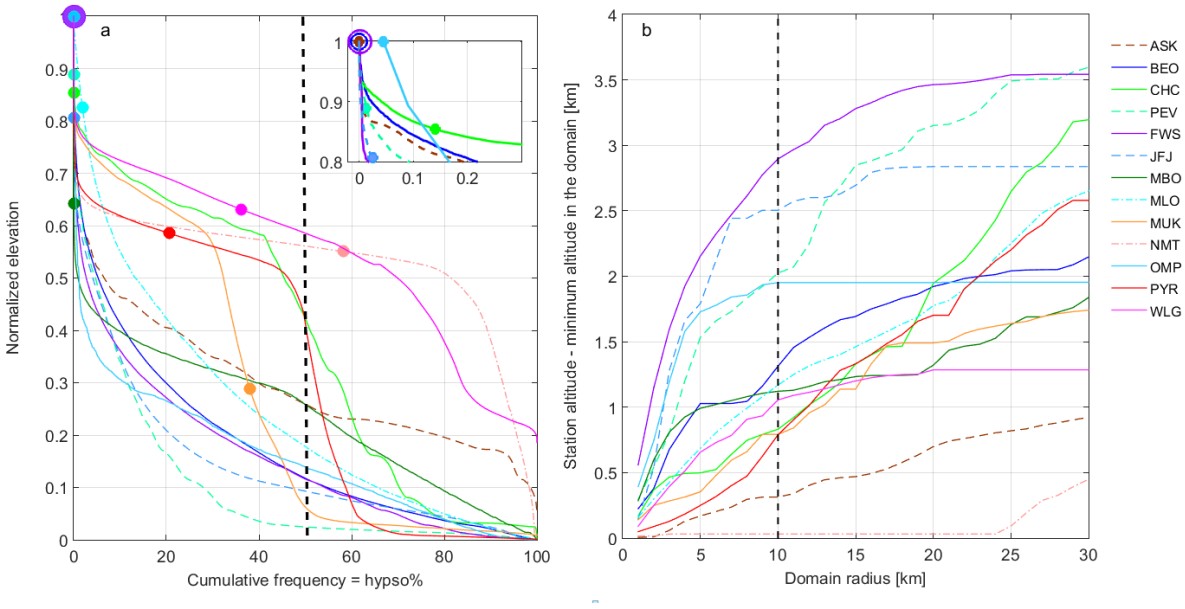

Figure 3: a) Normalized hypsocurves for some selected high altitude stations for a 750 km x 750 km domain centered on the station. The filled and open circles correspond to the normalized station elevations within the domain and indicate the value of hypso% (e.g., PYR hypso% is 21). The inset is an enlargement to show the stations with very low hypso%. The horizontal dashed line corresponds to 50% of the hypsometric curve b) Difference between the station altitude and the elevation minimum in a domain of radius R around the station as a function of R. The vertical dashed line indicates the part of the curve selected to calculate LocSlope.





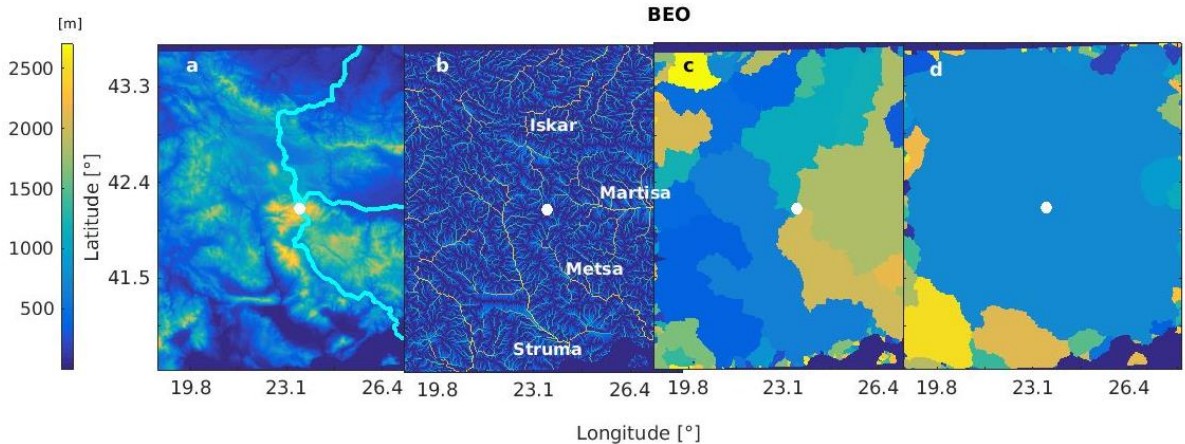

Figure 4: a) Topography on a 750x750 km$^2$ domain around BEO (Moussala, white dot) in Bulgaria. The color scale on the left only applies to Fig. 4a. b) hydrographical network, c) drainage basins calculated from the real topography, the different drainage basins are defined by various colors and d) "convective drainage basin" calculated from the inverse topography (DBinv).

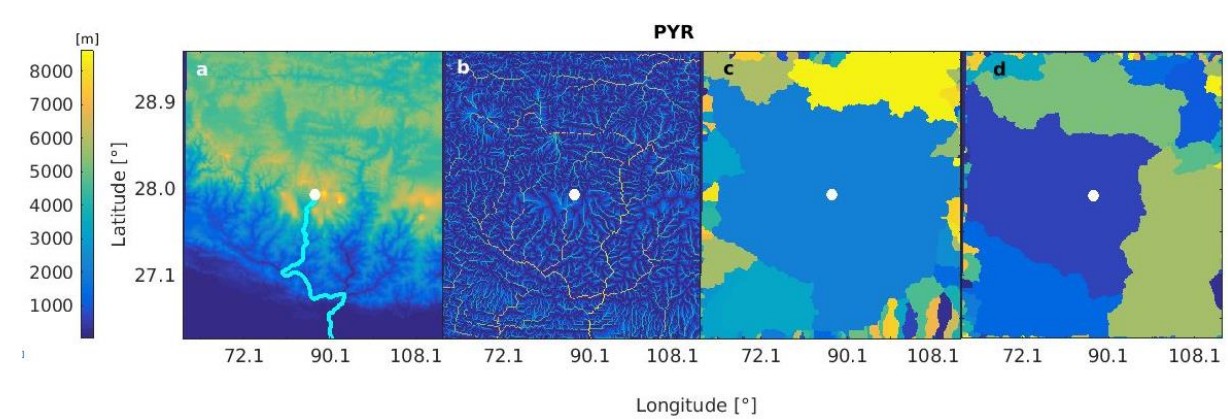

Figure 5: Idem Fig. 4 for PYR (Nepal Climate Observatory - Pyramid) station in the Himalaya, Nepal.





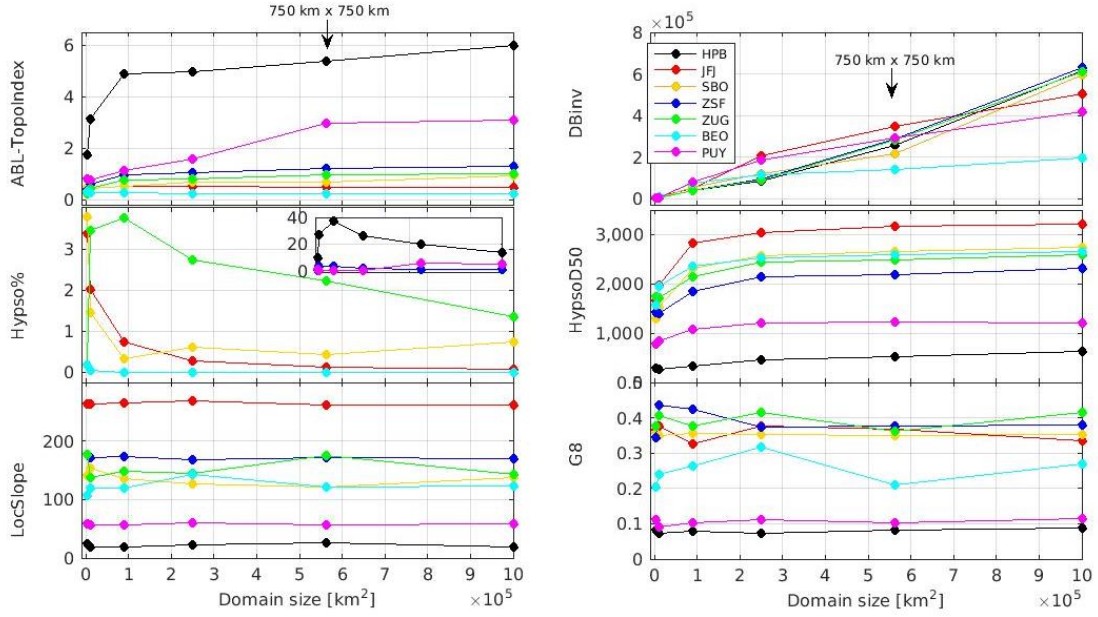

Figure 6: a) ABL-TopoIndex, b) drainage basin for convection, c) hypsometric percentage of the station elevation, d) hypsometric percentage of the station elevation minus the 50% hypsometry, e) local slope in a circle of 10 km radius centered on the station, f) gradient in elevation as a function of the domain size for some European high altitude stations.

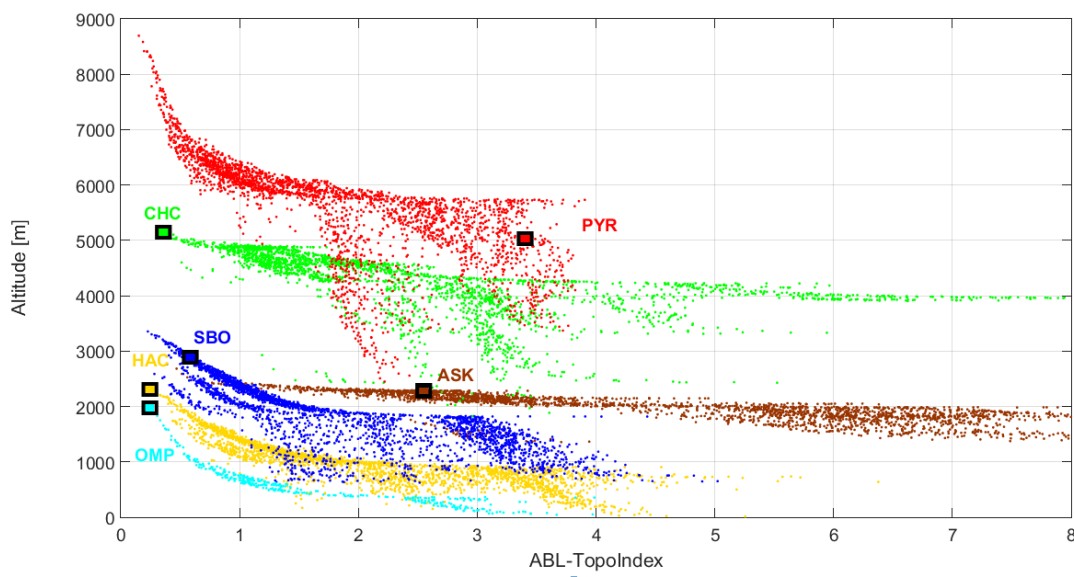

Figure 7: ABL-TopoIndex as a function of elevation of all grid points of a 625 km$^2$ domain centered on the ASK, CHC, HAC, OMP, PYR and SBO stations. The squares indicate the ABL-TopoIndex values and the altitudes of the stations.




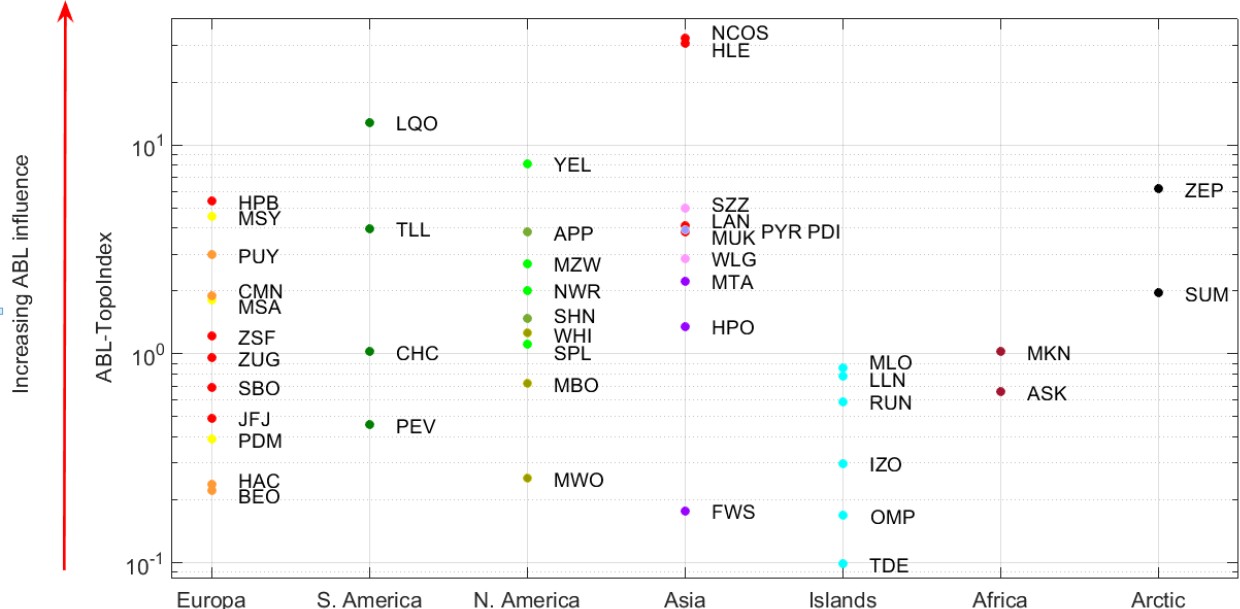

Figure 8: ABL-TopoIndex for all stations as a function of continents and mountainous ranges.




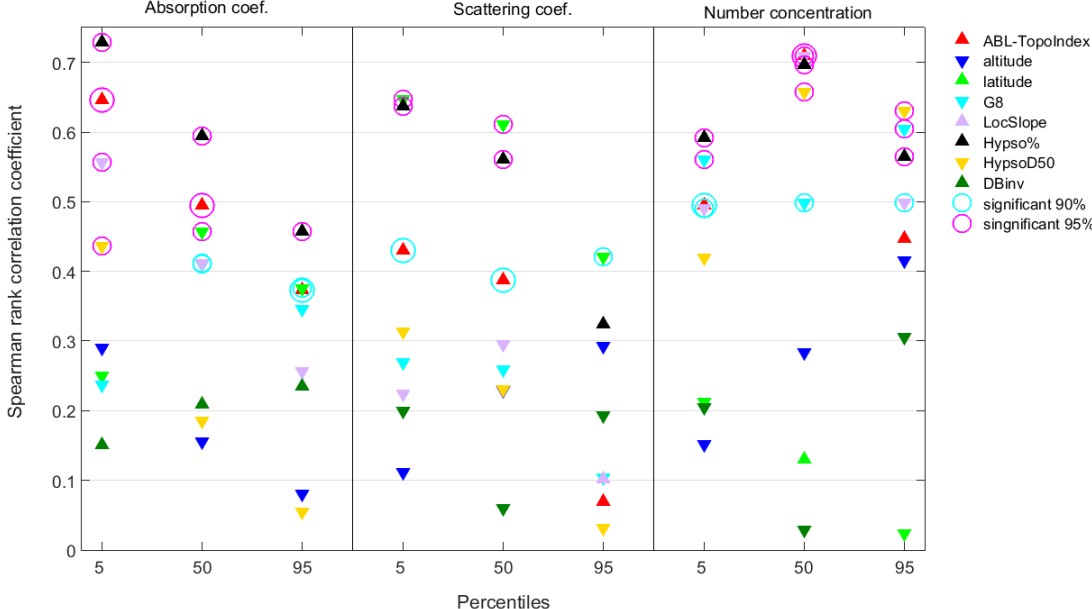

Figure 9: Spearman's rank correlation coefficient characterizing the correlation between aerosol parameters (absorption coefficient, scattering coefficient, number concentration) and various topographic parameters (the ABL-TopoIndex, station altitude, and the 5 parameters constituting the ABL-TopoIndex (G8, DB, LocSlope, hypso% and hypsoD50)). Correlations were calculated for the 5th, 50th and 95th percentiles of the aerosol parameters. Statistically significant correlation values at 95% and 90% confidence levels are surrounded by magenta and cyan circles, and the positive and negative correlations are plotted with upward and downward triangles, respectively. The correlations were performed with 21, 21 and 14 stations for the absorption coefficient, the scattering coefficient and the number concentration, respectively.





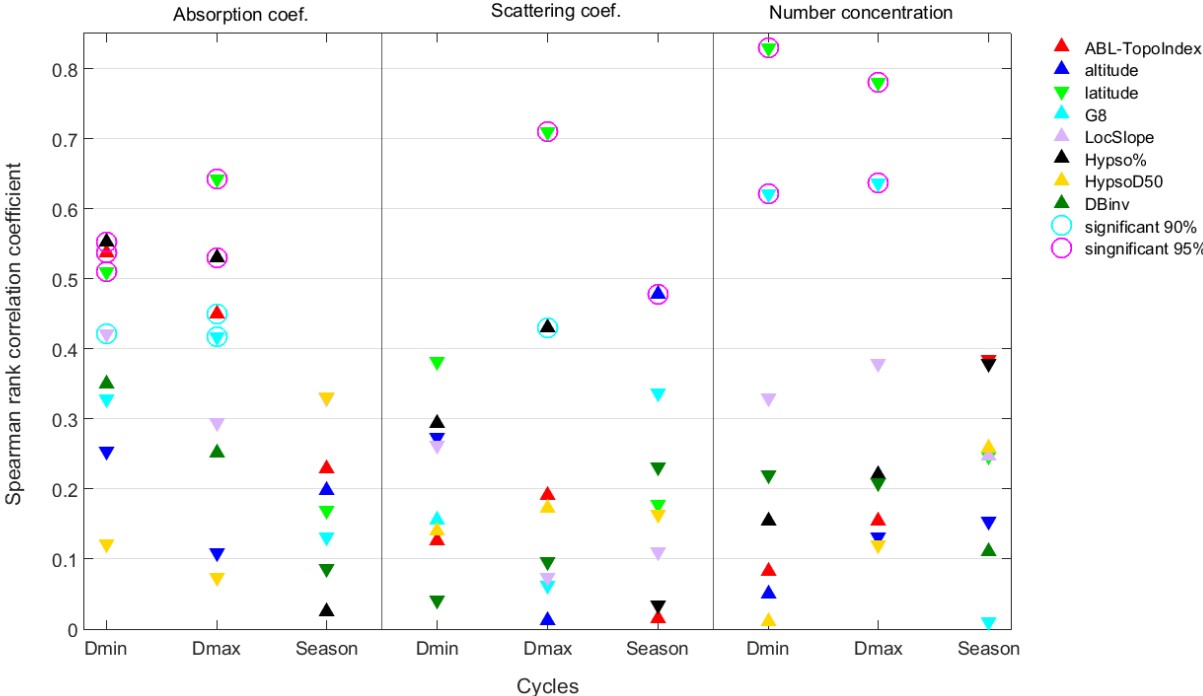

Figure 10: Spearman's rank correlation coefficient characterizing the correlation between all the topographic parameters (see Fig. 9) and the minimum and the maximum of the monthly diurnal cycles, as well as the seasonal cycle of the aerosol parameters. The correlations are performed with 19, 19 and 13 stations for the absorption coefficient, the scattering coefficient and the number concentration, respectively.



a)

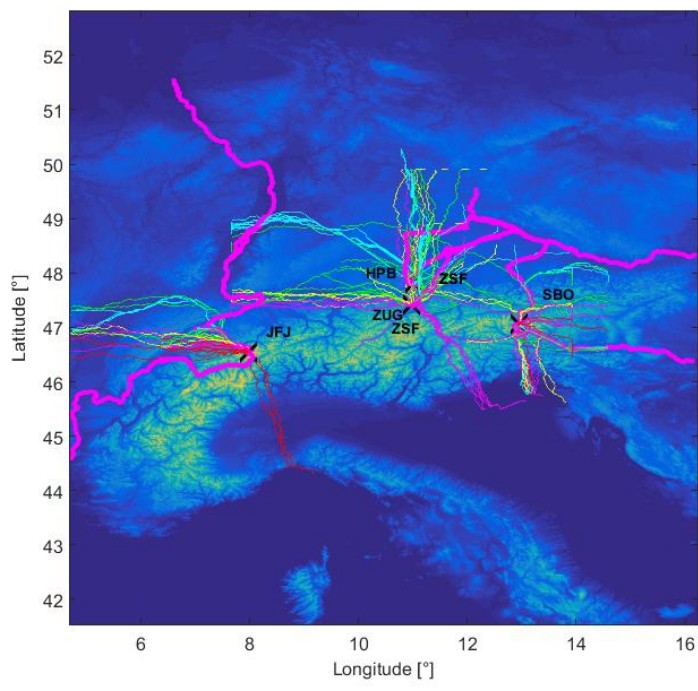

b)

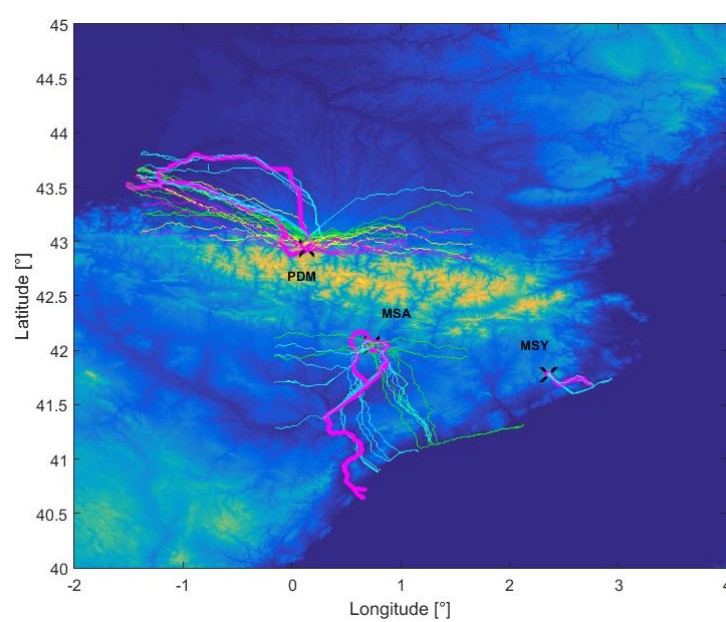





c)

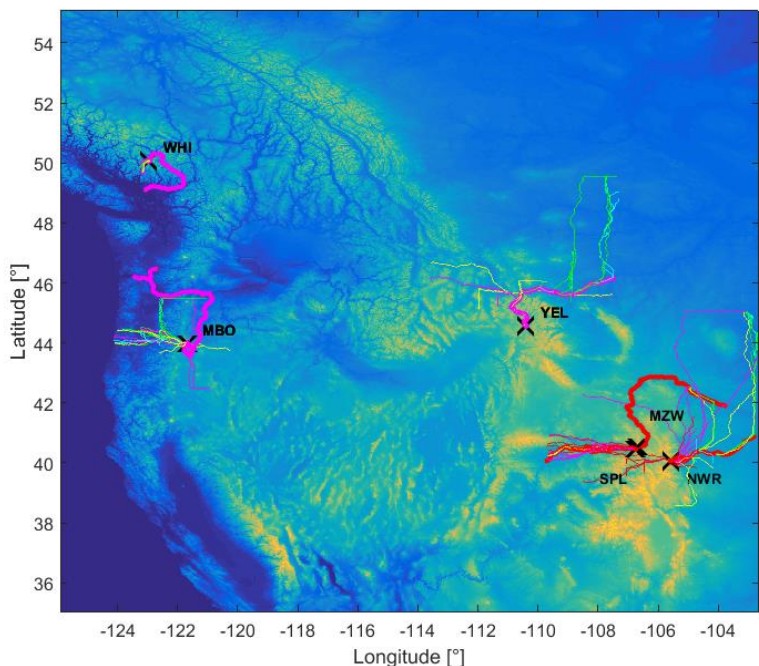

Figure 11: Topography of the various mountainous massifs with the stations studied (black crosses), the main flow paths in thick red and magenta lines for the station grid point and its eight neighbors, respectively for a) the Alps, b) the Pyrenees and c) North-West America. The thin lines correspond to the main flow paths when the top of the ABL is located at varying heights from 200 m a.s.l up to 400 m under the station altitude (cyan: ABL height between 500 and 1000 m a.s.l., light green: ABL height between 1000 and 1500 m a.s.l, yellow: ABL height between 1500 and 2000 m a.s.l., magenta: ABL height between 2000 and 2500 m a.s.l, red: ABL height higher than 2500 m a.s.l.). Calculations corresponding to the various domain sizes can be identified by the various flow paths lengths.



a)                  b)

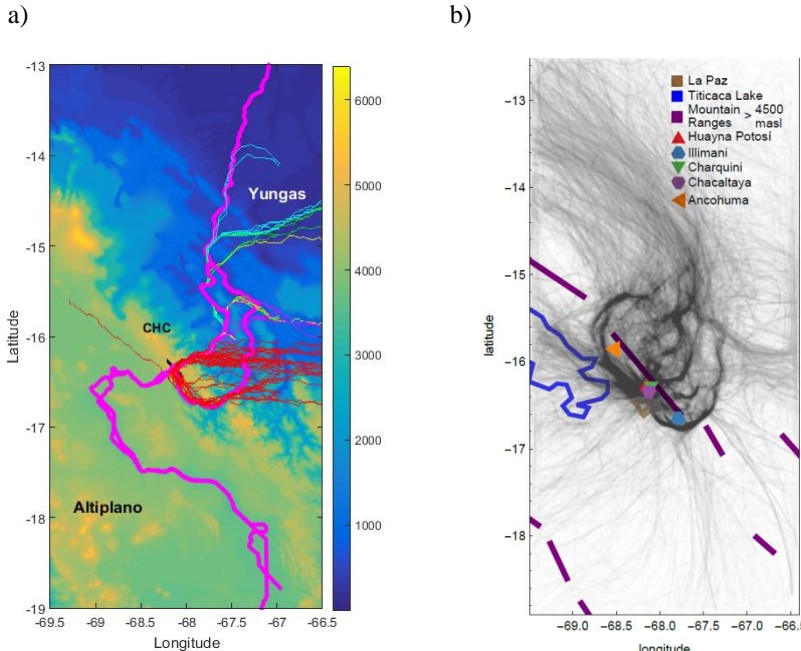

5     Figure 12: a) Topography of CHC region and main flow paths with ABL added to the topography, similar to Fig. 8 and b) Representative 96 hour back-trajectories arriving at CHC during 2012-2014.