# Peer review of "Identification of topographic features influencing aerosol observations at high altitude stations"

_Atmospheric Chemistry and Physics, 2017_

## Referee Comment (RC1) · Anonymous Referee #1 · 23 Oct 2017

This paper presents five metrics that can help quantify the boundary layer impact at high altitude stations. The metrics are based on topographic data and provide information on topographic characteristics including steepness, height difference between station and adjacent valley, and size of the drainage basin. The metrics are calculated for large number of stations. The focus in this paper is on a subset of these stations where aerosol measurements are made.

Overall, I think that the paper is a decent contribution to the scientific literature. The novel part is the quantification of the topographic characteristics surrounding a high altitude stations. The contribution of certain topographic characteristic to trace gas

measurements at these stations is often speculated and discussed and it is nice to see a paper where where an attempt is made to quantify the characteristics. I am not sure though how useful the characterization of the topography as done in the current study will be for future studies/site planning. I found it rather surprising that correlations between topography parameters and the diurnal cycle are weak. Some findings of the paper require more clarification/explanation. I have a few comments that I list below.

1) the choice of the five metrics appears somewhat subjective. At some point in the manuscript (section 4.3) it is stated that "Several other parameters such as the topographical wetness index, the catchment area, the accumulation, dispersion and transit percentages, the hypsometric index and the prominence were tested but were finally eliminated as being not relevant for various reasons." . It remains rather vague why these parameters were eliminated. It would be good if the authors could make a list (e.g, in a table) of all the relevant parameters that the "TopoToolBox" produces and then also clarify what exactly was done to come up with the final five parameters.

2) How are the parameters produced by TopoToolBox similar to or different from the more widely used ArcGis software packages? many people who would like to apply the concept of a topographic index may be familiar with ArcGis sofware packages so a way to make the concept more widely used is to explain how these parameters could be calculated using ArcGis software.

3) page 3, line 30/31: Free convection cannot be driven by forced mechanical convection. This sentence is technically incorrect.

4) section 2.3, line 13: It should be explained here why a domain size of 750x750 km was chosen. The authors discuss somewhat later in the manuscript the sensitivity to domain size but a justification for the chosen domain size should be provided here. The domain size currently sounds rather arbitrary.

5) page 7, line 6: "with the size of the local scale depending on latitude". Please explain/expand.

6) page 8, line 18: use plural "autocorrelations". Also on next line, "auto-correlations" is hyphenated. Find out how it needs to be written and be consistent.

7) page 17, line 6/7:"Usually the spring leads to higher concentration of ABL species than the autumn". Why?

8) page 18, line 14/15: Please explain why absorption coefficient is "the best tracer for anthropogenic pollution and biomass burning and consequently of ABL influence.". Unclear to me.

9) page 19, line 8: "by the smoother pressure decrease". I don't understand that explanation. Please clarify.

10) page 20, line 1: "and at all altitudes" awkward phrase. Rephrase sentence.

11) page 19, line 11: "There are consequently few correlations between topography parameters and the diurnal cycles". This is an important finding that should be explained better in this section. Does this imply that investigators trying to discuss diurnal cycles at high altitude locations waste their time by trying to find any correlation with topography? Please discuss this better.

12) Figure 3 caption, line 5: "horizontal" should be "vertical" here, I think.

13) Figure 11, caption: "Calculations corresponding to the various domain sizes can be identified by the various flow paths lengths.". I don't understand how the calculations can be identified. Please clarify.

14) Figure 12 caption. "similar to Fig. 8". I don't see how this is similar to Fig. 8.

---

## Referee Comment (RC2) · Anonymous Referee #2 · 7 Dec 2017

The manuscript "The topography contribution to the influence of the atmospheric boundary layer at high altitude stations" by Collaud Coen and co-authors investigates the role of the local to regional topography on aerosol observations made at high altitude sites. They derive parameters that are supposed to reflect the average influence of the atmospheric boundary layer on each site and rank the sites by these parameters. A comparison with different observed aerosol parameters is presented and supposed to show the validity and usefulness of the approach. However, I see several major problems with the suggested approach comprising all aspects of the presented work: the methods used to derive topographic parameters, their selection for a final index, and the choice of aerosol parameters that should reflect ABL influence. Although the
manuscript touches on an important question of atmospheric monitoring and could be valuable for future network planning, it cannot be published in the current form and has to undergo major revisions.

Specific concerns

1) The analysis is only focusing on the influence of thermally induced wind systems on the aerosol observations at high altitude stations. Other vertical lifting mechanisms like foehn, deep convection, and frontal passages are completely neglected, although they can be as important depending on location of the site and the season (e.g. tropical vs. high latitude stations, summer vs. winter). The relative contribution by other lifting mechanisms to local "ABL" events will vary strongly between sites (e.g. volcanic island in the subtropics (rare) vs. coastal range mountain in mid-latitude west wind drift (frequent)). The methods presented here need to consider these differences, for example by limiting the observed aerosol observations to cases where vertical lifting mechanisms other than thermally induced flow can be ruled out.

2) Furthermore, the method completely neglects the role of local to regional emissions. Emissions within the region of interest will be very different for the various sites and they will largely determine the amplitude of "ABL" events observed at the sites and also influence the larger scale tropospheric background. At least qualitatively emissions need to be considered and there is no lack of fairly high resolved, global emission inventories (e.g. for BC).

3) A similar problem is the selection of the observed aerosol parameters. Absolute aerosol parameters will depend on more factors than just the local to regional ABL input and are therefore not useful to access the question of FT vs. ABL influenced air mass. It would be more promising to identify pollution or "ABL" events in each data series and correlate the frequency of these with any set of topographic parameters. Why would the 5th percentile of the absorption coefficient be a good indicator of ABL influence? The 5th percentile only reflects the lowest concentrations and not the frequency of

pollution. Looking at the skewness of the distribution could be another indicator. Larger skewness would also indicate more frequent pollution events.

4) The selection and methods to derive the topographic parameters seem to be very arbitrary and no methodological way was followed to present a set of parameters that explains the observed inter-site variability. The final results seem to suggest that mainly one of the parameters is able to predict this variability (hypso%) showing even higher correlation coefficients than the final combined topographic parameter. It also remains unclear why a region as large as 750 km times 750 km was chosen for the analysis. Clearly the flow during one diurnal cycle (and that's what a thermally induced flow system spans) cannot advect air masses from a location as distant as 325 km. Assume an average advection velocity of 5 m/s, which is already a fair value for the kind of fair-weather, low pressure gradient situation required for thermally induced flow, then it would take 18 hours to cover the 325 km. Also plain to mountain winds are known not to extend from the mountains by more than around 100 km. Hence, the use of a smaller region or the use of several sets of parameters for smaller regions should have been considered. These larger sets of topographic parameters and/or any combination of them could than have been fed into a statistical model of the observed aerosol parameters using parameter selection techniques to derive the most important topographic parameters.

5) This continues from 4 but deserves its own point. The analogy between water flowing down a mountain and thermally induced flows rising up a mountain, which is used to derive the parameter DBinv and is used in the discussion of section 3.6, is not valid. It is simply not correct to assume that a large air catchment will result in large upward flow at the highest point of a mountain massif. Air does not flow up to the highest point as water flows down to the lowest point. The upward flow on a fair-weather day with small pressure gradients happens along individual slopes all along individual valleys and results in many convergence lines but not a single convergence point as suggested here. The presented parameter probably has some value on the

very local scale but may just be very similar to hypso% in the end. This parameter and its justification as well as the whole discussion of flow paths will need to be removed from the manuscript. It simply does not reflect the ongoing physics of thermally induced flow systems correctly.

Specific comments

Abstract: Clarify what is the scientific question at hand and what is your contribution to this problem. For example starting from line 21, start the sentence with something like "Here we ..."

Page 8: How comparable are the aerosol parameters between sites? Besides the detection limit adjustment what kind of common quality assurance, quality control was applied to assure that these parameters can really be used for a ranking between sites.

P2,L34: The whole terminology is confusing "flow paths for air convection". Convection does not happen along flow paths. Convection is a vertical transport and mixing mechanism at small scales and as such defined as mostly un-organised. See: http://glossary.ametsoc.org/wiki/Convection. Why not talk about "thermally induced flow paths" instead

P3,L20: Commercial airline programs such as IAGOS CARIBIC (http://www.caribic-atmospheric.com/) would be worth mentioning in this context as well.

P4,L26: Mention that this is the picture for a continental ABL not for a marine ABL.

P4,L29: This is not necessarily correct. In regions with emissions the nighttime accumulation of the emitted species in the shallow SBL usually leads to nighttime concentration maximum of these species.

P4,L17: The authors should mention other vertical lifting processes. Generally frontal lifting (synoptic systems), deep convection and, in mountainous terrain, foehn. The importance of these processes was nicely illustrated by Zellweger et al. (2003).

p4,L25: Zellweger et al. (2002) not in list of references. Probably meant Zellweger et al. 2003, but that does not include a discussion on CO2. Please correct.

p4,L34 to p4,L2: Here it is stated that there are other important influence factors other than thermally induced flow. But it is not explained why one should be able to neglect them. See major remark 1.

p5,L3: The term topographic index or topographic wetness index is already defined in hydrology (the authors used it as well). Therefore, the choice of this name for the parameter introduced here might be confusing, especially since some hydrological methods are applied to derive part of this parameter. Maybe just use ABL-index instead.

p5,L7: Unclear what is mend here by lakes. Again a wrong picture is drawn that suggests that there is a certain amount of air that can be transported by thermally induced flow systems. Lakes or cold air pools are more a phenomenon of the nighttime SBL but not an established concept for daytime flow.

p5,L19: Why was the relatively coarse dataset GTopo30 used? There are global DEMs with higher resolution. 1 km seems a bit coarse for the kind of sites in extremely steep terrain targeted in this study. Some of the local topography will be missed. In this context it would be interesting to see how the height of GTopo30 at the station locations actually compares to the real altitudes. I would encourage the authors to have a look at a higher resolution DEM like https://asterweb.jpl.nasa.gov/gdem.asp for any further analysis.

p6,L11: Very questionable that these parameters are quantitative

p6,L12 cont: Lots of arbitrary choices here. 750 km domain, median altitude vs. station altitude (could be any percentile; lower percentile would avoid negative values), slope between 1 and 10 km, 2-4 km mean gradients ... As mentioned above sets of parameters for different distances, etc. should have been derived and a statistical model with parameter selection been applied. It would also be nice to see all values for the

calculated parameters as part of table 1.

P7,L9: Confusing wording and concept. Drainage is a nighttime process, convection a daytime process???

p7,L25f: It is true that the geometric mean will change in the same way for any percentage change in any of its parameters. However, it does not normalise the variability in the parameters in the desired way. If parameter a has a 10 times larger relative variability than parameter b, the variability of the geometric mean will be dominated by a. If this is an issue in the current case could be easily tested by the authors by analysing the relationship of the original parameters and the derived geometric mean. Better than the geometric mean would be the use of parameters that were normalised for example by their variance.

p8,L17ff: It should be mentioned again when presenting the results that the seasonal and diurnal cycle that is looked at is actually the auto-correlation function. As such the amplitudes of the cycles is already normalised, which helps for the inter-comparability between sites.

p9,L15f: These changes are rather large. Especially considering that the ranking between sites changes with domain size. It should be possible to solve the transformation problem in such a way that G8 and LocSlope are really constant with domain size. Why would the domain size change the local transformation/interpolation anyway? This needs to be redone.

Section 4: The name of the section is misleading. The section does not present a ranking of the sites by TopoIndex but more a discussion along their geographic location.

p12,L5: The more correct name would be "Rocky Mountains".

p12,L15f: Why was MWO not discussed in this context as well?

p13,L13f: Looks like the authors themselves are surprised that there is any relationship between their TopoIndex and the chosen aerosol parameters ...

p13,L26f: But hypso% is an even better predictor than TopoIndex. I guess that means that all other parameters only partly destroy this relationship but do not add any useful information. Especially the suspicious parameter based on water flow analogy, DBinv, seems to show very bad predictive skills (worse than altitude alone in some cases).

p14,L18: Wasn't the point in Bianchi et al that the ABL influence is not a direct one, like you focus on here, but an indirect one of ABL air picked up a few days before arriving at the measurement site and therefore not being lifted by thermally induced flow but by convection or frontal systems.

p14,L30: Isn't the failure of the topoIndex to identify these lower altitude sites a clear indication that the suggested method does not work at all? Otherwise these clear cases of larger ABL influence should be detected and the correlation should actually improve.

p15,L29: All of a sudden back-trajectories appear. It seems clear that these are not the hydrological flow paths. But from which model do these trajectories come from and why were they not used for all sites to also characterise the thermal flow systems (even if not fully represented in the model).

p16,L16-17: This argument is going round in circles. The absorption coefficient is supposed to be an indicator of ABL influence because it correlates with topoIndex. But I though it needs to be shown that the topoIndex actually represents ABL influence ... Very confusing.

p16,L26: NO3 being NO3_aq or ions?

p19,L17f: These parameters are mostly know to the hydrological community but need additional introduction for the more atmospheric readership of the current journal. As mentioned before, it would have been better to provide such parameters to a statistical model with parameter selection in order to get an objective selection of parameters that may explain ABL influence. However, most these parameters would also follow

the misleading assumption that thermally induced flow works just opposite to water flowing downhill and, therefore, should possibly not be considered at all.

Table1: Add the GTopo30 altitude of the grid cell containing each site, along with all the parameters derived for the site (potentially as supplement).

Table2: The units for LocSlope should be m m-1 not Mm-1.

Figure1: The figure quality is not state of the art. I suggest to use a topographic image as background. Larger station labels or symbols. Legend for mountain ranges.

Figure2: The schematic is confusing. If you want to underline that there is a higher ABL influence on the right, why not show a visible, partially terrain following ABL in the mountainous area and an aerosol layer resulting from lift over processes. The schematic on the left is a very poor image of a mountain shape. Looks more like a life buoy with a signal post but not like the profile of a volcano.

Figure4: The thick cyan line is not mentioned in the caption.

Figure6: Sub-panel labels are missing in the figure but are used in the caption.

Figure8: What are the different shades of colours? Neither explained in caption nor text.

Figure9: Very difficult to comprehend. Too many colours and symbols in one plot. Why not display negative correlation coefficients as such on the negative part of the y axis. Instead of circles, different sized symbols should be used for different significance levels.

---

## Author Comment (AC1) · 2 Feb 2018

Dear editor, dear referees,

We would like to thank you for all your comments. This input has allowed us to refine the manuscript by adding more thorough detailed explanations, to correct some minor points and to improve in a large sense the manuscript. This response comprises three sections, first the answers to the main comments of both referees, then the answers to the specific comments of the first and second referees. The authors are conscious that the methodology and topic of this study are to some extent new concepts and that they consequently raise a number of comments. We hope however that in this document we have addressed the referees questions fully and clarified the aspects that needed further elucidation. The co-authors are unanimous that this manuscript presents a valuable methodology for interpreting atmospheric measurements at mountain sites across the globe. This manuscript presents a new technique and extensive data analysis applicable to many of your readers. Based on our extensive efforts in addressing each comment of the reviewers, we ask you to accept for publication the revised version of the manuscript in ACP.

First we want to mention that the values of the ABL-TopoIndex and of the correlation coefficients presented in figures 9 and 10 have changed from those presented in the first version of the manuscript. The differences in the ABL-TopoIndex values are due to the modification of the domain size to 500 km x 500 km. The correlation coefficients changes are due to the modification of the domain size, to the exclusion of SUM due to its outlier status similar to NCOS and to the inclusion of the middle altitude stations (HBP and MSY) in the correlation analysis. Further explanations are given in the following answers to the referee's comments.

1. **Answers to main comments of both referees**

   - **GIS and Topotoolbox**: TopoToolbox is a set of matlab functions that offers analytical GIS utilities in a non-GIS environment. In that sense it is possible to apply GIS-specific methods and to analyse aerosol parameters and cycles in the same environment as the topographic analysis. The TopoToolbox enables the analysis of relief and flow pathways in a DEM as well as the calculation of standard terrain attributes (slope, curvature, flow accumulation,…). The basic functionality of TopoToolbox was therefore used, but further programming was necessary in order to calculate all the necessary parameters constituting the ABL-TopoIndex. As suggested by the referees, the authors added some further clarifications to describe these parameters with sufficient details in the paper, so that the ABL-TopoIndex could be reproduced in any other programming language.

   - **Domain size for the calculation of the ABL-TopoIndex:** The ABL influence at high altitude sites can be divided into a local phenomenon bringing polluted air masses from the adjacent valleys to the measuring station and a broader impact including the whole mountainous massif and a possible influence of nearby plateaus and plains. Poltera et al. (2017) clearly demonstrate that convection above the adjacent valleys rarely influences the high altitude sites but, when it is the case, this local convection does lift air masses with a certain aerosol load. The aerosol layer that comes from a much broader region has a lower aerosol concentration but influences the high altitude stations over a long period of time. An airborne Lidar measurement of the ABL top over the whole alpine massif (Nyeki et al., 2002) clearly stated that the convective boundary layer is formed over a large-scale and leads to an elevated and extended layer. They also quantified that this "large-scale" extends more than 200 km from the mountainous massif. The rectangular domain size of 750 km x 750 km centered on each site corresponds to a distance of at least 375 km in each direction and was initially chosen to ensure the inclusion of the entire massif and a further portion of the adjacent plains. To address the concerns of the reviewer we have restricted the domain to 500 km per side, but think that a domain size smaller than that would no longer correspond to the reality of the aerosol layer formation.

The authors also agree with the second referee that CBL flow will not advect air masses from a distance as large as 375 km. Without precipitation, the residual layer or aerosol layer will however expand over several days. The distances of 375 km and 250 km are covered in 21 and 14 hours, respectively, at an average advection velocity of 5 m/s. The chosen domain size corresponds therefore largely to the development of the CBL and its merging into the residual layer.

- **Methodology and set of quantitative parameters:** The authors are aware that this study consists mostly of a new methodological approach with concepts probably unfamiliar to many atmospheric scientists. The goal was to try to statistically quantify the role of the topography in the ABL influence at high altitude sites. The authors intentionally did not include dynamical parameters such as wind fields that would have required the use of atmospheric models such as ECMWF. While the applied methodology was described with some detail in the original submission we have expanded, and to some extent reorganized the description based on the reviewers comments. First we define a number of topographic criteria that should determine the ABL influence at a high altitude site; second, quantitative parameters are found for each topographic criteria; finally, statistical methods that are valid for environmental studies are applied to the quantitative parameters. Tested qualitative parameters that were finally not selected are briefly described in the supplement to the manuscript. The reasons for not keeping these criteria to calculate the ABL-TopoIndex are now extensively described in the revised manuscript supplement, so that the reader can now better understand the final choice of parameters.
As already mentioned in the paper (section 4.3), the ABL-TopoIndex could probably be improved by adding some further parameters and its validity can also be assessed by other pollutants measurements at high altitude sites.

- **Weak correlations between topography and diurnal and seasonal cycles:** As mentioned by the first referee, the correlations between the topography parameters and the aerosol diurnal cycles are surprisingly weak. This is due to three main reasons: For most of the stations, there are a lot of days where the diurnal cycles are obviously visible. It is however quite difficult to extract the diurnal amplitude as a statistical value due to several factors including: non-regularity in diurnal cycle time of occurrence (e.g., due to different synoptic weather type, cloud presence, advections, long range transport); in the strength of the diurnal cycle (insolation amount, cloud presence); in the absolute level of aerosol present (e.g., due to presence of residual layer, superposition of long range transport); and to the superposition of both seasonal and diurnal cycles. The only possible methodology is to remove the first lag autocorrelation in the data, before extracting the diurnal cycle amplitude from the autocorrelation at 24h (see the supplement to the paper). The removing of the first-lag autocorrelation is a necessary step that introduces noise in the data. Additionally, as explained in the manuscript, only stations partly influenced by the ABL will show a clear diurnal cycle. Stations that remain the whole day in the FT should exhibit no diurnal cycle, whereas stations always in the ABL will have different diurnal cycles due to other periodicity in the sources and to the mixing conditions. As the location of the station with relationship to the ABL can change with season this further complicates the identification of diurnal cycles. Another factor is the presence of the residual layer during the night in summer, which drastically decreases the amplitude of the diurnal cycle. In terms of relating topography and seasonal cycles. Additionally an important thing to consider is that many of the datasets used here are shorter than 5-6 years leading to difficulties in the determination of the seasonal cycles. This is probably a primary cause for the lack of correlation between seasonal cycles and topography parameters. We have revised the manuscript to make this point more clearly as described in our response to referee#1 below.

**2. Answers to referee #1 comments:**

This paper presents five metrics that can help quantify the boundary layer impact at high altitude stations. The metrics are based on topographic data and provide information on topographic characteristics including steepness, height difference between station and adjacent valley, and size of the drainage basin. The metrics are calculated for large number of stations. The focus in this paper is on a subset of these stations where aerosol measurements are made.

Overall, I think that the paper is a decent contribution to the scientific literature. The novel part is the quantification of the topographic characteristics surrounding a high altitude stations. The contribution of certain topographic characteristic to trace gas measurements at these stations is often speculated and discussed and it is nice to see a paper where an attempt is made to quantify the characteristics. I am not sure though how useful the characterization of the topography as done in the current study will be for future studies/site planning.

I found it rather surprising that correlations between topography parameters and the diurnal cycle are weak.

Some further explanation are given in the answers to the main comments on page 2 of this document and the manuscript was revised in order to better explain the reasons for weak correlations. The manuscript was changed at § 3.5 : "*The ABL-TopoIndex is s.s. correlated with the diurnal cycle minimal and maximal strengths of the absorption coefficient. This correlation is once again principally due to the hypso% and G8, and to a lower extent, the LocSlope. The correlation with the diurnal cycle minimal amplitude occurs because the stations that remain in the FT during the whole day should not present any systematic diurnal cycles. The maximal amplitude of the diurnal cycles occurs when the site is in the FT during the night (without any influence of the RL) and influenced by the ABL during the day. The only s.s. correlation with station altitude is found for the scattering coefficient seasonal cycle. Similar to the correlation with the percentiles, there is a high anticorrelation between the particle number concentration diurnal cycles and G8 suggesting that the slope steepness in the vicinity of the stations inhibited both the transport of polluted air masses and NPF. Apart from a correlation at 90% confidence level between DBinv and the absorption coefficient, the lack of further s.s. correlations with the seasonal cycles can be attributed first to the relatively small time period (2-5 years) covered by most of the datasets leading to difficulties in the statistical determination of a yearly periodicity due to inter-annual variability, second to the low aerosol concentration at high altitude sites inducing measurements part of the time near the detection limits of the instruments (see for example the problem with the absorption coefficient at § 2.4) and third to the necessary whitening procedure (see supplement) increasing the dataset noise .*"

And also at § 4.1, the following sentence was added: "*The impact of the RL on the aerosol concentration is probably one of the most important reason to the low correlation between the topographical parameters and the aerosol cycles.*"

1) the choice of the five metrics appears somewhat subjective. At some point in the manuscript (section 4.3) it is stated that "Several other parameters such as the topographical wetness index, the catchment area, the accumulation, dispersion and transit percentages, the hypsometric index and the prominence were tested but were finally eliminated as being not relevant for various reasons." . It remains rather vague why these parameters were eliminated. It would be good if the authors could make a list (e.g, in a table) of all the relevant parameters that the "TopoToolBox" produces and then also clarify what exactly was done to come up with the final five parameters.

- As explained in the answers to the main comments (p. 1 of this document), the tested parameters are only partly provided by the TopoToolbox, some of them were developed or modified for this study. TopoToolbox provides a set of Matlab functions to analyze the relief

and the flow pathways in the digital elevation model, some of them having absolutely no direct relation with the ABL-TopoIndex. In that sense, it is not possible to list all the TopoToobox functions. We have now included more discussion of the reasons to choose the 5 used parameters (§ 2.3) and we have also added Table S2 in the supplement to describe some other topography and hydrology parameters and the motivation of their rejection as relevant parameter to calculate the ABL-TopoIndex: not to use some parameters have to be explicit and the manuscript was accordingly changed:

| Parameter | Definition | Reason for rejection |
|---|---|---|
| Upstream catchment area= flow accumulation | Upstream area contributing to the flow accumulation at the grid cell | 1) It has no direct effect on the ABL influence since it lies at higher altitude than the station
2) It is a partial measurement of the area higher than the station elevation, but only on the mountain side where the station is situated |
| Topographical wetness index = compound topographic index | =ln(A/tan(B)), where A= upstream catchment area and B= slope gradient. It is a measure of the extent of flow accumulation at the given point; it increase as A increases and B decreases. | The wetness index is a ratio of two parameters. The slope gradient is already used (G8) in the ABL-TopoIndex and A was not considered as useful to describe the ABL influence (see previous point). The authors prefer single to combined parameters |
| Drainage basin = dispersive area | Downslope area potentially exposed by flows passing through the given point on the topographic surface | Air convection flow paths cannot be directly assimilated to water flow. The drainage basin in the inverse topography was consequently used as describing the size of the "reservoir" for air convection. |
| Efremov-Krcho landform classification scheme, Dispersion and transit percentages | It is a landform classification scheme (Florinsky, 2012) attributing a characteristic (dissipation, transit or accumulation) to each grid cell. | This classification scheme depends on the curvature of the terrain and, contrary to water flow, it has no relevance for air masses transport. It was however tested on some stations but failed to give a clear characteristic for the station region. |
| Hypsometric curve (HC), hypsometric integral (HI) and | The shape of the HC and HI values provide vital information about erosional stages of the relief and tectonic, climatic and lithological factors controlling landforms development. Convex-up curves are typical for youthful stage and concave- | Both HC and HI characterize the shape of the whole mountainous range and are therefore not defined for the station location. They cannot be used to characterize the station location. |

| | up curves of old stage. (Siddiqui and Soldati, 2014) | |
|---|---|---|
| hypsometric index (HI) | HI= (mean elevation-minimum elevation)/(maximum elevation-minimum elevation) allows different watersheds to be compared regardless on scale. It could reflect both tectonic activity and lithological control. (Siddiqui and Soldati, 2014) | HI also concerns a domain and not the station location. It cannot be used to characterize the station location. |
| Topographic prominence | It is the vertical distance between a summit and the lowest contour line encircling it but containing to higher summits within it. It is a measure of the independence of a summit. | It is not applicable to stations that are not situated at a summit. Moreover, since it restricts the area to a domain without higher summits, it corresponds to domains with very different sizes depending on the station. |

2) How are the parameters produced by TopoToolBox similar to or different from the more widely used ArcGis software packages? many people who would like to apply the concept of a topographic index may be familiar with ArcGis software packages so a way to make the concept more widely used is to explain how these parameters could be calculated using ArcGis software.

- TopoToolbox just offers GIS utilities in a Matlab environment. The parameters used to calculate the ABL-TopoIndex are hopefully clearly enough described to allow any user to find or to write their equivalent in any GIS software packages, including ArcGis. For example, catchments or watersheds are probably calculated in a similar way. Since we have not used ArcGis, it is difficult to estimate exactly the potential of ArcGis compared to TopoToolbox.

3) page 3, line 30/31: Free convection cannot be driven by forced mechanical convection. This sentence is technically incorrect.
- The referee is correct that free convection cannot be due to any forced mechanism. The sentence was modified: "*In the case of cloudy or rainy conditions as well as in the case of advective weather situations, free convection is no longer driven primarily by solar heating, but by ground thermal inertia, cold air advection and/or cloud top radiative cooling.*"

4) section 2.3, line 13: It should be explained here why a domain size of 750x750 km was chosen. The authors discuss somewhat later in the manuscript the sensitivity to domain size but a justification for the chosen domain size should be provided here. The domain size currently sounds rather arbitrary.

- The answer to this question is given in the answers to the main comments (p. 1 of this document). A large domain size has to be chosen in order to take into account the whole mountainous range and part of the adjacent plains/plateau contributing to the formation of the aerosol layer. In that sense a domain of 500x500 km$^2$ could also be justified and was consequently used in the revised version of the manuscript, leading to small variation of the ABL-TopoIndex for some stations. We have also clarified our reason for the size of domain in the revised manuscript at § 2.3: "*A quantitative estimation of these criteria depends clearly on the domain considered. The minimal size requirement for such a topographical analysis is that the domain should contain the whole mountainous massif. An airborne Lidar measurement of the ABL over the Alps (Nyeki et al., 2002) clearly stated that the convective boundary layer is formed over a large-scale and leads to an elevated and extended layer. It also quantifies this "large-scale" to extend more than 200 km from the mountainous massif. A*

*rectangular domain size of 500 km x 500 km centered on each site was then chosen (see § 3.2 for a discussion of the effect of the domain size)."*

5) page 7, line 6: "with the size of the local scale depending on latitude". Please explain/expand.

- The gradient is applied between 2 grid cells and the length of the domain covered by 2 grid cells depends on latitude (see § 2.2) and correspond to 2-4 km. The manuscript was changed*:" This parameter takes into account the slopes towards lower and higher elevations over a local scale (2-4 km that is the distance covered by two grid cells, with the size of the grid depending on latitude)"*

6) page 8, line 18: use plural "autocorrelations". Also on next line, "auto-correlations" is hyphenated. Find out how it needs to be written and be consistent.

- OK, the text was changed and the hyphenation was removed. The supplement was also corrected.

7) page 17, line 6/7:"Usually the spring leads to higher concentration of ABL species than the autumn". Why?

- At most sites (and not only at high elevated sites) the CBL height is found to be higher in spring and summer than in autumn and winter. The correlation with the down welling solar radiation at the surface clearly explains the summer high ABL height and hence the summertime peaks. Some other authors found an anti-correlation with the surface pressure and the lower tropospheric stability and a correlation with the near surface wind speed and temperature. A cumulative effect of all these parameters leads to a usually higher CBL height in spring (Guo et al., 2016, Pal and Haeffelin, 2015). However, since I do not find a clear referenced explanation for the often observed difference in ABL height between spring and autumn, I prefer not to insert any further explanation in the manuscript. The sentence was however changed to: "*Usually the spring leads to higher aerosol species than the autumn probably bounded to higher ABL height."*

8) page 18, line 14/15: Please explain why absorption coefficient is "the best tracer for anthropogenic pollution and biomass burning and consequently of ABL influence.". Unclear to me.

- The GAW-recommended basic aerosol measurement program consists of the particle number concentration, the scattering and absorption coefficients. All three of these parameters are higher in ABL than in FT. As stipulated on page 18 (in originally submitted manuscript), the aerosol absorption coefficient (or black carbon (BC) concentration) is the best tracer for ABL influence among the three aerosol parameters discussed. This is because the main sources of BC (anthropogenic pollution due to combustion processes and biomass burning) are in the ABL but are scarce near the high altitude sites. Additionally, BC aerosol is not produced by any secondary processes. In contrast, the particle number concentration and, to a lesser extent, the scattering coefficient are also influenced by gas-to-particle conversion mechanisms such new particle formation and condensational growth, which are secondary processes depending on the ABL influence in a more complex way and also on other parameters such as the solar insolation, the temperature and other thermodynamic processes. In that sense and among the basic aerosol parameters measured at most stations, the absorption coefficient is the best tracer for ABL influence. § 4.2 was changed accordingly: "*The absorption coefficient is primarily due to the presence of black carbon emitted from combustion processes occurring mostly in the ABL and rarely near the high altitude stations; additionally, BC aerosol is not produced by any secondary processes. Among the aerosol parameters studied here, the absorption coefficient is consequently the best tracer for anthropogenic pollution and biomass burning and consequently of ABL influence."*

9) page 19, line 8: "by the smoother pressure decrease". I don't understand that explanation. Please clarify.

- An anti-correlation between the slopes around the station and the number concentration can be explained by new particle formation that is enhanced if the pressure difference experienced during the upslope transport is not too large. The sentence was clarified: "*The greater correlation of slope with the number concentration rather than with the absorption coefficient can be explained both by the very scarce sources of black carbon in the near vicinity of most of the high altitude stations and by the smoother pressure decrease experienced by the precursors during their upslope transport along gentle slopes leading to more condensation processes and nucleation.*"

10) page 20, line 1: "and at all altitudes" awkward phrase. Rephrase sentence.

- *This was just a mistake and was removed.*

11) page 19, line 11: "There are consequently few correlations between topography parameters and the diurnal cycles". This is an important finding that should be explained better in this section. Does this imply that investigators trying to discuss diurnal cycles at high altitude locations waste their time by trying to find any correlation with topography?
Please discuss this better.

See also the answers to the main comment on page 2 and 3 of this document. The study of the diurnal cycles at high altitude sites can really bring important results if specific cases are analyzed and compared. In this study, a statistical approach has to be used to obtain a reliable estimate of the diurnal cycle amplitude and leads consequently to weaker correlations. This paragraph was changed: "*The aerosol diurnal cycles are influenced by numerous phenomena (see Sect. 4.1) leading to a non-trivial relationship with the ABL influence. If the study of the diurnal cycles can bring valuable results if specific cases are analyzed and compared, the statistical approach is less obvious due to the noise in the data (low aerosol concentration and whitening process), to the inter-annual variability of the meteorological processes and to cloud, precipitation and long-range advection involving a large day to day variability. There are consequently few statistical correlations between topography parameters and the diurnal cycles. The clearest correlation is the influence of the insolation on the aerosol diurnal cycles amplitudes. This dependence between the latitude and the aerosol concentration was already mentioned by Kleissl et al. (Kleissl et al., 2007) and is easily understandable, the convection and the new particle formation being directly dependent on the solar radiation intensity. The other correlations are found between some topography parameters (ABL-TopoIndex, hypso%, G8 and LocSlope) and the absorption coefficient, which is the best tracer for ABL influence among the aerosol parameters.*"

12) Figure 3 caption, line 5: "horizontal" should be "vertical" here, I think.

- Yes, it was changed

13) Figure 11, caption: "Calculations corresponding to the various domain sizes can be identified by the various flow paths lengths.". I don't understand how the calculations can be identified. Please clarify.

- The plotted colored lines have various lengths that are for example visible around SBO station. This section was deleted following suggestions by the second referee.

14) Figure 12 caption. "similar to Fig. 8". I don't see how this is similar to Fig. 8.

-   You're right, it should be changed to Fig. 11. This section was deleted following suggestions by the second referee.

**3. Answers to referee #2 comments:**

The manuscript "The topography contribution to the influence of the atmospheric boundary layer at high altitude stations" by Collaud Coen and co-authors investigates the role of the local to regional topography on aerosol observations made at high altitude sites. They derive parameters that are supposed to reflect the average influence of the atmospheric boundary layer on each site and rank the sites by these parameters. A comparison with different observed aerosol parameters is presented and supposed to show the validity and usefulness of the approach. However, I see several major problems with the suggested approach comprising all aspects of the presented work: the methods used to derive topographic parameters, their selection for a final index, and the choice of aerosol parameters that should reflect ABL influence. Although the manuscript touches on an important question of atmospheric monitoring and could be valuable for future network planning, it cannot be published in the current form and has to undergo major revisions.

**Specific concerns**
1) The analysis is only focusing on the influence of thermally induced wind systems on the aerosol observations at high altitude stations. Other vertical lifting mechanisms like foehn, deep convection, and frontal passages are completely neglected, although they can be as important depending on location of the site and the season (e.g. tropical vs. high latitude stations, summer vs. winter). The relative contribution by other lifting mechanisms to local "ABL" events will vary strongly between sites (e.g. volcanic island in the subtropics (rare) vs. coastal range mountain in mid-latitude west wind drift (frequent)). The methods presented here need to consider these differences, for example by limiting the observed aerosol observations to cases where vertical lifting mechanisms other than thermally induced flow can be ruled out.

-   First, the authors agree with the referee that convection and thermally induced wind systems are not the only mechanisms that bring polluted air masses to high altitudes. The other vertical lifting mechanisms described by the referee contribute to indeed enhance the pollutant concentrations at high altitudes up to the free troposphere and it is also correct that these effects will vary depending on site, season, latitude, etc. However, as we explained in the introduction of the manuscript, we restricted this study solely to the influence of the topography on the thermally induced wind systems and the CBL growth. This study considers neither the dynamics of the atmosphere nor the soil properties. Such detailed and specific analysis is best left to the scientists responsible for the individual stations but is too complex when evaluating multiple sites with disparate data sets. To take into account the atmosphere dynamics, 3D models (and not only a 2D model of the earth surface) are necessary, which is clearly not the goal of this study and definitely outside its scope. Due to computational constraints, most current global models doesn't do a good job of representing the actual topography, the model grid spacing tends to be too large (on order of 1-2 degrees of latitude and longitude) most global models provide low frequency output – typically monthly (although sometime daily). This means that targeted regional models would need to be used to describe each of the 46 sites here, again – this is a topic best left to the local experts responsible for each observatory.
-   Second, our approach is to do a global and statistical analysis to understand the role of the topography in the ABL influence across an array of 46 mountain sites. Our hope was to begin to develop common rules that can be applied to all stations. It was never meant to analyze specific cases for clear thermally driven transport at individual stations. Doing so would also greatly reduce the usable time series and result in statistically small data sets. As stated in the

introduction, there is presently no single method to screen ABL-influenced from FT air masses at high altitude sites. It is therefore quite difficult to sort the cases where the ABL influence is only due to thermally driven transport. Even if possible for all types of environment, the limitation of the aerosol dataset to cases where vertical lifting mechanisms other than thermally induced flow can be ruled out would need further complex data sets for each station (for example: pressure, humidity, wind measurements at each side of the stations, 3D back-trajectories, synoptic classification scheme and probably some gaseous species concentrations).

- Finally, Zellweger et al. (2003) concluded that, in contrast to the NOy mixing ratio, the major process for upward transport of aerosol is the thermally induced vertical transport. The choice of aerosol parameters to validate the ABL-TopoIndex can therefore be considered as the best one to study the thermally driven air mass transport.

- For all these reasons, the authors consider that the inclusion of the atmosphere dynamics and of the wind systems is beyond the scope of this study.

2) Furthermore, the method completely neglects the role of local to regional emissions. Emissions within the region of interest will be very different for the various sites and they will largely determine the amplitude of "ABL" events observed at the sites and also influence the larger scale tropospheric background. At least qualitatively emissions need to be considered and there is no lack of fairly high resolved, global emission inventories (e.g. for BC).

- The authors agree that the regional emission sources have an influence on the pollutant concentration measured at high altitude sites. However, the timing and relative magnitude of temporal cycles (as determined by auto-correlation) with ABL influence does not depend on the pollutant concentration in the ABL. To use the emission inventories, the atmosphere dynamic and particularly the wind components should also be taken into account in order to assess which sources on the 500 km x 500 km influence the high altitude sites (see answer to previous referee comment). Moreover, while the absorption coefficient could perhaps be "normalized" by the BC emission inventories, the scattering coefficient and the number concentration depend also on gas to particle conversion (e.g., new particle formation and condensation). The modeling of the gas-to-particle conversion from the emissions inventories and meteorological data is however rather complex. Moreover, the highest aerosol concentrations at the high altitude sites often depend much more on long range transport of mineral dust or biomass burning than on the regional sources. The authors are therefore of the opinion that, first, these large uncertainties would annihilate the potential benefits of the inclusion of the emission inventories and, second, that the inclusion of the atmosphere dynamics is beyond the scope of the paper. Additionally the amplitude of the diurnal cycle which is discussed in section 3.5 should be independent of the regional sources; this is therefore another way to "normalize" the aerosol concentration without reference to emissions information.

3) A similar problem is the selection of the observed aerosol parameters. Absolute aerosol parameters will depend on more factors than just the local to regional ABL input and are therefore not useful to access the question of FT vs. ABL influenced air mass. It would be more promising to identify pollution or "ABL" events in each data series and correlate the frequency of these with any set of topographic parameters. Why would the 5th percentile of the absorption coefficient be a good indicator of ABL influence? The 5th percentile only reflects the lowest concentrations and not the frequency of pollution. Looking at the skewness of the distribution could be another indicator. Larger skewness would also indicate more frequent pollution events.

- To our knowledge the only method to detect local CBL development as well as the top of the aerosol layer is to use a ceilometer (or a lidar). There are however very few high altitude station around the world with a ceilometer time series from a lower altitude adjacent station thus limiting any statistical analysis.

- The 5th percentile clearly reflects the lowest concentrations and therefore the ability to sample clean FT air masses at the high altitude stations. The lower the ABL influence is (through the CBL and the aerosol layer heights), the lower the 5th percentile will be. The authors do agree that the median of the aerosol parameters is much more dependent on regional and local sources, whereas the 95% depends on rare high aerosol concentrations probably due to long-range transport of mineral dust or biomass burning. A normalization of the aerosol parameter with the 95% has consequently also not much sense.

- The aerosol parameters discussed in the manuscript (number concentration, absorption and scattering coefficient) are approximately lognormally distributed variables. The skewness toward the lower values is therefore not defined. The skewness toward the higher values reflects the occurrences of very high aerosol concentration that generally relates to long range transport of mineral dust and biomass burning. The skewness is consequently not the right parameter to detect ABL-influence.

- Apart from the 5th percentile, the best parameters are clearly the diurnal and seasonal cycles. These are however much more difficult to statistically extract from the time series (see the answers to the main comments on p. 1 in this document) and exhibit few correlations with the topography parameters.

4) The selection and methods to derive the topographic parameters seem to be very arbitrary and no methodological way was followed to present a set of parameters that explains the observed inter-site variability. The final results seem to suggest that mainly one of the parameters is able to predict this variability (hypso%) showing even higher correlation coefficients than the final combined topographic parameter. It also remains unclear why a region as large as 750 km times 750 km was chosen for the analysis. Clearly the flow during one diurnal cycle (and that's what a thermally induced flow system spans) cannot advect air masses from a location as distant as 325 km. Assume an average advection velocity of 5 m/s, which is already a fair value for the kind of fair-weather, low pressure gradient situation required for thermally induced flow, then it would take 18 hours to cover the 325 km. Also plain to mountain winds are known not to extend from the mountains by more than around 100 km. Hence, the use of a smaller region or the use of several sets of parameters for smaller regions should have been considered. These larger sets of topographic parameters and/or any combination of them could than have been fed into a statistical model of the observed aerosol parameters using parameter selection techniques to derive the most important topographic parameters.

- The methodology applied in this study consists first in identifying topographical criteria that would tend to increase the ABL influence and then finding parameters that can be quantitatively estimated and related to the topographic criteria. The authors do agree that some choices were not sufficiently motivated in the first version of the manuscript so that now both the used parameters (section 2.3: ABL-TopoIndex) and the rejected parameters (Table S2) are now better described. (see also the answer to the specific comment "p.6 L12" in this document)

- The reasons to choose a large size of the domain are given in p. 1 of this document (answers to the main comments) and also now discussed in the revised paper. The authors also have now restricted the size of the domain to 500km x 500km. This restriction has a very low impact on the results. : "*A quantitative estimation of these criteria depends clearly on the domain considered. The minimal size requirement for such a topographical analysis is that the domain should contain the whole mountainous massif. An airborne Lidar measurement of the ABL over the Alps (Nyeki et al., 2002) clearly stated that the convective boundary layer is formed over a large-scale and leads to an elevated and extended layer. It also quantifies this "large-scale" to extend more than 200 km from the mountainous massif. A rectangular domain size of 500 km x 500 km centered on each site was then chosen (see § 3.2 for a discussion of the effect of the domain size).*"

- The third specific concern of the referee clearly supports our contention that there are no parameters that can act as an indubitable sign of ABL influence. The best statistical parameter would be the annual cycle of the diurnal cycle amplitude which should be the greatest for

stations sampling the FT part of the year. It was however not statistically possible to extract this parameter from the available time series for the following two reasons: 1) a lot of the time series were too short ($< 2$-5 years), as explained in the answers to the main comments (p. 2 of this document), and 2) the low aerosol concentration measured at high altitude combined with the pre-whitening process lead to a large uncertainty in the statistical determination of the cycle amplitude. In that sense, there is, to our knowledge, no reference measurement that would definitively identify the ABL influence and allow selection through a statistical model the most important topographic parameters.
- Apart from the used and the rejected parameters (now more clearly described in the revised manuscript), the authors do not see any other "direct" parameters that can be possibly used. There are other more sophisticated parameters that are linked to the valley's topography that could be added to the ABL-TopoIndex in a further study (see § 4.3 describing possible future work), but the authors found it necessary to validate the present study by a publication before investing further time in exploring more complicated parameters.

5) This continues from 4 but deserves its own point. The analogy between water flowing down a mountain and thermally induced flows rising up a mountain, which is used to derive the parameter DBinv and is used in the discussion of section 3.6, is not valid. It is simply not correct to assume that a large air catchment will result in large upward flow at the highest point of a mountain massif. Air does not flow up to the highest point as water flows down to the lowest point. The upward flow on a fairweather day with small pressure gradients happens along individual slopes all along individual valleys and results in many convergence lines but not a single convergence point as suggested here. The presented parameter probably has some value on the very local scale but may just be very similar to hypso% in the end. This parameter and its justification as well as the whole discussion of flow paths will need to be removed from the manuscript. It simply does not reflect the ongoing physics of thermally induced flow systems correctly.

- The authors do agree with the referee that the analogy between water flowing down and thermally driven air flow has very well defined limitations. In that sense we have removed the whole discussion about flow paths (§ 3.6 and figures 11 and 12) that involve a direct analogy between the water and the air mass flow paths.
- The DBinv used in the ABL-TopoIndex has a completely different motivation and impact. DBinv is a quantitative parameter for the size of the reservoir for air convection (criterion number 4). The authors do agree that upward flows do not result in a single convergence point at the station. However DBinv is a measure of the territory that can directly influence the station air masses by upslope winds. It is true that the considered domain represented by DBinv is too large to represent the direct influence of the CBL at the station, but it is of reasonable size to describe the influence of the aerosol layer (AL) (or residual layer (RL) during the night). It was clearly shown that the AL (or RL) have a clear impact on the aerosol concentration at high altitude stations (Collaud Coen et al., 2011, Poltera et al., 2017, Andrews et al., 2011 and references therein). Due to these reasons and to the influence of DBinv on the correlation of the ABL-TopoIndex with the aerosol parameters, the authors have chosen to keep DBinv in the ABL-TopoIndex definition.

**Specific comments**
Abstract: Clarify what is the scientific question at hand and what is your contribution to this problem. For example starting from line 21, start the sentence with something like "Here we ..."

- The abstract was modified and the following sentence was added at line 21: "*In this study, a topography analysis is performed allowing calculation of a newly defined index called ABL-TopoIndex. The ABL-TopoIndex is constructed in order to correlate with the ABL influence at the high altitude stations and long-term aerosol time series are used to assess its validity.*"

Page 8: How comparable are the aerosol parameters between sites? Besides the detection limit adjustment what kind of common quality assurance, quality control was applied to assure that these parameters can really be used for a ranking between sites.

- 23 of the 28 aerosol datasets are provided by GAW stations and the data were obtained from the EBAS data center. GAW stations have to follow the measuring rules and quality assessment edited by the WMO/GAW aerosol advisory board. These measurement principles are extensively described in GAW report Nr 200 (WMO/GAW standard operating procedures for in-situ measurements of aerosol number concentration, light scattering and light absorption). As required by the GAW aerosol advisory board, all measurements were performed at low humidity (RH<40%). Moreover the data owners also follow the quality control procedures of the EBAS data center. Four of the datasets (MUK, NWR, PEV and OMP) are not GAW stations but the measurements were performed by research groups operating at other GAW stations. Individual exchanges with the data providers from those four sites indicated that they collected those datasets using methods similar to their operations at GAW stations so that the quality and traceability are assured. The GAW stations are now given in bold in Table S3. The umbrella provided by the WMO/GAW program is, to our point of view, sufficient so that a further description of the quality assurance of the aerosol measurements is not needed in this paper.
- Other procedures such as the STP correction, the truncation correction of Nephelometer data, the negative data of the absorption coefficient were controlled and handled similarly for all datasets. Small time series breakpoints are not important since no trends were calculated.
- All the time series were visually inspected and any doubtful data were removed after discussions with data providers.
- All the times series but 2 were done on TSP or PM10 inlets, so that similar aerosol size distributions were measured.

P2,L34: The whole terminology is confusing "flow paths for air convection". Convection does not happen along flow paths. Convection is a vertical transport and mixing mechanism at small scales and as such defined as mostly unorganised. See: http://glossary.ametsoc.org/wiki/Convection. Why not talk about "thermally induced flow paths" instead.

- The authors agree that "thermally induced flow paths" is a much better terminology. Since § 3.6 on "Flow paths as a function of ABL heights" was removed, the expression "flow paths for air convection" no longer appears in the manuscript.

P3,L20: Commercial airline programs such as IAGOS CARIBIC (http://www.caribicatmospheric. com/) would be worth mentioning in this context as well.

- The following text was added to the manuscript: "*Instrumented airplanes can make detailed measurements of the vertical and spatial distribution of atmospheric constituents and are used either during limited measurement campaigns or on regular civil aircraft (see for example the IAGOS CARIBIC project), but, because of the limited temporal scope of most measurement campaigns, cannot provide long-term, continuous context for the measurements*."

P4,L26: Mention that this is the picture for a continental ABL not for a marine ABL.

- Ok , done (@P3 L26): ". *In the case of fair-weather days, the continental ABL has a well-defined structure and diurnal cycle leading to the development of a Convective Boundary Layer (CBL), also called a mixing or mixed layer, during the day and a Stable Boundary Layer (SBL) which is capped by a Residual Layer (RL) during the night (Stull, 1988).*"

P4,L29: This is not necessarily correct. In regions with emissions the nighttime accumulation of the emitted species in the shallow SBL usually leads to nighttime concentration maximum of these species.

- As noted by the referee, it is completely correct that the emitted species accumulate during nighttime in the SBL, leading in some cases to high concentrations. This was now specified in the text at P3 L30.
*"During daytime, the aerosol concentration is maximum in the CBL and remains high in the RL. During nighttime, the surface-emitted species accumulate in the SBL."*

P4,L17: The authors should mention other vertical lifting processes. Generally frontal lifting (synoptic systems), deep convection and, in mountainous terrain, foehn. The importance of these processes was nicely illustrated by Zellweger et al. (2003).

- The text was changed to *"Finally, ABL air masses can also be dynamically lifted by frontal systems, deep convections or foehn as well as be advected from mesoscale or wider regions and influence high altitude measurements by all these atmospheric processes."*

p4,L25: Zellweger et al. (2002) not in list of references. Probably meant Zellweger et al. 2003, but that does not include a discussion on CO2. Please correct.

- This was indeed incorrect and was changed: *"Many methods have been used to separate FT from ABL influenced measurements, including those based on time of day and time of year approach (Baltensperger et al., 1997; Gallagher et al., 2011), wind sectors (Bodhaine et al., 1980), the vertical component of the wind (García et al., 2014), wind variability (Rose et al., 2016), NOx/NOy, NOy/CO ratios or radon concentrations (Griffiths et al., 2014; Herrmann et al., 2015a, 2015b; Zellweger et al., 2003) and water vapor concentrations (Ambrose et al., 2011; Obrist et al., 2008), although none of these methods leads to an absolute screening procedure to ensure the measurement of pure FT atmosphere."*

p4,L34 to p4,L2: Here it is stated that there are other important influence factors other than thermally induced flow. But it is not explained why one should be able to neglect them. See major remark 1.

- Please see our response to main comments p.1 of this document. Further, to our knowledge, most of the meteorological models are not able to solve all the dynamic processes in complex topography. This study therefore concentrates on one question and tries to identify some relations between the topography and the thermally induced ABL influence. This restricted, but nevertheless ambitious objective (as well as the factors that are not taken into account), are clearly specified in the manuscript.

p5,L3: The term topographic index or topographic wetness index is already defined in hydrology (the authors used it as well). Therefore, the choice of this name for the parameter introduced here might be confusing, especially since some hydrological methods are applied to derive part of this parameter. Maybe just use ABL-index instead.

- P6L3: The referee is, of course, right in saying that the terminology of "topography" and "index" are already widely used in several scientific domains. The use of only "ABL-Index" however seems too vague to the authors, since it does not specify that only the effects of the topography are taken into account. For example, an ABL-Index could represent any number of ways of assessing ABL influence. The authors chose therefore a name (ABL-TopoIndex) where the three main underlying concepts explored in the manuscript are cited. The word "topography" was also abbreviated in order to minimize possible confusion with existing hydrological terms. Moreover the manuscript was carefully checked so that the word "TopoIndex" was never used alone. The authors prefer to keep it as written.

p5,L7: Unclear what is mend here by lakes. Again a wrong picture is drawn that suggests that there is a certain amount of air that can be transported by thermally induced flow systems. Lakes or cold air pools are more a phenomenon of the nighttime SBL but not an established concept for daytime flow.

- The authors do agree that, even if used in quotation marks, the word "lakes" is misleading. It is now replaced by the expression "*air mass reservoirs*" in the revised version.

p5,L19: Why was the relatively coarse dataset GTopo30 used? There are global DEMs with higher resolution. 1 km seems a bit coarse for the kind of sites in extremely steep terrain targeted in this study. Some of the local topography will be missed. In this context it would be interesting to see how the height of GTopo30 at the station locations actually compares to the real altitudes. I would encourage the authors to have a look at a higher resolution DEM like https://asterweb.jpl.nasa.gov/gdem.asp for any further analysis.

- The authors thank the reviewer for giving the suggestions of another high resolution DEM that they will use in case of further studies. A higher resolution model will clearly be of interest. It has however to be noted that the ABL influence is not really a very local phenomena so that the mean over 9 grid cells was used to obtain the ABL-TopoIndex. As expected the GTopo30 altitude at the station grid cell differs by more than 20% for 3 of 28 stations used for the correlation analysis, the GTopo30 altitude being always lower than the station altitude. These differences do not correspond to the real altitude difference between the real and the GTopo30 mean altitude over each grid cell. Corresponding to the methodology applied to the ABL-TopoIndex, the correlations were also done with the mean altitude of the 9 grid cells. It has however to be noted that the use of the station altitude or of the 9 grid cells mean altitude does not change the correlation results. The GTopo30 manual gives a minimal vertical accuracy of 250 m at 90% confidence level and a RMSE of 152 m (the Peru map, which has a lower accuracy (see GTOPO30 manual), is not used). The altitude of the grid cell containing the station as well as the mean altitude of the 9 grid cells used to calculate the ABL-TopoIndex are now given for all stations in Table S1 with the following comments: "*The real altitude of the station, the mean altitude of the grid cell containing the station and the mean altitude of the grid cell containing the station and of its 8 adjacent grid cells are given in Table S1, the last 2 altitudes are calculated from the DEM after its projection in UTM coordinates. Since the stations are usually at high altitude, the altitude of the DEM grid cell is usually lower than the station altitude. The mean and median of the differences between the station altitude and the one of the grid cell are 190 m (8.6%) and 140 m (5.8%), whereas the mean and median of the differences between the station altitude and the one of the 9 grid cells are 270 m (11.7%) and 220 m (10.3%), respectively. The maximal altitude differences is found for SZZ (1153 m) that corresponds to 3% of the station altitude. Due to its peculiar situation (see paper), NCOS altitude is 1110 m lower that its DEM grid cell altitude (2.8%) and this can perhaps explain NCOS outlier status. ZEP is only 306 higher than its grid cell altitude, but this corresponds to 65% of its altitude and also explain its very high ABL-TopoIndex and its outlier status. It has however to be noted that The GTopo30 manual gives a minimal vertical accuracy of 250 m at 90% confidence level and a RMSE of 152 m (the Peru map being anyhow not used).*"

p6,L11: Very questionable that these parameters are quantitative

- The parameters described under 2.3 are quantitative parameters that can be calculated for each point of the earth using a DEM. In that sense we think that the adjective "quantitative" is not misleading.

p6,L12 cont: Lots of arbitrary choices here. 750 km domain, median altitude vs. station altitude (could be any percentile; lower percentile would avoid negative values), slope between 1 and 10 km, 2-4 km mean gradients ... As mentioned above sets of parameters for different distances, etc. should have been derived and a statistical model with parameter selection been applied. It would also be nice to see all values for the calculated parameters as part of table 1.

- We agree that all the choices should be explained in this section and rather than later on in the paper:
1) Concerning the size of the domain, please see the answers to the main comments on p.1 of this document. § 2.3 was also modified: "*A quantitative estimation of these criteria depends*

*clearly on the domain considered. The minimal size requirement for such a topographical analysis is that the domain should contain the whole mountainous massif. An airborne Lidar measurement of the ABL over the Alps (Nyeki et al., 2002) clearly stated that the convective boundary layer is formed over a large-scale and leads to an elevated and extended layer. It also quantifies this "large-scale" to extend more than 200 km from the mountainous massif. A domain size of 500 km x 500 km centered on each site was then chosen (see § 3.2 for a discussion of the effect of the domain size).",*

2) for the hypsD50, the referee is correct that any percentiles could be chosen. The median was taken first because it is a common averaging tool and second because presumably it would be lower than the location of each "high altitude station". The authors tried to summarize this more clearly in the manuscript by adding the following sentence: *"The median of the hypsometric curve was chosen first because a station claiming to be a high altitude site should typically be at higher altitude than half of its geographical environment."* Moreover, the station with hypsD50 can be found in Table S1.

3) LocSlope is defined on a radius of 10 km since the minimal distance between the station and the nearest plateau is usually equal to or larger than 10 km. This is now stated in the manuscript: *"The distance of 10 km to calculate the LocSlope was then chosen as representative of the maximal distance to the next adjacent plateau for almost all stations."*

4) the G8 is always calculated from one grid cell to the next, so that the distance of 2-4 km is given by GTopo30 and varies with latitude.

Moreover all values for the calculated parameters are now in Table S4

P7,L9: Confusing wording and concept. Drainage is a nighttime process, convection a daytime process???

- Yes, drainage winds are a nighttime process, but the manuscript discusses a "drainage basin". Drainage basin is a hydrologic term without time connotation and can be used for daytime processes. As defined by the dictionary, "a drainage basin is the area drained by a river and all its tributaries". It is also called catchment area, drainage area, watershed or river basin.

p7,L25f: It is true that the geometric mean will change in the same way for any percentage change in any of its parameters. However, it does not normalise the variability in the parameters in the desired way. If parameter a has a 10 times larger relative variability than parameter b, the variability of the geometric mean will be dominated by a. If this is an issue in the current case could be easily tested by the authors by analysing the relationship of the original parameters and the derived geometric mean. Better than the geometric mean would be the use of parameters that were normalized for example by their variance.

- The referee is correct that the geometric mean reports similarly any percentage change in any included parameters whatever the absolute value of the parameter is. This is the reason to apply the geometric mean for environmental indices that are built with very different parameters. The use of other types of averaging with any kind of normalization does not allow us to obtain this necessary (for this analysis) mathematical property. A normalization with either the maximum or with the variance will change the value of the ABL-TopoIndex but not the ranking of the stations. Moreover the authors checked that none of the included parameters dominates the results. To further develop this critical technical point, the manuscript was changed: *"Further, a given percentage change in any of the parameters will yield an identical change in the calculated geometric mean value. In that sense the variability of each parameter is also normalized, leading to similar modifications of the ABL-TopoIndex for similar parameter's variations."*

p8,L17ff: It should be mentioned again when presenting the results that the seasonal and diurnal cycle that is looked at is actually the auto-correlation function. As such the amplitudes of the cycles is already normalised, which helps for the inter-comparability between sites.

- Yes, it is a good idea to highlight this fact in the results section. The following sentence was therefore added to § 3.5: *"Both the diurnal and the seasonal cycles were calculated as the strength of the autocorrelation function (see § 2.4 and supplement) so that the underlying parameters are de facto normalized and that the cycles between the stations can be directly compared."*

p9,L15f: These changes are rather large. Especially considering that the ranking between sites changes with domain size. It should be possible to solve the transformation problem in such a way that G8 and LocSlope are really constant with domain size. Why would the domain size change the local transformation/interpolation anyway? This needs to be redone.

- The authors looked again at the problem of non-constant values of LocSlope and G8 for various domain sizes. Both these values are constant in the traditional latitude longitude coordinates. The UTM projection leads to minor changes in the LocSlope and G8 that can be explained by two reasons: 1) if the analyzed domain extends beyond 2 UTM zones, map distortion problems occurs. This is, for example, the case for BEO plotted in cyan on Fig. 6 and having large G8 modification as a function of the domain size. 2) the interpolations needed to do the UTM projection can also lead to variation and G8 is very sensitive to these variations. The UTM projection is however necessary to ensure a similar handling of stations at very different latitudes.

Section 4: The name of the section is misleading. The section does not present a ranking of the sites by TopoIndex but more a discussion along their geographic location.

- 3.4: The title was changed to "*Relation between the ABL-TopoIndex and the station location*"

p12,L5: The more correct name would be "Rocky Mountains".

- "Rockies" was changed "*Rocky Mountains*".

p12,L15f: Why was MWO not discussed in this context as well?

- It is right that MWO is the North America station with the lowest ABL-TopoIndex and needs some comments. The following text is now added: "*Mount Washington Observatory is located in the Presidential Range of the White Mountains. It is the highest peak in the Northeastern United States and the most prominent mountain east of the Mississipppi River. MWO is consequently the North American station with the lowest ABL-TopoIndex due to very low hypso% and relatively high G8 and low DBinv.*"

p13,L13f: Looks like the authors themselves are surprised that there is any relationship between their TopoIndex and the chosen aerosol parameters ...

- The authors just wanted to state that their hypothesis was verified. If wrong criteria or parameters (see § 2.3) had been chosen, the correlation with aerosol parameters would have shown it. The word "happily" is however inappropriate in a scientific context and is (sadly) removed in the revised version.

p13,L26f: But hypso% is an even better predictor than TopoIndex. I guess that means that all other parameters only partly destroy this relationship but do not add any useful information. Especially the suspicious parameter based on water flow analogy, DBinv, seems to show very bad predictive skills (worse than altitude alone in some cases).

- As explained at the beginning of this document, the various modifications required by the referee's (smaller domain size, inclusion of the middle atltitude stations) as well as the removing of SUM time series from the correlation analysis lead to a somewhat different values of the Spearman rank correlation coefficients, even if the statistical significances remain similar for most of the case. In case of the correlation with the absorption coefficient, the importance of hypso% with regard to the other parameters constituting the ABL-TopoIndex decreases. LocSlope and G8 are now equally important parameters, whereas hypsoD50 has usually a lower statistical significance. We also checked that the statistically significance of the correlation between the ABL-TopoIndex and the aerosol cycles is clearly decreased if DBinv is removed from the ABL-TopoIndex definition. This is effectively the case, even if DBinv has globally bad predictive skills. Sections 3.5 and 4.2 were consequently modified.

p14,L18: Wasn't the point in Bianchi et al that the ABL influence is not a direct one, like you focus on here, but an indirect one of ABL air picked up a few days before arriving at the measurement site and therefore not being lifted by thermally induced flow but by convection or frontal systems.

- Thank you for this comment. It is correct that the greater ABL influence due to longer daytimes and stronger insolation does not relate to Bianchi et al., 2016. At this point, the authors just wanted to mention that stronger insolation usually also promotes NPF formation. The manuscript was modified consequently: "*The high correlation between the maximal diurnal cycle and the number concentration can also be explained by the promotion of NPF by the stronger insolation at low latitude.*"

p14,L30: Isn't the failure of the ABL-TopoIndex to identify these lower altitude sites a clear indication that the suggested method does not work at all? Otherwise these clear cases of larger ABL influence should be detected and the correlation should actually improve.

- De facto, the concept of the ABL-TopoIndex is really developed for high altitude stations with complex topography and cannot be applied to low altitude sites. NCOS was already identified as an outlier in the first version of the manuscript, and we found during the revision of the manuscript that SUM should also be removed from the correlation analysis because it is located on a high altitude plateau with a very smooth relief due to the ice sheet formation.
- The aerosol parameters used for the correlation analysis are also chosen to reflect the ABL influence at stations that are at least occasionally located in the FT. The causes of the aerosol concentration minima and maxim as well as the diurnal and seasonal cycles are completely different for sites that remain in the ABL during the whole day. In that sense, neglecting stations situated at too low altitudes (like ZEP) is absolutely reasonable. In our study, HPB and MSY, two middle altitude stations, decrease the correlation coefficient values without destroying the correlation. They are now included into the correlation analysis and the related section (motly section 3.5 and 4.2) were modified.

p15,L29: All of a sudden back-trajectories appear. It seems clear that these are not the hydrological flow paths. But from which model do these trajectories come from and why were they not used for all sites to also characterise the thermal flow systems (even if not fully represented in the model).

- Back-trajectories were calculated by the CHC data owners and used in other studies. They are used in this study just as a comparison with the main flow paths as a function of the ABL altitude. Anyhow, the section 3.6 was removed in the revised manuscript as recommended by the second referee, so that this point does not need a more detailed discussion.

p16,L16-17: This argument is going round in circles. The absorption coefficient is supposed to be an indicator of ABL influence because it correlates with topoIndex. But I though it needs to be shown that the topoIndex actually represents ABL influence ... Very confusing.

- This sentence is actually mixing some statements from both the results and discussion sections. It was therefore modified: "*Our results showed that of the three aerosol parameters tested in this study (number concentration, absorption coefficient and scattering coefficient), absorption coefficient has the greatest correlation with the ABL-TopoIndex values.*"

p16,L26: NO3 being NO3_aq or ions?

- This correspond to particulate nitrate ( $NO_3^-$ ) (Zellweger et al., 2003) and this is included in the revised version of the manuscript.

p19,L17f: These parameters are mostly know to the hydrological community but need additional introduction for the more atmospheric readership of the current journal. As mentioned before, it would have been better to provide such parameters to a statistical model with parameter selection in order to get an objective selection of parameters that may explain ABL influence. However, most these parameters would also follow the misleading assumption that thermally induced flow works just opposite to water flowing downhill and, therefore, should possibly not be considered at all.

- The authors did not consider at all that thermally induced flow can be considered as the opposite of water flow and most of these parameters were actually not used because of such discrepancies. However, as explained in the answers to the main comments (p.1 of this document), these parameters and the reasons for their rejection are now detailed as a table in the supplement (see Table S2 on p. 3)

Table1: Add the GTopo30 altitude of the grid cell containing each site, along with all the parameters derived for the site (potentially as supplement).

- The GTopo30 altitude of the grid cell as well as the mean for the 9 considered grid cells were added in the supplement with some comments. The altitude of the DEM grid cell as well as the mean altitude on the 9 used grid cells are given in the supplement Table S1.

Table2: The units for LocSlope should be m m-1 not Mm-1.

- Thanks for catching this! LocSlope has no units but there is a factor of $10^{-3}$ because the altitude is given in m and the horizontal distance in km. The values and units in Tab. 2 are corrected in the revised version.

Figure1: The figure quality is not state of the art. I suggest to use a topographic image as background. Larger station labels or symbols. Legend for mountain ranges.

- You will find thereafter Fig. 1 similar to the first version but with the right color scheme and a second version with the continental topography beyond the station location. If the first version allows to clearly visualize all stations, the second version also gives some information about the highest massifs around the world. The authors put the second version in the revised manuscript, but let the editor chose which figure should be finally used in the manuscript.

[Figure]

Figure2: The schematic is confusing. If you want to underline that there is a higher ABL influence on the right, why not show a visible, partially terrain following ABL in the mountainous area and an aerosol layer resulting from lift over processes. The schematic on the left is a very poor image of a mountain shape. Looks more like a life buoy with a signal post but not like the profile of a volcano.

-The referee is right, the schematic view was somewhat crude. The left schema is now changed. Since section 3.6 was deleted following the referee's comments, the added ABL was removed from Figure 2.

Figure4: The thick cyan line is not mentioned in the caption.
- OK, this now mentioned in the figure caption: "*The main flow paths from the station grid cell are given by the cyan lines.*"

Figure6: Sub-panel labels are missing in the figure but are used in the caption.

- OK, the sub-panel labels are now written in the figure.

Figure8: What are the different shades of colours? Neither explained in caption nor text.

- Some colors were changed in both Fig. 1 and 8 so that the color scheme of both figures are now similar. This is now mentioned in the figure caption of Fig. 8*: "The color scheme corresponds to that in Fig. 1."*

Figure9: Very difficult to comprehend. Too many colours and symbols in one plot. Why not display negative correlation coefficients as such on the negative part of the y axis. Instead of circles, different sized symbols should be used for different significance levels.

- As suggested by the referee, the statistical significance is now given by different symbol sizes and this clearly increases the readability of the figure. We keep however the negative correlation as downward triangles to keep the direct comparison between the absolute value of the correlation coefficients. Since the anti-correlated topography parameters are used as 1/parameter in the ABL-TopoIndex, the absolute correlation value is more important that its sign.

**References:**

Andrews, E., Ogren, J. A., Bonasoni, P., Marinoni, A., Cuevas, E., Rodríguez, S., Sun, J. Y., Jaffe, D. A., Fischer, E. V., Baltensperger, U., Weingartner, E., Collaud Coen, M., Sharma, S., Macdonald, A. M., Leaitch, W. R., Lin, N.-H., Laj, P., Arsov, T., Kalapov, I., Jefferson, A. and Sheridan, P.: Climatology of aerosol radiative properties in the free troposphere, Atmos. Res., 102(4), 365–393, doi:10.1016/j.atmosres.2011.08.017, 2011.

Collaud Coen, M., Weingartner, E., Furger, M., Nyeki, S., Prévôt, A. S. H., Steinbacher, M. and Baltensperger, U.: Aerosol climatology and planetary boundary influence at the Jungfraujoch analyzed by synoptic weather types, Atmos. Chem. Phys., 11(12), 5931–5944, doi:10.5194/acp-11-5931-2011, 2011.

GAW Report No. 200. WMO/GAW Standard Operating Procedures for In-situ Measurements of Aerosol Mass Concentration, Light Scattering and Light Absorption, (Edited by John A. Ogren), 134 pp. October 2011.

Guo, J., Miao, Y., Zhang, Y., Liu, H., Li, Z., Zhang, W., He, J., Lou, M., Yan, Y., Bian, L. and Zhai, P.: The climatology of planetary boundary layer height in China derived from radiosonde and reanalysis data, Atmos. Chem. Phys., doi:10.5194/acp-16-13309-2016, 2016.

Pal, S. and Haeffelin, M.: Forcing mechanisms governing diurnal, seasonal, and interannual variability in the boundary layer depths: Five years of continuous lidar observations over a suburban site near Paris, J. Geophys. Res. - Atmos., 120(11), 936–956, doi:10.1002/2015JD023268, 2015.

Poltera, Y., Martucci, G., Collaud Coen, M. and Hervo, M.: PathfinderTURB : an automatic boundary layer algorithm . Development , validation and application to study the impact on in-situ measurements at the Jungfraujoch ., Atmos. Chem. Phys., 1–34, doi:10.5194/acp-2016-962, 2017.

---

## Referee Report (RR1)

2nd review of Collaud Coen et al. "The topography contribution to the influence of the atmospheric boundary layer at high altitude stations"

The authors have at length responded to both referee comments. They mostly argue against major changes in the applied methods only dropping one part of the manuscript which was using hydrological flow path as an analogy to atmospheric flow, which was strongly critisised. Reading the authors responses and the revised manuscript I am only partly satisfied with the replies and modifications and suggest further modifications before publication in ACP is possible.

Points raised in the first review and commented by the authors:

1) **Neglecting lifting processes other than thermally induced, convective transport**

My concern is not completely met by the authors reply. I still think it would be possible to create a meteorological criterion that would indicate situations with likely thermally induced flow from existing global scale model products. Don't forget that the latter have resolution down to 0.1 degree by now. Also it has been done successfully before with observational data, so why not check with model data. However, I see that this may go well beyond the scope of the current analysis and actually the authors' final point (although strictly valid only for one site) may provide an avenue of argumentation that is useful in the context of this study but may require some rethinking of ABL versus aerosol influence: It is correct that Zellweger et al. (2003) showed that increased aerosol surface area was mainly observed during thermally induced lifting events, whereas for other (gaseous) ABL tracers enhanced concentrations where also observed during other lifting events (synoptic and foehn). The reason for this is most likely washout of aerosol during the other lifting processes, since both foehn and synoptic lifting are usually connected with precipitation. So the argumentation in the current study should go somewhat like this. 1) In most cases aerosols can be a tracer for recent ABL contact, 2) however, many lifting processes co-occur with precipitation and, hence, aerosol washout. 3) The lifting process that often occurs without precipitation is thermally induced flow (but mind: thermally induced flow often also leads to deep convection and convective precipitation). 4) Therefore, the potential of this lifting process on aerosol concentrations at high altitude sites was studied. This should also be reflected in the title, which could be changed to: "Characterisation of topographic features influencing aerosol observations at high altitude stations"

This would not claim the ABL influence but rather focus on the aerosol concentrations themselves and take notice of the fact that not all ABL events carry aerosols with them.

2) **Emissions**

I don't agree that the emissions will not have an influence on the relative magnitudes of the diurnal cycle. The relative location of the dominating emission source is of crucial importance. One cannot assume that concentrations in the ABL or RL are horizontally homogeneous! For example the absence of local to regional emissions is most likely the reason why the sites ZEP, SUM and NCOS are later kicked out as outliers. Including a parameter that looks at the emission distribution as functions of altitude and distance from the sites would certainly be valuable and would potentially explain some of the inter-site differences. This should at least be given more though and attention in the outlook and should be looked at in future studies.

**3) ABL events**

My comment had in mind that transport frequencies will play a role as well when looking at the median data. I can see that 5[th] percentile is more likely to represent FT conditions and 95[th] percentile pollution events. But there should be more information available from the complete distribution function. The aerosol parameters may be distributed log-normally. In which case one could still look at the log-transformed values to identify different degrees of skewness. For a site frequently influenced by pollution events one would expect a positively skewed log-normal distribution. The diurnal cycles present a way forward from the simple absolute concentrations (as in Fig 9). But still these are averages over a whole year and the importance of thermally induced transport varies strongly with the amount of energy available for convective transport, depending on latitude and season. This might be the reason why, as the authors state, "these are however much more difficult to statistically extract from the time series". So if no kind of event detection or filtering of the observations for days with thermally induced transport is possible in the context of this study, I suggest to further comment on the factors that make the evaluation of the diurnal cycles difficult.

**4) Topographic parameters and their selection**

I am still not happy with one specific parameter and the way the parameters are finally selected and used in the ABL index. Just looking at Fig 9 and Fig 10 it is apparent that DBinv has almost no predictive power (besides the one lucky punch for the seasonal amplitude of the absorption coefficient which is not explained in any physical way). The parameter simply does not make any sense in a meteorological context (this was my point 5 in the previous review). There simply is no such thing as a reservoir for air convection. The parameter is large for sites that are at the highest point in their domain (e.g. BEO), so yes it has something to do with the relative topography (but hypso% seems to do the better job), but there is no reason whatsoever why a hydrological divide should also hinder atmospheric flow. Just look at the two examples given in Fig 4 and Fig 5. BEO is the highest point in the massif, so according to the authors, all air in the domain has the potential to flow upstream to the site. In contrast, the inverse drainage basin for PYR is limited to the north and east. But what happens during a fair-weather day with easterly winds at the site? FT air will move over the mountain ranges east of 90.1°E, receive the well know enrichments from ABL inputs (just as the authors describe for the Nyeki et al (2000, 2002) observations) and arrive at the PYR site. The hydrological border presents no boundary whatsoever for this transport. In contrast, for BEO I would argue that large parts of the "drainage basin" are at altitudes too low to trigger topographic convection to altitudes as high as the site itself. In addition and as mentioned before, a flow along the slopes to the highest point simply does not happen due to mass balance reasons (please check textbooks on mountain meteorology).  In summary, I would still suggest to drop DBinv from the calculation of the ABL index (as was already done for the flow paths which were based on the some wrong analogy). There is nothing lost in terms of explanatory power of the ABL index.

Instead one could also argue to also include the absolute altitude and the latitude in the ABL index, since they have even more predictive skills than DBinv (see Fig 9).

Another point is still the non-objective parameter selection. Thanks to the new table S1 one now has a chance to have a look at the used parameters. A quick "pairs plot" (see below) revealed that loc.slope and G8 are actually quite strongly correlated, which could mean that actually only one of them should be selected in a final predictor. Otherwise one gives this feature squared weights (due to geometric mean). The figure also shows that DBinv is largely independent of the other parameters (which could have been good), but that it also does not correlate well with the ABL index (the reason being its smaller relative variability compared to all other parameters). My idea about parameter selection was to build a regression model (linear or generalised) using the different topographic parameters as predictors and apply it to the aerosol parameters. This is in contrast to constructing one single index as the one and only predictor. Combined with a parameter selection procedure one could then test which parameters are the best predictors for which aerosol parameters. A regression model constructed in such a way would also allow some kind of physical understanding of the topographic parameters on the aerosol parameters that goes beyond rank correlations. A simple analysis of variance would also be beneficial and more conclusive as only looking at correlation coefficients between individual topographic parameters and aerosol parameters. I leave it up to the authors if they would like to improve their method in the suggested way or keep it in mind for future studies, but I strongly suggest to drop DBinv from the analysis.

[Figure]

**Figure 1: Pairs plot of topographic parameters and ABL index. Colour indicates strength of spearman rank correlation (cyan: negative correlation, magenta: positive correlation, darker colours represent stronger correlations).**

Further comments on the revised manuscript (page numbers and lines refer to the track-changes version attached to the authors' response). These are partly new and came up when having a closer look at the topographic parameters now available in Table S1.

**P2, L2**: "drainage basin for air convection". Once more there is no such thing and the analogy is not valid. Since I suggest dropping parameter DBinv this term and concept can also be removed from the manuscript altogether. See the discussion above.

**P6, L5f**: Not clear what the accuracy refers to: the altitude differences between sites and GTOPO or the specification of GTOPO? However, what you should be worried about is not the accuracy of GTOPO which refers to the average altitude in each grid cell but the representativeness of GTOPO for the whole grid cell and the resulting mismatch with your station altitudes. So the results in Table S1 are the more important numbers.

I am also confused about the new text inserted in the revised manuscript. It is not the same as given in the replies (see page 14 of replies), which discusses the altitude differences in much more detail. Just there it is said that the altitude difference for NCOS is 1100 m, which is not true (see Table S1). However-er, some of the other large altitude differences should be mentioned in the manuscript not just in the reply.

**P7, L5**: Why are these two sites discussed when there are sites like WLG and MUK also showing larger hypso%? There are many more sites with large values as seen in Table S1. So the statement that most sites have hypso% smaller 5 % is not true! Actually 22 of 46 sites have values larger 5 %. Please be more precise in all your discussion! Don't let the reviewers do your work!

**P7, L14**: 6 out of 46 sites are situated under hyps50. What is the consequence? Should they even be discussed? The new sentences just before suggest that these sites cannot claim to be high altitude stations!

**P7, L16f**: This is a hypothesis so far not a fact. Please rephrase.

**P7, L23**: A "plateau" is a high plain (https://en.wikipedia.org/wiki/Plateau). So here the word valley or plain should be used instead.

**P9, L30f**: This argumentation is confusing and misleading. First of all it is not the hydrological drainage basin that is used in the ABL index but the inverse drainage basin (5d rather than 5c). Then the size of the basin is made responsible for the high ABL index. However, just above we have seen that a large inverse drainage basin does not lead to a high ABL index (inverse drainage basin for BEO is actually larger than the one for PYR). So I would say that one of the other parameters is the more important one in terms of this discussion, most likely hypso% which differs by 3 orders of magnitude between the two sites. However, the discussion of DBinv will need to be removed from the manuscript anyway.

**P10, L6**: This should be 50 x 50 to 1000 x 1000 km$^2$.

**P13, L6**: MWO supposed to have a "low DBinv", but a value of 127457 for DBinv at MWO means that the site has a DBinv larger than the mean, larger than the median and even larger than the 75% percentile of all sites (see Table S1).

**P14, L12ff**: Once more, I can only say that it is surprising that the ABL-Index does not work for such sites that are clearly not high altitude sites. How can we then trust the method to rank sites at different degrees of high altitude? One common factor at all three sites is the absence of local to region emissions! Hence, aerosols no longer are a good estimator of FT vs ABL conditions.

**P15, L9ff**: One should mention that these two parameters are actually strongly correlated (r=0.76, spearman)! The highest correlation between all parameter pairs! So it is no surprise that they both show similar characteristics and actually one could be sufficient for this study.

---

## Editor Decision (ED1)

**Editors comment paper acp-2017-692 by Collaud Coen et al.; The topography contribution to the influence of the atmospheric boundary layer at high altitude stations; now changed to "The topography contribution to the influence of the aerosol layer at high altitude stations"**

Dear author, co-authors,

It has taken quite some extra time to come to the point of sharing an editor's comment with you. The last round of feedback was provided halfway April and you have then submitted your response and last revision the $7^{th}$ of May. Sorry for this > 1 month delay in my action upon this last submission but have been busy with work related travelling in May and last week's being out of office dealing with personal issues.

We are now at the stage that the one of the reviewers is satisfied with your responses and the revisions that have been made in the manuscripts in some iterations. However, the second reviewer still had some significant criticism and comments regarding your last revision and the response to the provided reviews. I have been going through all the files including the last review and your last response and the revised document. Based on this I invited the reviewer to indicate if this last round of revisions and your response has properly addressed the concerns/issues of the earlier version of the manuscript. In addition, reading over again the ms I still came across quite some statements, sentences that I had to read over again also not always being convinced that these were correct regarding grammar but also not always optimally expressing that what you would like to express. Below, you can find these points that came across and that also need to be further addressed in another revision. I recommend you to ask one of the native English speaking co-authors to carefully check once more again the whole documents for the text to remove these flaws.

Overall, given the significant feedback and required revisions of the ms, it has been so far challenging to keep carefully track of all the changes but hope that at the end all this feedback and the revisions will ultimately result in a strongly improved version of the paper.

Regards, Laurens Ganzeveld

Major and minor comments:

Title; you have changed the title also considering the changes introduced in the ms based on the reviewers comments. However, this changed title is also according to me still not optimally covering the actual contents of the paper; especially the use of the term aerosol layer is not reflecting the contents. The suggested change in title by one of the reviewers seems to cover much better the actual content. Your study focuses on identifying the influence of topography on aerosol measurements at high-altitude stations. In addition, it seems that have not tackled the issue of the reviewer on the use of global modelling products to also assess the role of topography in aerosol properties at higher elevations. I am myself not so convinced that the resolution is already sufficiently high to indeed use model products for this, at least not from global models. There are though meso-scale models that can resolve some of the fine-scale meteorological features at resolutions down to some km scale (e.g., WRF). My point is that you should at least address this remark by the reviewer even if at the end

you decide not to include this aspect in your paper.

Page 6, line "non-GIS environment(Schwanghart and Scherler, 2014)" put space there.

Page 6; line 20: "Based on these criteria, the red station on Fig. 2 will be less influenced by the ABL…", alternative; "Based on these criteria it can be inferred from Figure 2 that the "red" station will be less influenced by the ABL….

Page 7, line 2: "have hypso% *values* larger than 50%."

Page 7, you refer in line 16 for example to small spatial scale and in the definition of the previous criteria to large spatial scale; here it is essential to indicate (again) what you deem being a small and a large spatial scale.

Page 7; line 23, what do you mean with "and there are some steps for CHC and BEO", what steps ? and steps of what size?

Page 9: line 26: "PYR (5079 m) is the second highest station considered here, but..", reading this section and statement this line was confusing since the term "here" suggests that this is referring to the second highest station of this case study. You mean here that PYR is the second highest stations of all stations considered in the study and would then also state it this way for clarity.

Page 10, line 2: "The ABL-TopoIndex depends on the size of the chosen domain (Fig. 6a) so that the various algorithms were tested to several domain sizes ranging from 50 to 1000 km2. The gradient G8 and the local slope LocSlope are calculated on small fixed horizontal scales (0.5-1 and 10 km, respectively)". I had to read these two sentences a couple of times also not being convinced that this is the most optimal way to express what you intent to say. There are actually more of these sentences in the ms and would anyhow propose to still have once a native speaking English co-author (I guess there is one given the large list of co-authors) to critically check the ms for such potential flaws. Alternative: "Since the ABL-TopoIndex depends on the size of the chosen domain (Fig. 6a) we have conducted an evaluation of the sensitivity of the various algorithms to the domain size using a range from 50 to 1000 km2. The gradient criterion G8 and the local slope criterion LocSlope are calculated on small fixed horizontal scales (0.5-1 and 10 km, respectively)"

Page 10: line 9: "the concentration of thermally lifted pollutants"

Page 10, line 9/10: "The hypso% decreases continuously for stations situated in a dominant position in their mountainous massif such as JFJ, SBO or BEO (Fig. 6c)". This is another example of a sentence that should be read carefully and revised. "continuously" here expresses something like over time whereas you want to express here that this parameter increases with an increase in the domain size.

Line 15: "with domain size", change to "with an increase in domain size"

Page 10; line 23-24 "To compare these two parameters, we show in Fig. 7 the ABL-TopoIndex as a function of the altitude for all grid cells in a 5km x 5km domain

around a selection of stations"

Page 10; line 26: "..very steep and ASK a very flatt ABL-TopoIndex decrease with altitude"; first of all "flat", then secondly, what is a flat decrease?? I guess you would like to say that there is a strong increase and a small decrease or?

Line 33: "were constructed", alternative "are located"

Page 11: line 6: "are grouped on Fig. 8": are grouped as shown in Fig. 8 (change all this consistently in the text, e.g. "on Fig. 9"

Page 12, line 10: "their proximity *to* other massifs such as the Alps"

Page 14, line 14-15: "…ABL influence, in case of lifting processes without precipitation, is found for the ABL-TopoIndex…."

Page 14; line 18-19: "(mean  altitude over the 9 grid cells, similarly to the ABL-TopoIndex calculation)"

Page 15, line 12, "lowest and the greatest monthly amplitudes", should be according to me "smallest and largest monthly amplitudes", also check further the ms for this: e.g. "the greatest ABL influence" should be "the largest ABL influence" (possibly a matter of taste). You use many times the term greater it should be larger/largest or higher/highest, e.g. "higher correlations"

Page 15, line 21: "diurnal cycle minimal and maximal strengths of the absorption coefficient". This expression reads also not well. You figure caption text seems to express it in the proper way, modify this text.

Page 16, line 6: "First the possible species and phenomena enabling the estimation of the ABL influence", what do you mean here with species?? Do you refer here to compounds. I also think you generally want to refer here more generally to "parameters"

Page 16, line 27, "NOy" y lowercase

Page 16, line 28: "should be in most cases"; "could be in most cases"

Page 16, line 29 "hence involving aerosol washout"

Page 17: line 4-5: Your statement about using a model to further assess how pollutants can be used as a proxy for BL influence reads weird: what do you mean with a thermodynamic model? "…bounded to a 3D thermodynamic model adapted to complex topographies would be required before using absolute pollutant concentrations as indicators of ABL influence at high altitude sites", rather "..constraining simulations with meteorological models able to explicitly resolve the role of fine resolution orography would be …." (see also my previous comment on the title/introduction and first major comment of the reviewer).

In addition, this text is part of your modification of the ms responding another major

comment provided by one of the reviewers. There is another part of this modification that raises quite some questions: " A further use of DBinv to restrict the area of potential pollution sources could also be envisaged since this parameter describes the domain from which pollutants can reach the high altitude station by convection and without crossing topographical barriers. This delicate issue can however be avoided by instead considering dynamical parameters such as the various temporal cycles". What is delicate here? What are the dynamical parameters? You mention here the temporal cycles (diurnal and seasonal?) but in what parameters. I wonder what the reviewer will express about this modification. I am myself not very convinced and consequently suggest you to check this once more again carefully.

Page 17: line 14-15: " Usually the spring leads to higher aerosol loading than the autumn probably related to higher ABL height in the spring"; do you refer here to a higher aerosol loading/concentrations at the high-altitude stations?? I don't see how a deeper ABL would result in a higher aerosol loading in the ABL, actually the opposite would be expected not having any changes in the sources (more dilution)

Page 18: line 7: "has similar dependency as the ABL as a function of latitude", dependency on what? I know that in the following sentences you give examples but rephrase this sentence.

Page 18, line 16-17: "modify the theoretical cycles and lead to a broadening of the time of the extrema. These difficulties make obtaining clear statistical cycles another reason contributing to the observed low correlations", this text is another example of statements that need definitely to be rephrased. What are theoretical cycles: cycles that you would anticipate based on basic theory? What is meant with a broadening of the time? It would be broadening the time frame or increasing the duration, and which extrema?? And the second sentence needs complete revision.

Page 18, line 26- 27: Correlation between topography and aerosol parameters and "in Sect. 3.5"

Page 19, line 22-23: "Globally, NPF is the reason why the greatest correlations are found with the 50 percentile of the number concentration, instead of with the 5 percentile found for the absorption and scattering coefficients", also this sentence needs to be rewritten: correlations with?

Page 20, line 7,when you mention these terms "the Efremov-Krcho classification, the hypsometric curve" you should shortly explain them but also indicate why you considered to include these terms in the analysis. On the other hand, the discussion is already now (way too) long.

Page 20, line 28: "It is …"

References to "de Wekker" should be listed in the references under the "D" and not the "W" references.

Figure 1: the station names with the different colors come out sometimes quite poorly, like the stations in the US. Use only or white or black characters?

---

## Author Response (AR4)

**2$^{nd}$ of February 2018: Response to the reviewers comment's:**

Dear editor, dear referees,

We would like to thank you for all your comments. This input has allowed us to refine the manuscript by adding more thorough detailed explanations, to correct some minor points and to improve in a large sense the manuscript. This response comprises three sections, first the answers to the main comments of both referees, then the answers to the specific comments of the first and second referees. The authors are conscious that the methodology and topic of this study are to some extent new concepts and that they consequently raise a number of comments. We hope however that in this document we have addressed the referees questions fully and clarified the aspects that needed further elucidation. The co-authors are unanimous that this manuscript presents a valuable methodology for interpreting atmospheric measurements at mountain sites across the globe. This manuscript presents a new technique and extensive data analysis applicable to many of your readers. Based on our extensive efforts in addressing each comment of the reviewers, we ask you to accept for publication the revised version of the manuscript in ACP.

First we want to mention that the values of the ABL-TopoIndex and of the correlation coefficients presented in figures 9 and 10 have changed from those presented in the first version of the manuscript. The differences in the ABL-TopoIndex values are due to the modification of the domain size to 500 km x 500 km. The correlation coefficients changes are due to the modification of the domain size, to the exclusion of SUM due to its outlier status similar to NCOS and to the inclusion of the middle altitude stations (HBP and MSY) in the correlation analysis. Further explanations are given in the following answers to the referee's comments.

1. **Answers to main comments of both referees**

   - **GIS and Topotoolbox**: TopoToolbox is a set of matlab functions that offers analytical GIS utilities in a non-GIS environment. In that sense it is possible to apply GIS-specific methods and to analyse aerosol parameters and cycles in the same environment as the topographic analysis. The TopoToolbox enables the analysis of relief and flow pathways in a DEM as well as the calculation of standard terrain attributes (slope, curvature, flow accumulation,…). The basic functionality of TopoToolbox was therefore used, but further programming was necessary in order to calculate all the necessary parameters constituting the ABL-TopoIndex. As suggested by the referees, the authors added some further clarifications to describe these parameters with sufficient details in the paper, so that the ABL-TopoIndex could be reproduced in any other programming language.

   - **Domain size for the calculation of the ABL-TopoIndex:** The ABL influence at high altitude sites can be divided into a local phenomenon bringing polluted air masses from the adjacent valleys to the measuring station and a broader impact including the whole mountainous massif and a possible influence of nearby plateaus and plains. Poltera et al. (2017) clearly demonstrate that convection above the adjacent valleys rarely influences the high altitude sites but, when it is the case, this local convection does lift air masses with a certain aerosol load. The aerosol layer that comes from a much broader region has a lower aerosol concentration but influences the high altitude stations over a long period of time. An airborne Lidar measurement of the ABL top over the whole alpine massif (Nyeki et al., 2002) clearly stated that the convective boundary layer is formed over a large-scale and leads to an elevated and extended layer. They also quantified that this "large-scale" extends more than 200 km from the mountainous massif. The rectangular domain size of 750 km x750

km centered on each site corresponds to a distance of at least 375 km in each direction and was initially chosen to ensure the inclusion of the entire massif and a further portion of the adjacent plains. To address the concerns of the reviewer we have restricted the domain to 500 km per side, but think that a domain size smaller than that would no longer correspond to the reality of the aerosol layer formation. The authors also agree with the second referee that CBL flow will not advect air masses from a distance as large as 375 km. Without precipitation, the residual layer or aerosol layer will however expand over several days. The distances of 375 km and 250 km are covered in 21 and 14 hours, respectively, at an average advection velocity of 5 m/s. The chosen domain size corresponds therefore largely to the development of the CBL and its merging into the residual layer.

- **Methodology and set of quantitative parameters:** The authors are aware that this study consists mostly of a new methodological approach with concepts probably unfamiliar to many atmospheric scientists. The goal was to try to statistically quantify the role of the topography in the ABL influence at high altitude sites. The authors intentionally did not include dynamical parameters such as wind fields that would have required the use of atmospheric models such as ECMWF. While the applied methodology was described with some detail in the original submission we have expanded, and to some extent reorganized the description based on the reviewers comments. First we define a number of topographic criteria that should determine the ABL influence at a high altitude site; second, quantitative parameters are found for each topographic criteria; finally, statistical methods that are valid for environmental studies are applied to the quantitative parameters. Tested qualitative parameters that were finally not selected are briefly described in the supplement to the manuscript. The reasons for not keeping these criteria to calculate the ABL-TopoIndex are now extensively described in the revised manuscript supplement, so that the reader can now better understand the final choice of parameters.
As already mentioned in the paper (section 4.3), the ABL-TopoIndex could probably be improved by adding some further parameters and its validity can also be assessed by other pollutants measurements at high altitude sites.

- **Weak correlations between topography and diurnal and seasonal cycles:** As mentioned by the first referee, the correlations between the topography parameters and the aerosol diurnal cycles are surprisingly weak. This is due to three main reasons: For most of the stations, there are a lot of days where the diurnal cycles are obviously visible. It is however quite difficult to extract the diurnal amplitude as a statistical value due to several factors including: non-regularity in diurnal cycle time of occurrence (e.g., due to different synoptic weather type, cloud presence, advections, long range transport); in the strength of the diurnal cycle (insolation amount, cloud presence); in the absolute level of aerosol present (e.g., due to presence of residual layer, superposition of long range transport); and to the superposition of both seasonal and diurnal cycles. The only possible methodology is to remove the first lag autocorrelation in the data, before extracting the diurnal cycle amplitude from the autocorrelation at 24h (see the supplement to the paper). The removing of the first-lag autocorrelation is a necessary step that introduces noise in the data. Additionally, as explained in the manuscript, only stations partly influenced by the ABL will show a clear diurnal cycle. Stations that remain the whole day in the FT should exhibit no diurnal cycle, whereas stations always in the ABL will have different diurnal cycles due to other periodicity in the sources and to the mixing conditions. As the location of the station with relationship to the ABL can change with season this further complicates the identification of diurnal cycles. Another factor is the presence of the residual layer during the night in summer, which

drastically decreases the amplitude of the diurnal cycle. In terms of relating topography and seasonal cycles. Additionally an important thing to consider is that many of the datasets used here are shorter than 5-6 years leading to difficulties in the determination of the seasonal cycles. This is probably a primary cause for the lack of correlation between seasonal cycles and topography parameters. We have revised the manuscript to make this point more clearly as described in our response to referee#1 below.

**2.    Answers to referee #1 comments:**

This paper presents five metrics that can help quantify the boundary layer impact at high altitude stations. The metrics are based on topographic data and provide information on topographic characteristics including steepness, height difference between station and adjacent valley, and size of the drainage basin. The metrics are calculated for large number of stations. The focus in this paper is on a subset of these stations where aerosol measurements are made.

Overall, I think that the paper is a decent contribution to the scientific literature. The novel part is the quantification of the topographic characteristics surrounding a high altitude stations. The contribution of certain topographic characteristic to trace gas measurements at these stations is often speculated and discussed and it is nice to see a paper where an attempt is made to quantify the characteristics. I am not sure though how useful the characterization of the topography as done in the current study will be for future studies/site planning.

I found it rather surprising that correlations between topography parameters and the diurnal cycle are weak.

Some further explanation are given in the answers to the main comments on page 2 of this document and the manuscript was revised in order to better explain the reasons for weak correlations. The manuscript was changed at § 3.5 : "*The ABL-TopoIndex is s.s. correlated with the diurnal cycle minimal and maximal strengths of the absorption coefficient. This correlation is once again principally due to the hypso% and G8, and to a lower extent, the LocSlope. The correlation with the diurnal cycle minimal amplitude occurs because the stations that remain in the FT during the whole day should not present any systematic diurnal cycles. The maximal amplitude of the diurnal cycles occurs when the site is in the FT during the night (without any influence of the RL) and influenced by the ABL during the day. The only s.s. correlation with station altitude is found for the scattering coefficient seasonal cycle. Similar to the correlation with the percentiles, there is a high anticorrelation between the particle number concentration diurnal cycles and G8 suggesting that the slope steepness in the vicinity of the stations inhibited both the transport of polluted air masses and NPF. Apart from a correlation at 90% confidence level between DBinv and the absorption coefficient, the lack of further s.s. correlations with the seasonal cycles can be attributed first to the relatively small time period (2-5 years) covered by most of the datasets leading to difficulties in the statistical determination of a yearly periodicity due to inter-annual variability, second to the low aerosol concentration at high altitude sites inducing measurements part of the time near the detection limits of the instruments (see for example the problem with the absorption coefficient at § 2.4) and third to the necessary whitening procedure (see supplement) increasing the dataset noise .*"

And also at § 4.1, the following sentence was added: "*The impact of the RL on the aerosol concentration is probably one of the most important reason to the low correlation between the topographical parameters and the aerosol cycles.*"

1) the choice of the five metrics appears somewhat subjective. At some point in the manuscript (section 4.3) it is stated that "Several other parameters such as the topographical wetness index, the catchment area, the accumulation, dispersion and transit percentages, the hypsometric index and the prominence were tested but were finally eliminated as being not relevant for various reasons." . It remains rather vague why these parameters were eliminated. It would be good if the authors could make a list (e.g, in a table) of all the relevant parameters that the "TopoToolBox" produces and then also clarify what exactly was done to come up with the final five parameters.

- As explained in the answers to the main comments (p. 1 of this document), the tested parameters are only partly provided by the TopoToolbox, some of them were developed or modified for this study. TopoToolbox provides a set of Matlab functions to analyze the relief and the flow pathways in the digital elevation model, some of them having absolutely no direct relation with the ABL-TopoIndex. In that sense, it is not possible to list all the TopoToobox functions. We have now included more discussion of the reasons to choose the 5 used parameters (§ 2.3) and we have also added Table S2 in the supplement to describe some other topography and hydrology parameters and the motivation of their rejection as relevant parameter to calculate the ABL-TopoIndex: not to use some parameters have to be explicit and the manuscript was accordingly changed:

| Parameter | Definition | Reason for rejection |
|---|---|---|
| Upstream catchment area= flow accumulation | Upstream area contributing to the flow accumulation at the grid cell | 1) It has no direct effect on the ABL influence since it lies at higher altitude than the station
2) It is a partial measurement of the area higher than the station elevation, but only on the mountain side where the station is situated |
| Topographical wetness index = compound topographic index | =ln(A/tan(B)), where A= upstream catchment area and B= slope gradient. It is a measure of the extent of flow accumulation at the given point; it increase as A increases and B decreases. | The wetness index is a ratio of two parameters. The slope gradient is already used (G8) in the ABL-TopoIndex and A was not considered as useful to describe the ABL influence (see previous point). The authors prefer single to combined parameters |
| Drainage basin = dispersive area | Downslope area potentially exposed by flows passing through the given point on the topographic surface | Air convection flow paths cannot be directly assimilated to water flow. The drainage basin in the inverse topography was consequently used as describing the size of the "reservoir" for air convection. |
| Efremov-Krcho landform classification scheme, Dispersion and transit percentages | It is a landform classification scheme (Florinsky, 2012) attributing a characteristic (dissipation, transit or accumulation) to each grid cell. | This classification scheme depends on the curvature of the terrain and, contrary to water flow, it has no relevance for air masses transport. It was however tested on some stations but failed to give a clear characteristic for the station region. |
| Hypsometric curve (HC), hypsometric integral (HI) and | The shape of the HC and HI values provide vital information about erosional stages of the relief and tectonic, climatic and lithological | Both HC and HI characterize the shape of the whole mountainous range and are therefore not defined for the station location. |

| | factors controlling landforms development. Convex-up curves are typical for youthful stage and concave-up curves of old stage. (Siddiqui and Soldati, 2014) | They cannot be used to characterize the station location. |
|---|---|---|
| hypsometric index (HI) | HI= (mean elevation-minimum elevation)/(maximum elevation-minimum elevation) allows different watersheds to be compared regardless on scale. It could reflect both tectonic activity and lithological control. (Siddiqui and Soldati, 2014) | HI also concerns a domain and not the station location. It cannot be used to characterize the station location. |
| Topographic prominence | It is the vertical distance between a summit and the lowest contour line encircling it but containing to higher summits within it. It is a measure of the independence of a summit. | It is not applicable to stations that are not situated at a summit. Moreover, since it restricts the area to a domain without higher summits, it corresponds to domains with very different sizes depending on the station. |

2) How are the parameters produced by TopoToolBox similar to or different from the more widely used ArcGis software packages? many people who would like to apply the concept of a topographic index may be familiar with ArcGis software packages so a way to make the concept more widely used is to explain how these parameters could be calculated using ArcGis software.

- TopoToolbox just offers GIS utilities in a Matlab environment. The parameters used to calculate the ABL-TopoIndex are hopefully clearly enough described to allow any user to find or to write their equivalent in any GIS software packages, including ArcGis. For example, catchments or watersheds are probably calculated in a similar way. Since we have not used ArcGis, it is difficult to estimate exactly the potential of ArcGis compared to TopoToolbox.

3) page 3, line 30/31: Free convection cannot be driven by forced mechanical convection. This sentence is technically incorrect.
- The referee is correct that free convection cannot be due to any forced mechanism. The sentence was modified: "*In the case of cloudy or rainy conditions as well as in the case of advective weather situations, free convection is no longer driven primarily by solar heating, but by ground thermal inertia, cold air advection and/or cloud top radiative cooling.*"

4) section 2.3, line 13: It should be explained here why a domain size of 750x750 km was chosen. The authors discuss somewhat later in the manuscript the sensitivity to domain size but a justification for the chosen domain size should be provided here. The domain size currently sounds rather arbitrary.

- The answer to this question is given in the answers to the main comments (p. 1 of this document). A large domain size has to be chosen in order to take into account the whole mountainous range and part of the adjacent plains/plateau contributing to the formation of the aerosol layer. In that sense a domain of 500x500 $km^2$ could also be justified and was consequently used in the revised version of the manuscript, leading to small variation of the ABL-TopoIndex for some stations. We have also clarified our reason for the size of domain in the revised manuscript at § 2.3: "*A quantitative estimation of these criteria depends clearly on the domain considered. The minimal size requirement for such a topographical analysis is that the domain should contain the whole mountainous massif. An airborne Lidar measurement of the ABL over the Alps (Nyeki et al., 2002) clearly stated that the convective boundary layer is formed over a large-scale and leads to an elevated and extended layer. It*

*also quantifies this "large-scale" to extend more than 200 km from the mountainous massif. A rectangular domain size of 500 km x 500 km centered on each site was then chosen (see § 3.2 for a discussion of the effect of the domain size)."*

5) page 7, line 6: "with the size of the local scale depending on latitude". Please explain/expand.

- The gradient is applied between 2 grid cells and the length of the domain covered by 2 grid cells depends on latitude (see § 2.2) and correspond to 2-4 km. The manuscript was changed:*" This parameter takes into account the slopes towards lower and higher elevations over a local scale (2-4 km that is the distance covered by two grid cells, with the size of the grid depending on latitude)"*

6) page 8, line 18: use plural "autocorrelations". Also on next line, "auto-correlations" is hyphenated. Find out how it needs to be written and be consistent.

- OK, the text was changed and the hyphenation was removed. The supplement was also corrected.

7) page 17, line 6/7:"Usually the spring leads to higher concentration of ABL species than the autumn". Why?

- At most sites (and not only at high elevated sites) the CBL height is found to be higher in spring and summer than in autumn and winter. The correlation with the down welling solar radiation at the surface clearly explains the summer high ABL height and hence the summertime peaks. Some other authors found an anti-correlation with the surface pressure and the lower tropospheric stability and a correlation with the near surface wind speed and temperature. A cumulative effect of all these parameters leads to a usually higher CBL height in spring (Guo et al., 2016, Pal and Haeffelin, 2015). However, since I do not find a clear referenced explanation for the often observed difference in ABL height between spring and autumn, I prefer not to insert any further explanation in the manuscript. The sentence was however changed to: *"Usually the spring leads to higher aerosol species than the autumn probably bounded to higher ABL height."*

8) page 18, line 14/15: Please explain why absorption coefficient is "the best tracer for anthropogenic pollution and biomass burning and consequently of ABL influence.". Unclear to me.

- The GAW-recommended basic aerosol measurement program consists of the particle number concentration, the scattering and absorption coefficients. All three of these parameters are higher in ABL than in FT. As stipulated on page 18 (in originally submitted manuscript), the aerosol absorption coefficient (or black carbon (BC) concentration) is the best tracer for ABL influence among the three aerosol parameters discussed. This is because the main sources of BC (anthropogenic pollution due to combustion processes and biomass burning) are in the ABL but are scarce near the high altitude sites. Additionally, BC aerosol is not produced by any secondary processes. In contrast, the particle number concentration and, to a lesser extent, the scattering coefficient are also influenced by gas-to-particle conversion mechanisms such new particle formation and condensational growth, which are secondary processes depending on the ABL influence in a more complex way and also on other parameters such as the solar insolation, the temperature and other thermodynamic processes. In that sense and among the basic aerosol parameters measured at most stations, the absorption coefficient is the best tracer for ABL influence. § 4.2 was changed accordingly: *"The absorption coefficient is primarily due to the presence of black carbon emitted from combustion processes occurring mostly in the ABL and rarely near the high altitude stations; additionally, BC aerosol is not produced by any secondary processes. Among the aerosol*

*parameters studied here, the absorption coefficient is consequently the best tracer for anthropogenic pollution and biomass burning and consequently of ABL influence."*

9) page 19, line 8: "by the smoother pressure decrease". I don't understand that explanation. Please clarify.

- An anti-correlation between the slopes around the station and the number concentration can be explained by new particle formation that is enhanced if the pressure difference experienced during the upslope transport is not too large. The sentence was clarified: "*The greater correlation of slope with the number concentration rather than with the absorption coefficient can be explained both by the very scarce sources of black carbon in the near vicinity of most of the high altitude stations and by the smoother pressure decrease experienced by the precursors during their upslope transport along gentle slopes leading to more condensation processes and nucleation*."

10) page 20, line 1: "and at all altitudes" awkward phrase. Rephrase sentence.

- *This was just a mistake and was removed.*

11) page 19, line 11: "There are consequently few correlations between topography parameters and the diurnal cycles". This is an important finding that should be explained better in this section. Does this imply that investigators trying to discuss diurnal cycles at high altitude locations waste their time by trying to find any correlation with topography?
Please discuss this better.

See also the answers to the main comment on page 2 and 3 of this document. The study of the diurnal cycles at high altitude sites can really bring important results if specific cases are analyzed and compared. In this study, a statistical approach has to be used to obtain a reliable estimate of the diurnal cycle amplitude and leads consequently to weaker correlations. This paragraph was changed:
"*The aerosol diurnal cycles are influenced by numerous phenomena (see Sect. 4.1) leading to a non-trivial relationship with the ABL influence. If the study of the diurnal cycles can bring valuable results if specific cases are analyzed and compared, the statistical approach is less obvious due to the noise in the data (low aerosol concentration and whitening process), to the inter-annual variability of the meteorological processes and to cloud, precipitation and long-range advection involving a large day to day variability. There are consequently few statistical correlations between topography parameters and the diurnal cycles. The clearest correlation is the influence of the insolation on the aerosol diurnal cycles amplitudes. This dependence between the latitude and the aerosol concentration was already mentioned by Kleissl et al. (Kleissl et al., 2007) and is easily understandable, the convection and the new particle formation being directly dependent on the solar radiation intensity. The other correlations are found between some topography parameters (ABL-TopoIndex, hypso%, G8 and LocSlope) and the absorption coefficient, which is the best tracer for ABL influence among the aerosol parameters.*"

12) Figure 3 caption, line 5: "horizontal" should be "vertical" here, I think.

- Yes, it was changed

13) Figure 11, caption: "Calculations corresponding to the various domain sizes can be identified by the various flow paths lengths.". I don't understand how the calculations can be identified. Please clarify.

- The plotted colored lines have various lengths that are for example visible around SBO station. This section was deleted following suggestions by the second referee.

14) Figure 12 caption. "similar to Fig. 8". I don't see how this is similar to Fig. 8.

- You're right, it should be changed to Fig. 11. This section was deleted following suggestions by the second referee.

**3. Answers to referee #2 comments:**

The manuscript "The topography contribution to the influence of the atmospheric boundary layer at high altitude stations" by Collaud Coen and co-authors investigates the role of the local to regional topography on aerosol observations made at high altitude sites. They derive parameters that are supposed to reflect the average influence of the atmospheric boundary layer on each site and rank the sites by these parameters. A comparison with different observed aerosol parameters is presented and supposed to show the validity and usefulness of the approach. However, I see several major problems with the suggested approach comprising all aspects of the presented work: the methods used to derive topographic parameters, their selection for a final index, and the choice of aerosol parameters that should reflect ABL influence. Although the manuscript touches on an important question of atmospheric monitoring and could be valuable for future network planning, it cannot be published in the current form and has to undergo major revisions.

**Specific concerns**

1) The analysis is only focusing on the influence of thermally induced wind systems on the aerosol observations at high altitude stations. Other vertical lifting mechanisms like foehn, deep convection, and frontal passages are completely neglected, although they can be as important depending on location of the site and the season (e.g. tropical vs. high latitude stations, summer vs. winter). The relative contribution by other lifting mechanisms to local "ABL" events will vary strongly between sites (e.g. volcanic island in the subtropics (rare) vs. coastal range mountain in mid-latitude west wind drift (frequent)). The methods presented here need to consider these differences, for example by limiting the observed aerosol observations to cases where vertical lifting mechanisms other than thermally induced flow can be ruled out.

- First, the authors agree with the referee that convection and thermally induced wind systems are not the only mechanisms that bring polluted air masses to high altitudes. The other vertical lifting mechanisms described by the referee contribute to indeed enhance the pollutant concentrations at high altitudes up to the free troposphere and it is also correct that these effects will vary depending on site, season, latitude, etc. However, as we explained in the introduction of the manuscript, we restricted this study solely to the influence of the topography on the thermally induced wind systems and the CBL growth. This study considers neither the dynamics of the atmosphere nor the soil properties. Such detailed and specific analysis is best left to the scientists responsible for the individual stations but is too complex when evaluating multiple sites with disparate data sets. To take into account the atmosphere dynamics, 3D models (and not only a 2D model of the earth surface) are necessary, which is clearly not the goal of this study and definitely outside its scope. Due to computational constraints, most current global models doesn't do a good job of representing the actual topography, the model grid spacing tends to be too large (on order of 1-2 degrees of latitude and longitude) most global models provide low frequency output – typically monthly

(although sometime daily). This means that targeted regional models would need to be used to describe each of the 46 sites here, again – this is a topic best left to the local experts responsible for each observatory.

- Second, our approach is to do a global and statistical analysis to understand the role of the topography in the ABL influence across an array of 46 mountain sites. Our hope was to begin to develop common rules that can be applied to all stations. It was never meant to analyze specific cases for clear thermally driven transport at individual stations. Doing so would also greatly reduce the usable time series and result in statistically small data sets. As stated in the introduction, there is presently no single method to screen ABL-influenced from FT air masses at high altitude sites. It is therefore quite difficult to sort the cases where the ABL influence is only due to thermally driven transport. Even if possible for all types of environment, the limitation of the aerosol dataset to cases where vertical lifting mechanisms other than thermally induced flow can be ruled out would need further complex data sets for each station (for example: pressure, humidity, wind measurements at each side of the stations, 3D back-trajectories, synoptic classification scheme and probably some gaseous species concentrations).

- Finally, Zellweger et al. (2003) concluded that, in contrast to the NOy mixing ratio, the major process for upward transport of aerosol is the thermally induced vertical transport. The choice of aerosol parameters to validate the ABL-TopoIndex can therefore be considered as the best one to study the thermally driven air mass transport.

- For all these reasons, the authors consider that the inclusion of the atmosphere dynamics and of the wind systems is beyond the scope of this study.

2) Furthermore, the method completely neglects the role of local to regional emissions. Emissions within the region of interest will be very different for the various sites and they will largely determine the amplitude of "ABL" events observed at the sites and also influence the larger scale tropospheric background. At least qualitatively emissions need to be considered and there is no lack of fairly high resolved, global emission inventories (e.g. for BC).

- The authors agree that the regional emission sources have an influence on the pollutant concentration measured at high altitude sites. However, the timing and relative magnitude of temporal cycles (as determined by auto-correlation) with ABL influence does not depend on the pollutant concentration in the ABL. To use the emission inventories, the atmosphere dynamic and particularly the wind components should also be taken into account in order to assess which sources on the 500 km x 500 km influence the high altitude sites (see answer to previous referee comment). Moreover, while the absorption coefficient could perhaps be "normalized" by the BC emission inventories, the scattering coefficient and the number concentration depend also on gas to particle conversion (e.g., new particle formation and condensation). The modeling of the gas-to-particle conversion from the emissions inventories and meteorological data is however rather complex. Moreover, the highest aerosol concentrations at the high altitude sites often depend much more on long range transport of mineral dust or biomass burning than on the regional sources. The authors are therefore of the opinion that, first, these large uncertainties would annihilate the potential benefits of the inclusion of the emission inventories and, second, that the inclusion of the atmosphere dynamics is beyond the scope of the paper. Additionally the amplitude of the diurnal cycle which is discussed in section 3.5 should be independent of the regional sources; this is therefore another way to "normalize" the aerosol concentration without reference to emissions information.

3) A similar problem is the selection of the observed aerosol parameters. Absolute aerosol parameters will depend on more factors than just the local to regional ABL input and are therefore not useful to access the question of FT vs. ABL influenced air mass. It would be more promising to

identify pollution or "ABL" events in each data series and correlate the frequency of these with any set of topographic parameters. Why would the 5th percentile of the absorption coefficient be a good indicator of ABL influence? The 5th percentile only reflects the lowest concentrations and not the frequency of pollution. Looking at the skewness of the distribution could be another indicator. Larger skewness would also indicate more frequent pollution events.

- To our knowledge the only method to detect local CBL development as well as the top of the aerosol layer is to use a ceilometer (or a lidar). There are however very few high altitude station around the world with a ceilometer time series from a lower altitude adjacent station thus limiting any statistical analysis.

- The $5^{th}$ percentile clearly reflects the lowest concentrations and therefore the ability to sample clean FT air masses at the high altitude stations. The lower the ABL influence is (through the CBL and the aerosol layer heights), the lower the $5^{th}$ percentile will be. The authors do agree that the median of the aerosol parameters is much more dependent on regional and local sources, whereas the 95% depends on rare high aerosol concentrations probably due to long-range transport of mineral dust or biomass burning. A normalization of the aerosol parameter with the 95% has consequently also not much sense.

- The aerosol parameters discussed in the manuscript (number concentration, absorption and scattering coefficient) are approximately lognormally distributed variables. The skewness toward the lower values is therefore not defined. The skewness toward the higher values reflects the occurrences of very high aerosol concentration that generally relates to long range transport of mineral dust and biomass burning. The skewness is consequently not the right parameter to detect ABL-influence.

- Apart from the $5^{th}$ percentile, the best parameters are clearly the diurnal and seasonal cycles. These are however much more difficult to statistically extract from the time series (see the answers to the main comments on p. 1 in this document) and exhibit few correlations with the topography parameters.

4) The selection and methods to derive the topographic parameters seem to be very arbitrary and no methodological way was followed to present a set of parameters that explains the observed inter-site variability. The final results seem to suggest that mainly one of the parameters is able to predict this variability (hypso%) showing even higher correlation coefficients than the final combined topographic parameter. It also remains unclear why a region as large as 750 km times 750 km was chosen for the analysis. Clearly the flow during one diurnal cycle (and that's what a thermally induced flow system spans) cannot advect air masses from a location as distant as 325 km. Assume an average advection velocity of 5 m/s, which is already a fair value for the kind of fair-weather, low pressure gradient situation required for thermally induced flow, then it would take 18 hours to cover the 325 km. Also plain to mountain winds are known not to extend from the mountains by more than around 100 km. Hence, the use of a smaller region or the use of several sets of parameters for smaller regions should have been considered. These larger sets of topographic parameters and/or any combination of them could than have been fed into a statistical model of the observed aerosol parameters using parameter selection techniques to derive the most important topographic parameters.

- The methodology applied in this study consists first in identifying topographical criteria that would tend to increase the ABL influence and then finding parameters that can be quantitatively estimated and related to the topographic criteria. The authors do agree that some choices were not sufficiently motivated in the first version of the manuscript so that now both the used parameters (section 2.3: ABL-TopoIndex) and the rejected parameters (Table S2) are now better described. (see also the answer to the specific comment "p.6 L12" in this document)

- The reasons to choose a large size of the domain are given in p. 1 of this document (answers to the main comments) and also now discussed in the revised paper. The authors also have now restricted the size of the domain to 500km x 500km. This restriction has a very low impact on the results. : "*A quantitative estimation of these criteria depends clearly on the domain considered. The minimal size requirement for such a topographical analysis is that the domain should contain the whole mountainous massif. An airborne Lidar measurement of the ABL over the Alps (Nyeki et al., 2002) clearly stated that the convective boundary layer is formed over a large-scale and leads to an elevated and extended layer. It also quantifies this "large-scale" to extend more than 200 km from the mountainous massif. A rectangular domain size of 500 km x 500 km centered on each site was then chosen (see § 3.2 for a discussion of the effect of the domain size).*"

- The third specific concern of the referee clearly supports our contention that there are no parameters that can act as an indubitable sign of ABL influence. The best statistical parameter would be the annual cycle of the diurnal cycle amplitude which should be the greatest for stations sampling the FT part of the year. It was however not statistically possible to extract this parameter from the available time series for the following two reasons: 1) a lot of the time series were too short (< 2-5 years), as explained in the answers to the main comments (p. 2 of this document), and 2) the low aerosol concentration measured at high altitude combined with the pre-whitening process lead to a large uncertainty in the statistical determination of the cycle amplitude. In that sense, there is, to our knowledge, no reference measurement that would definitively identify the ABL influence and allow selection through a statistical model the most important topographic parameters.

- Apart from the used and the rejected parameters (now more clearly described in the revised manuscript), the authors do not see any other "direct" parameters that can be possibly used. There are other more sophisticated parameters that are linked to the valley's topography that could be added to the ABL-TopoIndex in a further study (see § 4.3 describing possible future work), but the authors found it necessary to validate the present study by a publication before investing further time in exploring more complicated parameters.

5) This continues from 4 but deserves its own point. The analogy between water flowing down a mountain and thermally induced flows rising up a mountain, which is used to derive the parameter DBinv and is used in the discussion of section 3.6, is not valid. It is simply not correct to assume that a large air catchment will result in large upward flow at the highest point of a mountain massif. Air does not flow up to the highest point as water flows down to the lowest point. The upward flow on a fairweather day with small pressure gradients happens along individual slopes all along individual valleys and results in many convergence lines but not a single convergence point as suggested here. The presented parameter probably has some value on the very local scale but may just be very similar to hypso% in the end. This parameter and its justification as well as the whole discussion of flow paths will need to be removed from the manuscript. It simply does not reflect the ongoing physics of thermally induced flow systems correctly.

- The authors do agree with the referee that the analogy between water flowing down and thermally driven air flow has very well defined limitations. In that sense we have removed the whole discussion about flow paths (§ 3.6 and figures 11 and 12) that involve a direct analogy between the water and the air mass flow paths.

- The DBinv used in the ABL-TopoIndex has a completely different motivation and impact. DBinv is a quantitative parameter for the size of the reservoir for air convection (criterion number 4). The authors do agree that upward flows do not result in a single convergence point at the station. However DBinv is a measure of the territory that can directly influence the station air masses by upslope winds. It is true that the considered domain represented by DBinv is too large to represent the direct influence of the CBL at the station, but it is of reasonable size to describe the influence of the aerosol layer (AL) (or residual layer (RL) during the night). It was clearly shown that the AL (or RL) have a clear impact on the aerosol

concentration at high altitude stations (Collaud Coen et al., 2011, Poltera et al., 2017, Andrews et al., 2011 and references therein). Due to these reasons and to the influence of DBinv on the correlation of the ABL-TopoIndex with the aerosol parameters, the authors have chosen to keep DBinv in the ABL-TopoIndex definition.

**Specific comments**

Abstract: Clarify what is the scientific question at hand and what is your contribution to this problem. For example starting from line 21, start the sentence with something like "Here we ..."

- The abstract was modified and the following sentence was added at line 21: "*In this study, a topography analysis is performed allowing calculation of a newly defined index called ABL-TopoIndex. The ABL-TopoIndex is constructed in order to correlate with the ABL influence at the high altitude stations and long-term aerosol time series are used to assess its validity.*"

Page 8: How comparable are the aerosol parameters between sites? Besides the detection limit adjustment what kind of common quality assurance, quality control was applied to assure that these parameters can really be used for a ranking between sites.

- 23 of the 28 aerosol datasets are provided by GAW stations and the data were obtained from the EBAS data center. GAW stations have to follow the measuring rules and quality assessment edited by the WMO/GAW aerosol advisory board. These measurement principles are extensively described in GAW report Nr 200 (WMO/GAW standard operating procedures for in-situ measurements of aerosol number concentration, light scattering and light absorption). As required by the GAW aerosol advisory board, all measurements were performed at low humidity (RH<40%). Moreover the data owners also follow the quality control procedures of the EBAS data center. Four of the datasets (MUK, NWR, PEV and OMP) are not GAW stations but the measurements were performed by research groups operating at other GAW stations. Individual exchanges with the data providers from those four sites indicated that they collected those datasets using methods similar to their operations at GAW stations so that the quality and traceability are assured. The GAW stations are now given in bold in Table S3. The umbrella provided by the WMO/GAW program is, to our point of view, sufficient so that a further description of the quality assurance of the aerosol measurements is not needed in this paper.
- Other procedures such as the STP correction, the truncation correction of Nephelometer data, the negative data of the absorption coefficient were controlled and handled similarly for all datasets. Small time series breakpoints are not important since no trends were calculated.
- All the time series were visually inspected and any doubtful data were removed after discussions with data providers.
- All the times series but 2 were done on TSP or PM10 inlets, so that similar aerosol size distributions were measured.

P2,L34: The whole terminology is confusing "flow paths for air convection". Convection does not happen along flow paths. Convection is a vertical transport and mixing mechanism at small scales and as such defined as mostly unorganised. See: http://glossary.ametsoc.org/wiki/Convection. Why not talk about "thermally induced flow paths" instead.

- The authors agree that "thermally induced flow paths" is a much better terminology. Since § 3.6 on "Flow paths as a function of ABL heights" was removed, the expression "flow paths for air convection" no longer appears in the manuscript.

P3,L20: Commercial airline programs such as IAGOS CARIBIC (http://www.caribicatmospheric. com/) would be worth mentioning in this context as well.

- The following text was added to the manuscript: "*Instrumented airplanes can make detailed measurements of the vertical and spatial distribution of atmospheric constituents and are used either during limited measurement campaigns or on regular civil aircraft (see for example the IAGOS CARIBIC project), but, because of the limited temporal scope of most measurement campaigns, cannot provide long-term, continuous context for the measurements.*"

P4,L26: Mention that this is the picture for a continental ABL not for a marine ABL.

- Ok , done (@P3 L26): "*. In the case of fair-weather days, the continental ABL has a well-defined structure and diurnal cycle leading to the development of a Convective Boundary Layer (CBL), also called a mixing or mixed layer, during the day and a Stable Boundary Layer (SBL) which is capped by a Residual Layer (RL) during the night (Stull, 1988).*"

P4,L29: This is not necessarily correct. In regions with emissions the nighttime accumulation of the emitted species in the shallow SBL usually leads to nighttime concentration maximum of these species.

- As noted by the referee, it is completely correct that the emitted species accumulate during nighttime in the SBL, leading in some cases to high concentrations. This was now specified in the text at P3 L30.
  "*During daytime, the aerosol concentration is maximum in the CBL and remains high in the RL. During nighttime, the surface-emitted species accumulate in the SBL.*"

P4,L17: The authors should mention other vertical lifting processes. Generally frontal lifting (synoptic systems), deep convection and, in mountainous terrain, foehn. The importance of these processes was nicely illustrated by Zellweger et al. (2003).

- The text was changed to "*Finally, ABL air masses can also be dynamically lifted by frontal systems, deep convections or foehn as well as be advected from mesoscale or wider regions and influence high altitude measurements by all these atmospheric processes.*"

p4,L25: Zellweger et al. (2002) not in list of references. Probably meant Zellweger et al. 2003, but that does not include a discussion on CO2. Please correct.

- This was indeed incorrect and was changed: "*Many methods have been used to separate FT from ABL influenced measurements, including those based on time of day and time of year approach (Baltensperger et al., 1997; Gallagher et al., 2011), wind sectors (Bodhaine et al., 1980), the vertical component of the wind (García et al., 2014), wind variability (Rose et al., 2016), NOx/NOy, NOy/CO ratios or radon concentrations (Griffiths et al., 2014; Herrmann et al., 2015a, 2015b; Zellweger et al., 2003) and water vapor concentrations (Ambrose et al., 2011; Obrist et al., 2008), although none of these methods leads to an absolute screening procedure to ensure the measurement of pure FT atmosphere.*"

p4,L34 to p4,L2: Here it is stated that there are other important influence factors other than thermally induced flow. But it is not explained why one should be able to neglect them. See major remark 1.

- Please see our response to main comments p.1 of this document. Further, to our knowledge, most of the meteorological models are not able to solve all the dynamic processes in

complex topography. This study therefore concentrates on one question and tries to identify some relations between the topography and the thermally induced ABL influence. This restricted, but nevertheless ambitious objective (as well as the factors that are not taken into account), are clearly specified in the manuscript.

p5,L3: The term topographic index or topographic wetness index is already defined in hydrology (the authors used it as well). Therefore, the choice of this name for the parameter introduced here might be confusing, especially since some hydrological methods are applied to derive part of this parameter. Maybe just use ABL-index instead.

- P6L3: The referee is, of course, right in saying that the terminology of "topography" and "index" are already widely used in several scientific domains. The use of only "ABL-Index" however seems too vague to the authors, since it does not specify that only the effects of the topography are taken into account. For example, an ABL-Index could represent any number of ways of assessing ABL influence. The authors chose therefore a name (ABL-TopoIndex) where the three main underlying concepts explored in the manuscript are cited. The word "topography" was also abbreviated in order to minimize possible confusion with existing hydrological terms. Moreover the manuscript was carefully checked so that the word "TopoIndex" was never used alone. The authors prefer to keep it as written.

p5,L7: Unclear what is mend here by lakes. Again a wrong picture is drawn that suggests that there is a certain amount of air that can be transported by thermally induced flow systems. Lakes or cold air pools are more a phenomenon of the nighttime SBL but not an established concept for daytime flow.

- The authors do agree that, even if used in quotation marks, the word "lakes" is misleading. It is now replaced by the expression "*air mass reservoirs*" in the revised version.

p5,L19: Why was the relatively coarse dataset GTopo30 used? There are global DEMs with higher resolution. 1 km seems a bit coarse for the kind of sites in extremely steep terrain targeted in this study. Some of the local topography will be missed. In this context it would be interesting to see how the height of GTopo30 at the station locations actually compares to the real altitudes. I would encourage the authors to have a look at a higher resolution DEM like https://asterweb.jpl.nasa.gov/gdem.asp for any further analysis.

- The authors thank the reviewer for giving the suggestions of another high resolution DEM that they will use in case of further studies. A higher resolution model will clearly be of interest. It has however to be noted that the ABL influence is not really a very local phenomena so that the mean over 9 grid cells was used to obtain the ABL-TopoIndex. As expected the GTopo30 altitude at the station grid cell differs by more than 20% for 3 of 28 stations used for the correlation analysis, the GTopo30 altitude being always lower than the station altitude. These differences do not correspond to the real altitude difference between the real and the GTopo30 mean altitude over each grid cell. Corresponding to the methodology applied to the ABL-TopoIndex, the correlations were also done with the mean altitude of the 9 grid cells. It has however to be noted that the use of the station altitude or of the 9 grid cells mean altitude does not change the correlation results. The GTopo30 manual gives a minimal vertical accuracy of 250 m at 90% confidence level and a RMSE of 152 m (the Peru map, which has a lower accuracy (see GTOPO30 manual), is not used). The altitude of the grid cell containing the station as well as the mean altitude of the 9 grid cells used to calculate the ABL-TopoIndex are now given for all stations in Table S1 with the following comments: "*The real altitude of the station, the mean altitude of the grid cell containing the station and the mean altitude of the grid cell containing the station and of its 8 adjacent grid cells are given in Table S1, the last 2 altitudes are calculated from the DEM after its projection in UTM coordinates. Since the stations are usually at high altitude, the*

*altitude of the DEM grid cell is usually lower than the station altitude. The mean and median of the differences between the station altitude and the one of the grid cell are 190 m (8.6%) and 140 m (5.8%), whereas the mean and median of the differences between the station altitude and the one of the 9 grid cells are 270 m (11.7%) and 220 m (10.3%), respectively. The maximal altitude differences is found for SZZ (1153 m) that corresponds to 3% of the station altitude. Due to its peculiar situation (see paper), NCOS altitude is 1110 m lower that its DEM grid cell altitude (2.8%) and this can perhaps explain NCOS outlier status. ZEP is only 306 higher than its grid cell altitude, but this corresponds to 65% of its altitude and also explain its very high ABL-TopoIndex and its outlier status. It has however to be noted that The GTopo30 manual gives a minimal vertical accuracy of 250 m at 90% confidence level and a RMSE of 152 m (the Peru map being anyhow not used)."*

p6,L11: Very questionable that these parameters are quantitative

- The parameters described under 2.3 are quantitative parameters that can be calculated for each point of the earth using a DEM. In that sense we think that the adjective "quantitative" is not misleading.

p6,L12 cont: Lots of arbitrary choices here. 750 km domain, median altitude vs. station altitude (could be any percentile; lower percentile would avoid negative values), slope between 1 and 10 km, 2-4 km mean gradients ... As mentioned above sets of parameters for different distances, etc. should have been derived and a statistical model with parameter selection been applied. It would also be nice to see all values for the calculated parameters as part of table 1.

- We agree that all the choices should be explained in this section and rather than later on in the paper:
  1) Concerning the size of the domain, please see the answers to the main comments on p.1 of this document. § 2.3 was also modified: "*A quantitative estimation of these criteria depends clearly on the domain considered. The minimal size requirement for such a topographical analysis is that the domain should contain the whole mountainous massif. An airborne Lidar measurement of the ABL over the Alps (Nyeki et al., 2002) clearly stated that the convective boundary layer is formed over a large-scale and leads to an elevated and extended layer. It also quantifies this "large-scale" to extend more than 200 km from the mountainous massif. A domain size of 500 km x 500 km centered on each site was then chosen (see § 3.2 for a discussion of the effect of the domain size).",*
  2) for the hypsD50, the referee is correct that any percentiles could be chosen. The median was taken first because it is a common averaging tool and second because presumably it would be lower than the location of each "high altitude station". The authors tried to summarize this more clearly in the manuscript by adding the following sentence: "*The median of the hypsometric curve was chosen first because a station claiming to be a high altitude site should typically be at higher altitude than half of its geographical environment.*" Moreover, the station with hypsD50 can be found in Table S1.
  3) LocSlope is defined on a radius of 10 km since the minimal distance between the station and the nearest plateau is usually equal to or larger than 10 km. This is now stated in the manuscript: "*The distance of 10 km to calculate the LocSlope was then chosen as representative of the maximal distance to the next adjacent plateau for almost all stations.*"
  4) the G8 is always calculated from one grid cell to the next, so that the distance of 2-4 km is given by GTopo30 and varies with latitude.

  Moreover all values for the calculated parameters are now in Table S4

P7,L9: Confusing wording and concept. Drainage is a nighttime process, convection a daytime process???

- Yes, drainage winds are a nighttime process, but the manuscript discusses a "drainage basin". Drainage basin is a hydrologic term without time connotation and can be used for daytime processes. As defined by the dictionary, "a drainage basin is the area drained by a river and all its tributaries". It is also called catchment area, drainage area, watershed or river basin.

p7,L25f: It is true that the geometric mean will change in the same way for any percentage change in any of its parameters. However, it does not normalise the variability in the parameters in the desired way. If parameter a has a 10 times larger relative variability than parameter b, the variability of the geometric mean will be dominated by a. If this is an issue in the current case could be easily tested by the authors by analysing the relationship of the original parameters and the derived geometric mean. Better than the geometric mean would be the use of parameters that were normalized for example by their variance.

- The referee is correct that the geometric mean reports similarly any percentage change in any included parameters whatever the absolute value of the parameter is. This is the reason to apply the geometric mean for environmental indices that are built with very different parameters. The use of other types of averaging with any kind of normalization does not allow us to obtain this necessary (for this analysis) mathematical property. A normalization with either the maximum or with the variance will change the value of the ABL-TopoIndex but not the ranking of the stations. Moreover the authors checked that none of the included parameters dominates the results. To further develop this critical technical point, the manuscript was changed: "*Further, a given percentage change in any of the parameters will yield an identical change in the calculated geometric mean value. In that sense the variability of each parameter is also normalized, leading to similar modifications of the ABL-TopoIndex for similar parameter's variations.*"

p8,L17ff: It should be mentioned again when presenting the results that the seasonal and diurnal cycle that is looked at is actually the auto-correlation function. As such the amplitudes of the cycles is already normalised, which helps for the inter-comparability between sites.

- Yes, it is a good idea to highlight this fact in the results section. The following sentence was therefore added to § 3.5: "*Both the diurnal and the seasonal cycles were calculated as the strength of the autocorrelation function (see § 2.4 and supplement) so that the underlying parameters are de facto normalized and that the cycles between the stations can be directly compared.*"

p9,L15f: These changes are rather large. Especially considering that the ranking between sites changes with domain size. It should be possible to solve the transformation problem in such a way that G8 and LocSlope are really constant with domain size. Why would the domain size change the local transformation/interpolation anyway? This needs to be redone.

- The authors looked again at the problem of non-constant values of LocSlope and G8 for various domain sizes. Both these values are constant in the traditional latitude longitude coordinates. The UTM projection leads to minor changes in the LocSlope and G8 that can be explained by two reasons: 1) if the analyzed domain extends beyond 2 UTM zones, map distortion problems occurs. This is, for example, the case for BEO plotted in cyan on Fig. 6 and having large G8 modification as a function of the domain size. 2) the interpolations needed to do the UTM projection can also lead to variation and G8 is very sensitive to these

variations. The UTM projection is however necessary to ensure a similar handling of stations at very different latitudes.

Section 4: The name of the section is misleading. The section does not present a ranking of the sites by TopoIndex but more a discussion along their geographic location.

- 3.4: The title was changed to "*Relation between the ABL-TopoIndex and the station location*"

p12,L5: The more correct name would be "Rocky Mountains".

- "Rockies" was changed "*Rocky Mountains*".

p12,L15f: Why was MWO not discussed in this context as well?

- It is right that MWO is the North America station with the lowest ABL-TopoIndex and needs some comments. The following text is now added: "*Mount Washington Observatory is located in the Presidential Range of the White Mountains. It is the highest peak in the Northeastern United States and the most prominent mountain east of the Mississipppi River. MWO is consequently the North American station with the lowest ABL-TopoIndex due to very low hypso% and relatively high G8 and low DBinv.*"

p13,L13f: Looks like the authors themselves are surprised that there is any relationship between their TopoIndex and the chosen aerosol parameters ...

- The authors just wanted to state that their hypothesis was verified. If wrong criteria or parameters (see § 2.3) had been chosen, the correlation with aerosol parameters would have shown it. The word "happily" is however inappropriate in a scientific context and is (sadly) removed in the revised version.

p13,L26f: But hypso% is an even better predictor than TopoIndex. I guess that means that all other parameters only partly destroy this relationship but do not add any useful information. Especially the suspicious parameter based on water flow analogy, DBinv, seems to show very bad predictive skills (worse than altitude alone in some cases).

- As explained at the beginning of this document, the various modifications required by the referee's (smaller domain size, inclusion of the middle atltitude stations) as well as the removing of SUM time series from the correlation analysis lead to a somewhat different values of the Spearman rank correlation coefficients, even if the statistical significances remain similar for most of the case. In case of the correlation with the absorption coefficient, the importance of hypso% with regard to the other parameters constituting the ABL-TopoIndex decreases. LocSlope and G8 are now equally important parameters, whereas hypsoD50 has usually a lower statistical significance. We also checked that the statistically significance of the correlation between the ABL-TopoIndex and the aerosol cycles is clearly decreased if DBinv is removed from the ABL-TopoIndex definition. This is effectively the case, even if DBinv has globally bad predictive skills. Sections 3.5 and 4.2 were consequently modified.

p14,L18: Wasn't the point in Bianchi et al that the ABL influence is not a direct one, like you focus on here, but an indirect one of ABL air picked up a few days before arriving at the measurement site and therefore not being lifted by thermally induced flow but by convection or frontal systems.

- Thank you for this comment. It is correct that the greater ABL influence due to longer daytimes and stronger insolation does not relate to Bianchi et al., 2016. At this point, the

authors just wanted to mention that stronger insolation usually also promotes NPF formation. The manuscript was modified consequently: *"The high correlation between the maximal diurnal cycle and the number concentration can also be explained by the promotion of NPF by the stronger insolation at low latitude."*

p14,L30: Isn't the failure of the ABL-TopoIndex to identify these lower altitude sites a clear indication that the suggested method does not work at all? Otherwise these clear cases of larger ABL influence should be detected and the correlation should actually improve.

- De facto, the concept of the ABL-TopoIndex is really developed for high altitude stations with complex topography and cannot be applied to low altitude sites. NCOS was already identified as an outlier in the first version of the manuscript, and we found during the revision of the manuscript that SUM should also be removed from the correlation analysis because it is located on a high altitude plateau with a very smooth relief due to the ice sheet formation.
- The aerosol parameters used for the correlation analysis are also chosen to reflect the ABL influence at stations that are at least occasionally located in the FT. The causes of the aerosol concentration minima and maxim as well as the diurnal and seasonal cycles are completely different for sites that remain in the ABL during the whole day. In that sense, neglecting stations situated at too low altitudes (like ZEP) is absolutely reasonable. In our study, HPB and MSY, two middle altitude stations, decrease the correlation coefficient values without destroying the correlation. They are now included into the correlation analysis and the related section (motly section 3.5 and 4.2) were modified.

p15,L29: All of a sudden back-trajectories appear. It seems clear that these are not the hydrological flow paths. But from which model do these trajectories come from and why were they not used for all sites to also characterise the thermal flow systems (even if not fully represented in the model).

- Back-trajectories were calculated by the CHC data owners and used in other studies. They are used in this study just as a comparison with the main flow paths as a function of the ABL altitude. Anyhow, the section 3.6 was removed in the revised manuscript as recommended by the second referee, so that this point does not need a more detailed discussion.

p16,L16-17: This argument is going round in circles. The absorption coefficient is supposed to be an indicator of ABL influence because it correlates with topoIndex. But I though it needs to be shown that the topoIndex actually represents ABL influence ... Very confusing.

- This sentence is actually mixing some statements from both the results and discussion sections. It was therefore modified: *"Our results showed that of the three aerosol parameters tested in this study (number concentration, absorption coefficient and scattering coefficient), absorption coefficient has the greatest correlation with the ABL-TopoIndex values."*

p16,L26: NO3 being NO3_aq or ions?

- This correspond to particulate nitrate ( $NO_3^-$) (Zellweger et al., 2003) and this is included in the revised version of the manuscript.

p19,L17f: These parameters are mostly know to the hydrological community but need additional introduction for the more atmospheric readership of the current journal. As mentioned before, it would have been better to provide such parameters to a statistical model with parameter selection in order to get an objective selection of parameters that may explain ABL influence. However, most these parameters would also follow the misleading assumption that thermally induced flow works just opposite to water flowing downhill and, therefore, should possibly not be considered at all.

- The authors did not consider at all that thermally induced flow can be considered as the opposite of water flow and most of these parameters were actually not used because of such discrepancies. However, as explained in the answers to the main comments (p.1 of this document), these parameters and the reasons for their rejection are now detailed as a table in the supplement (see Table S2 on p. 3)

Table1: Add the GTopo30 altitude of the grid cell containing each site, along with all the parameters derived for the site (potentially as supplement).

- The GTopo30 altitude of the grid cell as well as the mean for the 9 considered grid cells were added in the supplement with some comments. The altitude of the DEM grid cell as well as the mean altitude on the 9 used grid cells are given in the supplement Table S1.

Table2: The units for LocSlope should be m m-1 not Mm-1.

- Thanks for catching this! LocSlope has no units but there is a factor of $10^{-3}$ because the altitude is given in m and the horizontal distance in km. The values and units in Tab. 2 are corrected in the revised version.

Figure1: The figure quality is not state of the art. I suggest to use a topographic image as background. Larger station labels or symbols. Legend for mountain ranges.

- You will find thereafter Fig. 1 similar to the first version but with the right color scheme and a second version with the continental topography beyond the station location. If the first version allows to clearly visualize all stations, the second version also gives some information about the highest massifs around the world. The authors put the second version in the revised manuscript, but let the editor chose which figure should be finally used in the manuscript.

[Figure]

[Figure]

Figure2: The schematic is confusing. If you want to underline that there is a higher ABL influence on the right, why not show a visible, partially terrain following ABL in the mountainous area and an aerosol layer resulting from lift over processes. The schematic on the left is a very poor image of a mountain shape. Looks more like a life buoy with a signal post but not like the profile of a volcano.

-The referee is right, the schematic view was somewhat crude. The left schema is now changed. Since section 3.6 was deleted following the referee's comments, the added ABL was removed from Figure 2.

Figure4: The thick cyan line is not mentioned in the caption.
- OK, this now mentioned in the figure caption: "*The main flow paths from the station grid cell are given by the cyan lines.*"

Figure6: Sub-panel labels are missing in the figure but are used in the caption.

- OK, the sub-panel labels are now written in the figure.

Figure8: What are the different shades of colours? Neither explained in caption nor text.

- Some colors were changed in both Fig. 1 and 8 so that the color scheme of both figures are now similar. This is now mentioned in the figure caption of Fig. 8*: "The color scheme corresponds to that in Fig. 1."*

Figure9: Very difficult to comprehend. Too many colours and symbols in one plot. Why not display negative correlation coefficients as such on the negative part of the y axis. Instead of circles, different sized symbols should be used for different significance levels.

- As suggested by the referee, the statistical significance is now given by different symbol sizes and this clearly increases the readability of the figure. We keep however the negative correlation as downward triangles to keep the direct comparison between the absolute value of the correlation coefficients. Since the anti-correlated topography

parameters are used as 1/parameter in the ABL-TopoIndex, the absolute correlation value is more important that its sign.

   - Our first answer was somewhat imprecise on this subject. The authors do agree that the amplitude of the diurnal cycle will be influenced by the aerosol concentration emitted into the ABL and consequently by the emission sources in the case of a high altitude site situated part of the time in the FT. In case of a large influence of the residual layer during the night, the dependence of the diurnal cycle amplitude on the emission sources will be largely decreased or even suppressed. The timing of the diurnal cycle will however not be affected by local pollutant sources.
   - The reason why ZEP, SUM and NCOS were considered as outliers is not at all due to the low aerosol concentration or the low emissions near the station. As explained in the paper (see 3.5 first §) the topographical environments of SUM and NCOS (high plains that cannot be considered as a complex topography) lead to very high ABL-TopoIndex that are considered as outliers. The exclusion of ZEP is due to its very bad representation in GTOPO30: ZEP is only 306 higher than its grid cell altitude, but this corresponds to 65% of its altitude and also explains its very high ABL-TopoIndex and its outlier status.
   - The referee completely identifies the problem of the pollutant sources when he says that the source inventory should include a factor depending on the altitude and on the horizontal distance. These altitude and horizontal distance factors would however also depend on the time of the year and/or the surface temperature in order to take into

account the seasonal variability of the ABL height. For example in the alps, wood heating leads to very high aerosol concentration during winter when the ABL influence to high altitude sites are so low that they can be neglected. This is a great subject for a future study. The authors have consequently developed the part of the discussion about the emission problem on p. 16 :

> "*Because there are variable pollution levels in the vicinity of the stations, a single absolute value of a pollutant cannot be used to evaluate the ABL-TopoIndex (or ABL influence in general) when considering multiple high altitude stations. An inventory of the proximate pollution sources bounded to a 3D thermodynamic model adapted to complex topographies would be required before using absolute pollutant concentrations as indicators of ABL influence at high altitude sites. Another possibility would be to weight the pollution source inventories by factors depending on their vertical and horizontal distances to the high altitude stations, as well as on a seasonal parameter representing the potential ABL height. A further use of DBinv to restrict the area of potential pollution sources could also be envisaged since this parameter describes the domain from which pollutants can reach the high altitude station by convection and without crossing topographical barriers. This delicate issue can however be avoided by instead considering dynamical parameters such as the various temporal cycles.* "

3) **ABL events**:
   - **Skewness**: The aerosol parameters used here have a distribution that is neither normal nor lognormal but a Johnson distribution, even if it can be considered as lognormal for some applications (but not all).  The skewness is also defined for lognormally distributed data. The skewness of the data and of the logarithm of the data were calculated for the JFJ station for which data of different instruments measuring the same quantity  are available. For all aerosol parameters and all instruments, the skewness of the logarithm of each parameter is always negative, leading to the conclusion that the low concentration events are more frequent than the high concentration events. The data skewness (without taking the logarithm) clearly depends on the instrument and on the wavelength. For example, the absorption coefficient is measured at the JFJ by 3 different instruments, two Aethalometers (AE31 and AE33) measuring at 7 wavelengths and the MAAP at 637 nm. The absorption coefficient skewness at similar wavelengths (between 630 and 660 nm) varies from  -7.9 to 22.8 among the three instruments. For the Aethalometers , the skewness varies between 5.1 to 6.7 (AE31) and from 9.8 to -7.9 (AE33) as a function of the wavelength. In conclusion, the skewness results depend mostly on the resolution (and artifact) of the instruments and cannot be used to make any assessments about the frequency of pollution events.
   - **The diurnal cycles** are not averaged over a whole year (see §2.4 and supplement 1), but for each month. This allows us to find the minimal and maximal amplitude of the diurnal cycles over the entire year. These minima and maxima of the amplitude of the diurnal cycles are used in Fig. 9, as well as the amplitude of the seasonal cycle. The seasonal variation of the energy available for convection is consequently taken into account and is not the reason for the difficulties in statistically extracting the cycles from the time series. This was not so clearly described in the previous version of the results section so that the following sentence was added to p. 15:

> *"The diurnal cycles were calculated for the 12 months, so that Dmin and Dmax correspond to the lowest and the greatest monthly amplitudes, respectively. "*

- The authors do agree that the difficulties in evaluating the diurnal cycle are not fully described in the discussion so that they add further comments in § 4.1:

> *"However, the statistical determination of the diurnal and seasonal cycle amplitudes suffer from several difficulties: 1) the low aerosol concentration at high altitude often results in measurements near the detection limit leading to high uncertainties, 2) the high hourly autocorrelation of the data require a pre-whitening procedure (see supplement) in order to be able to detect the diurnal and seasonal cycle, 3) meteorological conditions (e.g., cloud coverage, precipitation or seasonal fluctuations) modify the theoretical cycles and lead to a broadening of the time of the extrema. These difficulties make obtaining clear statistical cycles another reason contributing to the observed low correlations."*

4) **Topographic parameters and their selection**:

- **DBinv**: The referee is still not convinced that DBinv is a necessary parameter of the ABL-TopoIndex. The authors allow themselves to argue once again in favor of DBinv:

   a. One of the referee's arguments is that "there is no reason what so-ever why a hydrological divide should also hinder atmospheric flow" and the authors do agree with this sentence. The main idea to introduce the DBinv is to determine from where air masses can reach the high altitude station only due to convection, that is without crossing topographical barriers. In that sense, DBinv is a kind of "reservoir" not for air convection (that does not exist, as explicitly mentioned by the referee), but for pollutants that could reach the high altitude sites during long fair-weather periods without precipitation. Let's take the unrealistic example of a summit encircled by a continuous ridge at higher altitude than this isolated summit. The potential influence of ground emitted pollutants will be lower than if the ridge is not continuous but is lowered at several places by valleys. DBinv will take this difference into account much better than Hypso% or hypsoD50.

   b. As explained in the discussion (§ 4.3), the present study uses only aerosol measurements to validate the ABL-TopoIndex. The authors however expect that the ABL-TopoIndex will be further validated with gaseous species or radon measurements as well as used with other studies involving, for example, new particle formation. DBinv presents an only marginal correlation with the aerosol parameters and cycles, but could be an important parameter in other contexts, so that the authors still wish to include DBinv as a ABL-TopoIndex parameter. Moreover, the use of DBinv does not decrease the significance of the correlation between the ABL-TopoIndex and the aerosol parameters. In summary, DBinv has a potential for explaining other variables and does not disturb the present analysis.

   c. The very accurate remarks of the referee concerning the use of source inventories clearly assess the difficulties of using these data in a meaningful way. The authors think that DBinv could be used to define the domain whose sources will contribute to the pollutant concentrations at the high altitude sites. This problem could be solved either by using NWP models to calculate a probability

density for a region to affect high altitude sites with the disadvantage of using a lot of computer time to obtain a wind climatology for each site and the not always accurate topography. One could also consider only all the sources in the DBinv or, as suggested by the referee, use DBinv as a probability weight for sources to influence the high altitude site, taking into account that transport through advection is not considered.

In order to replace the misleading/incorrect description of CBinv as "a reservoir for air convection" and to better explain this concept, its description was modified and it is now referred in the paper as a "drainage basin for thermally lifted pollutants":

> *"**Parameter 5 – DBinv:** Since the air masses have to be thermally lifted from the valleys and plains towards the summit to influence the station measurements, the size of the drainage basin (DBinv) for thermally lifted pollutants can be calculated with standard hydrology tools using an inverse topography, where the altitude Z is changed to –Z allowing the summit to become a hole. It represents the region from which pollutants such as aerosols can be thermally lifted without crossing any topographical barrier. Figures 4d and 5d are examples of the DBinv calculation for BEO and PYR. The DBinv is related to criterion 4. The ABL influence should increase with increasing size of the convection drainage basin."*

The authors hope that these modifications associated with the developed arguments will convince the referee as well as the editor to accept that DBinv should be kept in the determination of the ABL-TopoIndex. The authors remain however open to further discussion on this particular point.

- **The PYR example** : The phenomena described by the referee corresponds to an advection, which is not taken into account in our paper dealing with convection. The authors do agree that the convection from low altitude places in the DBinv for BEO is not a usual phenomenon, but it still corresponds to a region from which pollution is more likely to influence the high altitude station, particularly in case of long summer periods with fair weather days when the pollutants stay in the atmosphere for a long time.

- **Latitude and altitude**: The referee suggested to eventually add the latitude and the altitude as ABL-TopoIndex parameters. One aim of this study if to show that the altitude is not the right parameter to quantify the ABL influence, so that the altitude has not been included into the ABL-TopoIndex. However the correlation with the altitude was also calculated and results in poor correlations. The latitude is really a predictive parameter, but it does not relate to the topography of the high altitude stations. In that sense, it has to be considered as another independent parameter.

- **G8 and LocSlope:** The authors do agree with the referee that G8 and LocSlope are the two parameters having the greatest correlation since the morphology of mountainous ranges are usually similar at several scale ranges. G8 describes however the slope steepness in the direct vicinity (about 1km or less depending on the latitude) of the station whereas LocSLope characterizes the steepness in a 10 km range. The authors find that it is worth analyzing the steepness locally as well as at a regional scale.

- **Regression model** : First, the authors are unaware of a non-parametric regression model allowing to estimate the better proxy variables to explain, for example, the aerosol measurements. Regression models are constructed for normally distributed data or can be applied on the logarithm of the data. In our case, the aerosol parameters have a Johnson distribution that can be approximated by a lognormal distribution in some cases, but this approximation can also lead to clear errors. Second, the authors hope that the ABL-TopoIndex will be tested with other measured parameters and not only with aerosol optical properties and number concentration. For example, new particle formation, radon and gaseous compounds could be also used. In that sense, the ABL-TopoIndex should not be restricted to the parameters explaining the aerosol measurements, so that a restriction of the ABL-TopoIndex parameters to the those presenting a high correlation with the aerosol measurements is not desirable.

  The type of the aerosol distribution was added to the manuscript on p. 14:
  "*In order to have a robust estimate of the correlation between the aerosol measurements (following a Johnson distribution) and the topographical parameters (following a normal or a log-normal distribution depending on the parameters) the Spearman's rank correlation was calculated.*"

5) **Further minor comments**:

**P2, L2**: "drainage basin for air convection". Once more there is no such thing and the analogy is not valid. Since I suggest dropping parameter DBinv this term and concept can also be removed from the manuscript altogether. See the discussion above.
- As developed under §4) of this document, the authors do agree that the expression "drainage basin for air convection" is not adequate and have changed it to "drainage basin for thermally lifted pollutants". The explanation of this parameter was also modified in the manuscript (p. 7-8). While remaining open for a further discussion, the authors do consider DBinv as a potentially important parameter for further correlation studies.

**P6, L5f**: Not clear what the accuracy refers to: the altitude differences between sites and GTOPO or the specification of GTOPO? However, what you should be worried about is not the accuracy of GTOPO which refers to the average altitude in each grid cell but the representativeness of GTOPO for the whole grid cell and the resulting mismatch with your station altitudes. So the results in Table S1 are the more important numbers.
- The accuracy refers to the specification of GTOPO. The referee is right in the sense that the important point is really the representativeness of GTOPO. As already commented in the first review round, further studies should use topographical models with lower grid size in order to better represent the reality.

I am also confused about the new text inserted in the revised manuscript. It is not the same as given in the replies (see page 14 of replies), which discusses the altitude differences in much more detail. Just there it is said that the altitude difference for NCOS is 1100 m, which is not true (see Table S1). However, some of the other large altitude differences should be mentioned in the manuscript not just in the reply.
- Table S1: Thank you for pointing this out. It is the station LAN and not NCOS (also called NAM) that has the greatest negative altitude difference with the GTOPO altitude. The text was consequently modified:

> "*Due to its location at 1.2 km horizontal distance from the Yala peak (5500 m), LAN's altitude (3920 m) is 1110 m lower than its DEM grid point altitude (by 28%) and this corresponds to the largest overestimation of the station altitude.*"

- The text given on page 14 of the answer was inserted into the supplement. The authors think that it is too much detailed to be part of the manuscript, which was however complemented (in this version of the manuscript) with the following information (p. 6):

> "*Due to the altitude averaging over each grid cell, there is typically an altitude difference between the true station altitude and its corresponding grid location. The mean and median of the differences between the station altitude and its representative grid point are 190 m (8.6%) and 140 m (5.8%). For stations situated near the summits, the difference can be significant: five stations have an underestimation of their altitude greater than 500 m corresponding to 15-32 % (see the supplement and particularly Table S1 for further details), despite , according to the manual, the high reported GTOPO30 accuracy (minimal accuracy of 250 m at 90% confidence level with a RMSE of 152 m).*"

**P7, L5**: Why are these two sites discussed when there are sites like WLG and MUK also showing larger hypso%? There are many more sites with large values as seen in Table S1. So the statement that most sites have hypso% smaller 5 % is not true! Actually 22 of 46 sites have values larger 5 %. Please be more precise in all your discussion! Don't let the reviewers do your work!

- The stations mentioned at this place in the text are the ones taken as examples on the plot and were not intended at all as a description of all station hypso% values. The limit of 5% was taken from the analysis on the larger domain size (750km x 750 km) and was unfortunately not changed in the revised manuscript. The manuscript was consequently modified (p. 7):

> "*Figure 3a presents several normalized hypsometric curves with dots indicating the station hypsometric value. While most of the high altitude stations (65%) have hypso% values less than 10%, some stations are situated on wide inflection points of the hypsometric curve (see for example PYR and NCOS on Figure 3a). Six stations (see Table S1) have hypso% larger than 50%.*"

**P7, L14**: 6 out of 46 sites are situated under hyps50. What is the consequence? Should they even be discussed? The new sentences just before suggest that these sites cannot claim to be high altitude stations!

- As explained in the paper, a very small value of 10 was attributed to the 6 stations with negative hypsoD50. The main consequence is that these 6 stations have the same hypsoD50 even if the negative values are fall between -120 m and -872 m. In order to take into account the difference in hypsoD50 negative values, we introduce a new parametrization in the new version of the manuscript: instead of replacing the negative hypsoD50 by a constant value of 10, the value of 1000/abs(hypsoD50) is taken. The only consequence is to increase, to some extent, the ABL-TopoIndex of these stations that are already the stations with the highest ABL-TopoIndex values. This new parametrization induces no change in the correlation analysis because a method based on rank is used to determine that.
- If we consider the real altitude of the station instead of the GTOPO altitude, there is only 3 stations having a negative hypsoD50 (HLE: -392, LQO: -68 m and TLL:-97 m). Due to our decision to average over 9 grid cells for this parameter, it is not possible to use the real station altitude since that is not clearly defined for the adjacent grid cells.

**P7, L16f**: This is a hypothesis so far not a fact. Please rephrase.

- The sentence was changed to "*The ABL influence should decrease with increasing values of hypsoD50*".

**P7, L23**: A "plateau" is a high plain (https://en.wikipedia.org/wiki/Plateau). So here the word valley or plain should be used instead.
- The word "plateau" was changed with "high plain" in the whole text, but for "the Tibetan plateau" that is a tradition expression used in geography.

**P9, L30f**: This argumentation is confusing and misleading. First of all it is not the hydrological drainage basin that is used in the ABL index but the inverse drainage basin (5d rather than 5c). Then the size of the basin is made responsible for the high ABL index. However, just above we have seen that a large inverse drainage basin does not lead to a high ABL index (inverse drainage basin for BEO is actually larger than the one for PYR). So I would say that one of the other parameters is the more important one in terms of this discussion, most likely hypso% which differs by 3 orders of magnitude between the two sites. However, the discussion of DBinv will need to be removed from the manuscript anyway.
- The authors wanted to say that PYR has a high ABL-TopoIndex not only due to the large DB but also due to its position at the foot of Mount Everest. The authors really referred to the hydrological drainage basin of Fig. 5c, since high altitude stations are frequently at the edges of several drainage basin due to their dominant position. The sentences were misleading and was changed:
  "*PYR (5079 m) is the second highest station considered here, but the station is located at the foot of Mount Everest (8848 m) at a confluence point of several valleys (Fig. 5a and b). Figure 5c shows that PYR is situated in the middle of a very large hydrological drainage basin enhancing the fact that PYR station isn't placed at a dominant position. The PYR ABL-TopoIndex is consequently quite high (3.43) and supports the observation of a large ABL influence in the Himalaya region (Bonasoni et al., 2008).*"
- In that §, no mention of DBinv are made.

**P10, L6**: This should be 50 x 50 to 1000 x 1000 km2.
- If the mentioned page and line are correct, the size given in the manuscript corresponds to the sizes on which G8 and LocSlope are computedand not to the various sizes on which the ABL-TopoIndex was tested.

**P13, L6**: MWO supposed to have a "low DBinv", but a value of 127457 for DBinv at MWO means that the site has a DBinv larger than the mean, larger than the median and even larger than the 75% per-centile of all sites (see Table S1).
- The referee is right, this is a mistake. MWO has one the lowest hypso% and the greatest LocSlope and hypsoD50 of North America as well as the second greatest G8 of North America. The manuscript was modified:
  "*MWO is consequently the North American station with the lowest ABL-TopoIndex due to very low hypso% and relatively high G8, LocSlope and hypsoD50.*"

**P14, L12ff**: Once more, I can only say that it is surprising that the ABL-Index does not work for such sites that are clearly not high altitude sites. How can we then trust the method to rank sites at different degrees of high altitude? One common factor at all three sites is the absence of local to region emis-sions! Hence, aerosols no longer are a good estimator of FT vs ABL conditions.
- The reason why the SUM and NCOS stations are discarded is that the topography at these places is not a "complex topography". Both stations are really situated on high plains and the ABL-TopoIndex is usable only in a "mountain" context and not in a "plain" context. This is clearly explained in the manuscript (§ 3.5).

- ZEP was not included at all at the beginning of the study. ZEP is also usually not presented at all as a "high altitude" station. In Gawsis (https://gawsis.meteoswiss.ch/GAWSIS//index.html#/search/station/stationReportDetails/459), the ZEP station is described as "land". It was just my own wish to test the limits of the ABL-TopoIndex by introducing some middle altitude stations for comparison purpose. I tried to clearly describe the peculiar status of these sites in the manuscript (§ 2.1 and 3.5).

**P15, L9ff**: One should mention that these two parameters are actually strongly correlated (r=0.76, spearman)! The highest correlation between all parameter pairs! So it is no surprise that they both show similar characteristics and actually one could be sufficient for this study.

- As described on p.2 of this document, G8 and LocSlope both correspond to the steepness of the slope but at different ranges (about 1 and 10 km, respectively). The authors think that keeping both parameters could be important for various phenomena not yet studied, such as air lifting improving the new particle formation (G8) or valley winds (LocSlope).

[revised manuscript text omitted]

**Jungfraujoch, Switzerland** | 46.5477 | 7.985 | 3580 | Alps | Europe | (Bukowiecki et al., 2016) |
| **SBO**
**Sonnblick, Austria** | 47.0539 | 12.951 | 3106 | Alps | Europe | (Schauer et al., 2016) |
| **ZSF**
**Schneefernhaus, Germany** | 47.4165 | 10.9796 | 2671 | Alps | Europe | (Birmili et al., 2009) |
| **ZUG**
**Zugspitze, Germany** | 47.4211 | 10.9859 | 2962 | Alps | Europe | -- |
| **MSA**
**Montsec, Spain** | 42.05 | 0.7333 | 1570 | Pyrenees | Europe | (Ealo et al., 2016; Pandolfi et al., 2014; Ripoll et al., 2014) |
| **MSY**
**Montseny, Spain** | 41.7795 | 2.3579 | 700 | Pyrenees | Europe | (Pandolfi et al., 2011) |
| **PDM**
**Pic du Midi, France** | 42.9372 | 0.1411 | 2877 | Pyrenees | Europe | (Gheusi et al., 2011, Hulin et al., 2017) |
| **BEO**
**Moussala, Bulgaria** | 42.1792 | 23.5856 | 2925 | Balkan | Europe | (Angelov et al., 2016) |
| **CMN**
**Monte Cimone, Italy** | 44.1667 | 10.6833 | 2165 | Apennines | Europe | (Cristofanelli et al., 2016; Marinoni et al., 2008) |
| HAC
Mount Helmos, Greece | 37.9843 | 22.1963 | 2314 | Peloponnese | Europe | |
| **PUY**
**Puy de Dôme, France** | 45.7723 | 2.9658 | 1465 | Central massif | Europe | (Venzac et al., 2009) |
| **CHC**
**Chacaltaya, Bolivia** | -16.200 | -68.100 | 5320 | Andes | South America | (Andrade et al., 2015) |
| LQO
La Quiaca Observatorio, Argentina | -22.100 | -65.599 | 3459 | Andes | South America | |
| **PEV**
**Pico Espeje, Venezuela** | 8.5167 | -71.05 | 4765 | Andes | South America | (Hamburger et al., 2013; Schmeissner et al., 2011) |
| **TLL**
**Cerro Tololo, Chile** | -30.1725 | -70.7992 | 2220 | Andes | South America | Velasquez, 2016 |
| MZW
Mount Zirkel Wildness, USA | 40.5433 | -106.6844 | 3243 | Rocky Mountains | North America | |
| **NWR**
**Niwot Ridge, USA** | 40.04 | -105.54 | 3035 | Rocky Mountains | North America | |
| **SPL**
**Steamboat, USA** | 40.455 | -106.744 | 3220 | Rocky Mountains | North America | (Hallar et al., 2015) |
| YEL
Yellowstone NP, USA | 44.5654 | -110.4003 | 2430 | Rocky Mountains | North America | YEL
Yellowstone NP, USA |

| Station | Latitude | Longitude | Elevation | Sub-region | Region | Reference |
|---|---|---|---|---|---|---|
| APP
Appalachian State University, USA | 36.2130 | -81.6920 | 1076 | Appalachian | | |
| SHN
Shenandoah National Park, USA | 38.5226 | -78.4358 | 1074 | | | |
| MBO
Mount Bachelor, USA | 43.979 | -121.687 | 2743 | | | |
| MWO
Mount Washington | 44.2703 | -71.3033 | 1916 | | | |
| **WHI**
**Whistler, Canada** | 50.0593 | -122.9576 | 2182 | | | (Gallagher et al., 2011) |
| HLE
Henle, India | 32.7794 | 78.9642 | 4517 | | | |
| LAN
Langtang, Nepal | 28.2200 | 85.6200 | 3920 | Himalaya | | |
| **MUK**
**Mukteshwar, India** | 29.4371 | 79.6194 | 2180 | | | (Hyvärinen et al., 2009; Panwar et al., 2013) |
| **NCOS**
**Nam Co, China** | 30.7728 | 90.9621 | 4730 | | Asia | (Zhang et al., 2017) |
| **PYR**
**ABC Pyramid, Nepal** | 27.9578 | 86.8149 | 5079 | | | (Bonasoni et al., 2010; Marcq et al., 2010; Marinoni et al., 2010) |
| SZZ
Shangrimla ZhuZhang, China | 27.9998 | 99.4266 | 3583 | Tibetan Plateau | | |
| **WLG**
**Mount Waligan, China** | 36.2875 | 100.8963 | 3810 | | | (Andrews et al., 2011) |
| **PDI**
**Pha Din, Vietnam** | 21.5728 | 103.5160 | 1466 | -- | | -- |
| FWS
Mount Fuji, Japan | 35.3606 | 138.7273 | 3776 | | | |
| HPO
Mount Happo, Japan | 36.6972 | 137.7989 | 1850 | Japan Alps | | |
| MTA
Mount Takayama, Japan | 36.1461 | 137.4230 | 1420 | | | |
| **IZO**
**Izaña, Spain** | 28.309 | -16.4994 | 2373 | Atlantic | | (Rodríguez et al., 2012) |
| **LLN**
**Mount Lulin, Taiwan** | 23.4686 | 120.8736 | 2862 | Pacific | | (Hsiao et al., 2017) |
| **MLO**
**Mauna Loa, USA** | 19.5362 | -155.576 | 3397 | Pacific | | (Bodhaine, 1995) |
| **OMP (previously PICO-NARE)**
**Pico Mountain, Azores, Portugal** | 38.4704 | -28.4039 | 2225 | Atlantic | Islands | (Fialho et al., 2004) |
| RUN
Ile de la Réunion, France | -21.0795 | 55.3831 | 2160 | Indian | | |
| TDE
Izaña, Spain | 28.2702 | -16.6385 | 3538 | Atlantic | | |
| ASK
Assekrem, Algeria | 23.2667 | 5.6333 | 2710 | | Africa | |
| MKN
Mount Kenia | -0.0622 | 37.2972 | 3678 | | | |
| **SUM**
**Summit, Arctic** | 72.58 | -38.48 | 3238 | | Arctic | (Backman et al., 2016) |
| **ZEP**
**Zeppelin Observatory,  Norway** | 78.9067 | 11.8893 | 475 | | | (Tunved et al., 2013) |

Table 2: Extrema, median and mean of the topographical parameters for the 46 stations studied.

| Parameter | min | median | mean | max |
|---|---|---|---|---|
| ABL-TopoIndex | 0.22 | 1.72 | 4.11 | 30.12 |
| Hypso% [%] | 0.005 | 4.8 | 16.4 | 79.1 |
| HypsoD50 [m] | -872 | 1192 | 1160 | 4019 |
| LocSlope ($*10^{-3}$) | 1.7 | 86 | 93 | 259 |
| G8 [tangent] | 0.0024 | 0.1743 | 0.2053 | 0.4982 |
| DBinv [$km^2$] | 423 | 86426 | 93287 | 249464 |
| Altitude [m] | 475 | 2771 | 2802 | 5320 |
| \|Latitude\| [°] | 0.06 | 37.3 | 36.2 | 78.9 |

**Figures:**

[Figure]

Figure 1: Map of the stations colored by their mountain ranges or region.

[Figure]

Figure 2: Schematic view of the topographical features underlying the ABL-TopoIndex.

[Figure]

Figure 3: a) Normalized hypsocurves for some selected high altitude stations for a 500 km x 500 km domain centered on the station. The filled and open circles correspond to the normalized station elevations within the domain and indicate the value of hypso% (e.g., PYR hypso% is 26). The vertical dashed line corresponds to 50% of the hypsometric curve b) Difference between the station altitude and the elevation minimum in a domain of radius R around the station as a function of R. The vertical dashed line indicates the part of the curve selected to calculate LocSlope.

[Figure]

Figure 4: a) Topography on a 750x750 km² domain around BEO (Moussala, white dot) in Bulgaria. The main hydrologic flow paths from the station grid cell are given by the cyan lines. The color scale on the left only applies to Fig. 4a. b) hydrographical network, c) hydrologic drainage basins calculated from the real topography, the different drainage basins are defined by various colors and d) "convective drainage basin" calculated from the inverse topography (DBinv).

[Figure]

Figure 5: Idem Fig. 4 for PYR (Nepal Climate Observatory - Pyramid) station in the Himalaya, Nepal.

[Figure]

Figure 6: a) ABL-TopoIndex, b) drainage basin for convection, c) hypsometric percentage of the station elevation, d) hypsometric percentage of the station elevation minus the 50% hypsometry, e) local slope in a circle of 10 km radius centered on the station, f) gradient in elevation as a function of the domain size for some European high altitude stations.

[Figure]

Figure 7: ABL-TopoIndex as a function of elevation of all grid cells of a 625 km$^2$ domain centered on the ASK, CHC, HAC, OMP, PYR and SBO stations. The squares indicate the ABL-TopoIndex values and the altitudes of the stations.

[Figure]

Figure 8: ABL-TopoIndex for all stations as a function of continents and mountainous ranges. The color scheme corresponds to that in Fig. 1.

[Figure]

Figure 9: Spearman's rank correlation coefficient characterizing the correlation between aerosol parameters (absorption coefficient, scattering coefficient, number concentration) and various topographic parameters (the ABL-TopoIndex, mean altitude over the 9 grid cells, station latitude and the 5 parameters constituting the ABL-TopoIndex (G8, DB, LocSlope, hypso% and hypsoD50)). Correlations were calculated for the 5th, 50th and 95th percentiles of the aerosol parameters. Statistically significant correlation values at 95% and 90% confidence levels are marked by large and medium symbol sizes and the positive and negative correlations are plotted with upward and

downward triangles, respectively. The correlations were performed with 21, 23 and 17 stations for the absorption coefficient, the scattering coefficient and the number concentration, respectively.

[Figure]

Figure 10: Spearman's rank correlation coefficient characterizing the correlation between all the topographic parameters (see Fig. 9) and the minimum and the maximum of the monthly diurnal cycles, as well as the seasonal cycle of the aerosol parameters. The correlations are performed with 21, 22 and 15 stations for the absorption coefficient, the scattering coefficient and the number concentration, respectively.

**17[th] of July 2018: Response to the co-editor comment's:**

The authors thank the co-editor for his comments; these are addressed in what follows:

- We are now at the stage that the one of the reviewers is satisfied with your responses and the revisions that have been made in the manuscripts in some iterations. However, the second reviewer still had some significant criticism and comments regarding your last revision and the response to the provided reviews. I have been going through all the files including the last review and your last response and the revised document. Based on this I invited the reviewer to indicate if this last round of revisions and your response has properly addressed the concerns/issues of the earlier version of the manuscript. In addition, reading over again the ms I still came across quite some statements, sentences that I had to read over again also not always being convinced that these were correct regarding grammar but also not always optimally expressing that what you would like to express. Below, you can find these points that came across and that also need to be further addressed in another revision. I recommend you to ask one of the native English speaking co-authors to carefully check once more again the whole documents for the text to remove these flaws.

*Elisabeth Andrews is the second author of this paper. She is a native English speaker with more than 60 publications in peer-reviewed journals. She has reviewed each version of the manuscript as well as all the responses to the referees' comments. She has also helped in addressing the  language usage comments of the co-editor noted below. As usual, she will check the final version of the manuscript.*

Major comments:

- Title; you have changed the title also considering the changes introduced in the ms based on the reviewers comments. However, this changed title is also according to me still not optimally covering the actual contents of the paper; especially the use of the term aerosol layer is not reflecting the contents. The suggested change in title by one of the reviewers seems to cover much better the actual content. Your study focuses on identifying the influence of topography on aerosol measurements at high-altitude stations.

  *The authors agree to change the title to: "Identification of topographic features influencing aerosol observations at high altitude stations".*

- In addition, it seems that have not tackled the issue of the reviewer on the use of global modelling products to also assess the role of topography in aerosol properties at higher elevations. I am myself not so convinced that the resolution is already sufficiently high to indeed use model products for this, at least not from global models. There are though meso-scale models that can resolve some of the fine-scale meteorological features at resolutions down to some km scale (e.g., WRF). My point is that you should at least address this remark by the reviewer even if at the end you decide not to include this aspect in your paper.
  *The authors do agree that the modelling approach could provide additional information and would be a useful direction for future studies. While the second referee clearly would like to see a modelling component added to this current investigation, even s/he acknowledges in his/her second review that this may be beyond the scope of this paper saying:*
  *"I still think it would be possible to create a meteorological criterion that would indicate situations with likely thermally induced flow from existing global scale model products. Don't forget that the latter have resolution down to 0.1 degree by now. Also it has been done successfully before with observational data, so why not check with model data. **However, I see that this may go well beyond the scope of the current analysis.**" Below we've expanded our response to this comment.*

  *Most global models are typically not of sufficiently high resolution to simulate complex terrain (e.g., Benedetti et al., 2018).  A preliminary study at MeteoSwiss (trainee work, personal communication) with the COSMO model shows that a higher degree of detail in the topography and surface fields leads to a better estimation of the convergence/divergence of the mesoscale flow over the Alps during the day/night (Alpine pumping). However, this was for limited locations in the Swiss Alps  and for limited meteorological conditions (i.e., 14 clear-sky summer days). In contrast, the analysis presented in the manuscript aims to define an index that can be applied to all high altitude stations around the world. A study by Wang et al (2016) shows that, in addition to model resolution, the spatial resolution and distribution of emissions sources is also an important factor in how well models are able to represent observations of aerosol. We expect that complex topography would make high spatial resolution of emissions even more critical.  We feel that it is clearly beyond the scope of this paper to utilize output from one or more high resolution models (regional or global) to*

*evaluate the aerosol observations presented here. Doing so would involve evaluating the model assumptions and parameterizations of aerosol and aerosol precursor emission distributions, aerosol transformation and removal processes as well as the details about how the meteorology is modelled for each high altitude station. The effects of mountain topography is complex to model and a topic of ongoing research (see for example, Serafin et al., 2018, Arnold et al., 2012) while models also struggle with simulating observations of aerosol particles - even in less complex terrain (see for example, Mann et al., 2014; Tsigaridis et al., 2014; Eskes et al 2018.). In the future we hope to partner with members of the modelling community to further explore our initial findings on how topography influences aerosol particle observations at high elevation sites.*

*Arnold, D., et al.:" Issues in high-resolution atmospheric modeling in complex topography – the HiRCot Workshop ", Croatian Meteorological Journal, 47, 3–11, 2012.*

*Eskes, H.J. et al.:" Validation report of the CAMS near-real time global atmospheric composition service, December 2017 - February 2018", Copernicus atmosphere monitoring service (CAMS) report, CAMS84_2015SC3_D84.1.1.11_2018DJF_v1.pdf, 2018. Available at [HTTP://ATMOSPHERE.COPERNICUS.EU/QUARTERLY_VALIDATION_REPORTS](HTTP://ATMOSPHERE.COPERNICUS.EU/QUARTERLY_VALIDATION_REPORTS)*

*Benedetti, A. et al., "Status and future of Numerical Atmospheric Aerosol Prediction with a focus on data requirements," Atmos. Chem. Phys. Discuss., [https://doi.org/10.5194/acp-2018-42](https://doi.org/10.5194/acp-2018-42). 2018.*

*Mann, G. et al., "Intercomparison and evaluation of global aerosol microphysical properties among AeroCom models of a range of complexity," Atmos. Chem. Phys., 14, 4679–4713, 2014.*

*Serafin, S., et al. :"Exchange Processes in the Atmospheric Boundary layer Over Mountainous Terrain", Atmosphere, 9, 1–32. [https://doi.org/10.3390/atmos9030102](https://doi.org/10.3390/atmos9030102), 2018.*

*Tsigaridis, K. et al. "The AeroCom evaluation and intercomparison of organic aerosol in global models," Atmos. Chem. Phys., 14, 10845–10895, 2014.*

*Wang, R. et al., "Estimation of global black carbon direct radiative forcing and its uncertainty constrained by observations." J. Geophys. Res.: Atmospheres, 121, 5948–5971, 2016.*

Minor comments:
- Page 6, line "non-GIS environment(Schwanghart and Scherler, 2014)" put space there.
  *Done*

- Page 6; line 20: "Based on these criteria, the red station on Fig. 2 will be less influenced by the ABL…", alternative; "Based on these criteria it can be inferred from Figure 2 that the "red" station will be less influenced by the ABL….
  *The sentence was corrected as proposed by the co-editor.*

- Page 7, line 2: "have hypso% *values* larger than 50%."
  *The sentence was corrected as proposed by the co-editor*

- Page 7, you refer in line 16 for example to small spatial scale and in the definition of the previous criteria to large spatial scale; here it is essential to indicate (again) what you deem being a small and a large spatial scale.

*The size of the spatial scale is now mentioned explicitly for each parameter constituting the ABL-TopoIndex.*

- Page 7; line 23, what do you mean with "and there are some steps for CHC and BEO", what steps ? and steps of what size?
*The nature and significance of the "steps" is now detailed in the text:" For example, there is a rapid decrease of the altitude difference with increasing distance that gradually levels off for radius larger than 7 km for JFJ and for radius larger than 4 km for MBO; there is a continuous decrease of the altitude difference for PYR and ASK up to radius larger than 30 km; and there are sites for which the altitude difference stays constant for a portion of the domain radius (see for example CHC and BEO) indicating the presence of flat terrain.."*

- Page 9: line 26: "PYR (5079 m) is the second highest station considered here, but..", reading this section and statement this line was confusing since the term "here" suggests that this is referring to the second highest station of this case study. You mean here that PYR is the second highest stations of all stations considered in the study and would then also state it this way for clarity.
*The sentence was modified as proposed by the co-editor:" PYR (5079 m) is the second highest station of all stations considered in this study, but the station is located at the foot of Mount Everest (8848 m) at a confluence point of several valleys (Fig. 5a and b)."*

- Page 10, line 2: "The ABL-TopoIndex depends on the size of the chosen domain (Fig. 6a) so that the various algorithms were tested to several domain sizes ranging from 50 to 1000 km2. The gradient G8 and the local slope LocSlope are calculated on small fixed horizontal scales (0.5-1 and 10 km, respectively)". I had to read these two sentences a couple of times also not being convinced that this is the most optimal way to express what you intent to say. There are actually more of these sentences in the ms and would anyhow propose to still have once a native speaking English co-author (I guess there is one given the large list of co-authors) to critically check the ms for such potential flaws. Alternative: "Since the ABL-TopoIndex depends on the size of the chosen domain (Fig. 6a) we have conducted an evaluation of the sensitivity of the various algorithms to the domain size using a range from 50 to 1000 km2. The gradient criterion G8 and the local slope criterion LocSlope are calculated on small fixed horizontal scales (0.5-1 and 10 km, respectively)"
*The sentence was replaced by the one proposed by the co-editor. The whole manuscript has been checked by a native English speaker and the last version of the manuscript will be once again carefully gone over.*

- Page 10: line 9: "the concentration of thermally lifted pollutants"
*The sentence was corrected as proposed by the co-editor.*

- Page 10, line 9/10: "The hypso% decreases continuously for stations situated in a dominant position in their mountainous massif such as JFJ, SBO or BEO (Fig. 6c)". This is another example of a sentence that should be read carefully and revised. "continuously" here expresses something like over time whereas you want to express here that this parameter increases with an increase in the domain size.
*The sentence was modified: "The hypso% decreases continuously with domain size for stations situated in a dominant position in their mountainous massif such as JFJ, SBO or BEO (Fig. 6c)"*

- Line 15: "with domain size", change to "with an increase in domain size"
*The sentence was corrected as proposed by the co-editor. The sentence on Line 16 was similarly modified.*

- Page 10; line 23-24 "To compare these two parameters, we show in Fig. 7 the ABL-TopoIndex as a function of the altitude for all grid cells in a 5km x 5km domain around a selection of stations"
  *The sentence was modified: To compare these two parameters, we show in Fig. 7 the ABL-TopoIndex as a function of the altitude for all grid cells in a 5km x 5km domain \*for a subset\* of stations.*

- Page 10; line 26: "..very steep and ASK a very flatt ABL-TopoIndex decrease with altitude"; first of all "flat", then secondly, what is a flat decrease?? I guess you would like to say that there is a strong increase and a small decrease or?
  *The sentence was modified as proposed by the co-editor:" Fig. 7 shows that the OMP and PYR regions have a very large ABL-TopoIndex decrease with altitude while ASK exhibits a very small ABL-TopoIndex decrease with altitude."*

- Line 33: "were constructed", alternative "are located"
  *The sentence was corrected as proposed by the co-editor.*

- Page 11: line 6: "are grouped on Fig. 8": are grouped as shown in Fig. 8 (change all this consistently in the text, e.g. "on Fig. 9"
  *This was corrected in the whole text.*

- Page 12, line 10: "their proximity *to* other massifs such as the Alps"
  *The sentence was corrected as proposed by the co-editor.*

- Page 14, line 14-15: "…ABL influence, in case of lifting processes without precipitation, is found for the ABL-TopoIndex…."
  *The sentence was changed: "…ABL influence in the case of lifting processes without precipitation…"*

- Page 14; line 18-19: "(mean the altitude over the 9 grid cells, similarly to the ABL-TopoIndex calculation)"
  *The sentence was rearranged and changed to "The Spearman's rank correlation coefficients of the 5th, 50th and 95th percentiles of the measured aerosol parameters with site altitude latitude, ABL-TopoIndex as well as all the individual parameters constituting the ABL-TopoIndex are presented in Fig. 9. (Similar to the ABL-TopoIndex calculation, the mean of the altitudes of the grid cell containing the station and its eight neighboring cells was used.)" "*

- Page 15, line 12, "lowest and the greatest monthly amplitudes", should be according to me "smallest and largest monthly amplitudes", also check further the ms for this: e.g. "the greatest ABL influence" should be "the largest ABL influence" (possibly a matter of taste). You use many times the term greater it should be larger/largest or higher/highest, e.g. "higher correlations"
  *The whole paper was checked and the antonyms low/high, big/small, least/greatest are now used. Moreover the "greatest ABL influence" was always replaced by "the largest ABL influence" and "greatest correlation" with "highest correlation".*

- Page 15, line 21: "diurnal cycle minimal and maximal strengths of the absorption coefficient". This expression reads also not well. You figure caption text seems to express it in the proper way, modify this text.
  *The sentence was modified:" The ABL-TopoIndex is s.s. correlated with the minimum and the maximum of the monthly diurnal cycles of the absorption coefficient."*

- Page 16, line 6: "First the possible species and phenomena enabling the estimation of the ABL influence", what do you mean here with species?? Do you refer here to compounds. I also think you generally want to refer here more generally to "parameters"
  *The word "species" was changed to "parameters.*

- Page 16, line 27, "NOy" y lowercase
  *Done*

- Page 16, line 28: "should be in most cases"; "could be in most cases"
  *The sentence was corrected as proposed by the co-editor.*

- Page 16, line 29 "hence involving aerosol washout"
  *The sentence was modified:" However, one has to remain conscious that many lifting processes co-occur with precipitation and, hence, potential aerosol washout.".*

- Page 17: line 4-5: Your statement about using a model to further assess how pollutants can be used as a proxy for BL influence reads weird: what do you mean with a thermodynamic model? "…bounded to a 3D thermodynamic model adapted to complex topographies would be required before using absolute pollutant concentrations as indicators of ABL influence at high altitude sites", rather "..constraining simulations with meteorological models able to explicitly resolve the role of fine resolution orography would be …." (see also my previous comment on the title/introduction and first major comment of the reviewer).
  *The sentence proposed by the co-editor was inserted in the ms.*

- In addition, this text is part of your modification of the ms responding another major comment provided by one of the reviewers. There is another part of this modification that raises quite some questions: " A further use of DBinv to restrict the area of potential pollution sources could also be envisaged since this parameter describes the domain from which pollutants can reach the high altitude station by convection and without crossing topographical barriers. This delicate issue can however be avoided by instead considering dynamical parameters such as the various temporal cycles". What is delicate here? What are the dynamical parameters? You mention here the temporal cycles (diurnal and seasonal?) but in what parameters. I wonder what the reviewer will express about this modification. I am myself not very convinced and consequently suggest you to check this once more again carefully
  *The expression "delicate issue" was replaced by the following sentence: "The identification of pollution source areas potentially affecting the high altitude stations can be avoided by instead considering dynamic parameters such as the temporal cycles of various pollutants".*

- Page 17: line 14-15: " Usually the spring leads to higher aerosol loading than the autumn probably related to higher ABL height in the spring"; do you refer here to a higher aerosol loading/concentrations at the high-altitude stations?? I don't see how a deeper ABL would result in a higher aerosol loading in the ABL, actually the opposite would be expected not having any changes in the sources (more dilution)
  *The ambiguity was removed by changing the sentence:" Usually, at high altitude stations, the spring leads to higher aerosol loading than the autumn; this is probably related to higher ABL height in the spring"*

- Page 18: line 7: "has similar dependency as the ABL as a function of latitude", dependency on what? I know that in the following sentences you give examples but rephrase this sentence.

*The sentence was modified:" Further, the RL has a similar dependency as the ABL on latitude, i.e., the RL's maximum height also depends on the duration of the incoming radiation."*

- Page 18, line 16-17: "modify the theoretical cycles and lead to a broadening of the time of the extrema. These difficulties make obtaining clear statistical cycles another reason contributing to the observed low correlations", this text is another example of statements that need definitely to be rephrased. What are theoretical cycles: cycles that you would anticipate based on basic theory? What is meant with a broadening of the time? It would be broadening the time frame or increasing the duration, and which extrema?? And the second sentence needs complete revision.
  *Both sentences were modified:" However, the statistical determination of the diurnal and seasonal cycle amplitudes suffer from several difficulties: 1) the low aerosol concentrations observed at high altitude often result in measurements near the detection limit leading to large uncertainties, 2) the high hourly autocorrelation of the data requires a pre-whitening procedure (see supplement) in order to be able to detect the diurnal and seasonal cycle, 3) meteorological conditions (e.g., cloud coverage, precipitation, seasonal fluctuations, etc.) modify the clear-sky diurnal cycles. These factors constraining the observation of clear statistical temporal cycles in the measurement data also contribute to the observed low correlations between the diurnal and seasonal cycles of the aerosol parameters and the ABL-TopoIndex."*

- Page 18, line 26- 27: Correlation between topography and aerosol parameters and "in Sect. 3.5"
  *Done*

- Page 19, line 22-23: "Globally, NPF is the reason why the greatest correlations are found with the 50 percentile of the number concentration, instead of with the 5 percentile found for the absorption and scattering coefficients", also this sentence needs to be rewritten: correlations with?
  *The sentence was deleted.*

- Page 20, line 7,when you mention these terms "the Efremov-Krcho classification, the hypsometric curve" you should shortly explain them but also indicate why you considered to include these terms in the analysis. On the other hand, the discussion is already now (way too) long.
  *The authors agree that description in the manuscript is very short. On the request of one referee, a complete description of these parameters were added in the supplement (see Table 2). In order to keep the discussion as short as possible, only the basic domains from which these eliminated parameters were taken are now added:" Several other parameters taken from the topography, morphology or hydrology fields such as the topographical wetness index, the upstream catchment area, the Efremov-Krcho landform classification, the integral and index of the hypsometric curve, and the topographic prominence were tested but eliminated as being not relevant for various reasons (Table S2)."*

- Page 20, line 28: "It is …"
  *Done*

- References to "de Wekker" should be listed in the references under the "D" and not the "W" references.
  *Done*

- Figure 1: the station names with the different colors come out sometimes quite poorly, like the stations in the US. Use only or white or black characters?
  *The authors propose to use the same color code for figures 1 and 8 and to add a grey scale topography as a background.*

[Figure]

**25th of July 2018: Response to the second reviewer comment's:**

*The authors thank the second reviewer for the fruitful discussions through all the three rounds of comments and responses. The manuscript ends up with a much more collective point of view taking into account not only the aerosol research domain but also that of atmospheric dynamics. The co-editor and the reviewer will find thereafter the response to the last comments. It has to be noted that the page and line numbers relate to the third version submitted to ACP that was sent to the co-editor and editorial board on the 19th of July 2018.*

*The authors also add at the end of the response to the second reviewer comment's the response to the co-editor comment's sent on the 19th of July 2018.*

After reading through the authors' replies and the careful comments by the editor I only have a few additional remarks. Most of the replies and corrections help to clarify the paper but I think with a little additional work some more meaningful results could have been obtained. But I will leave this for future work.

Some remaining concerns:

- DBinv: Since the authors are really so insisting on keeping this parameter in their analysis I suggest at least to change the name of it throughout the manuscript. Instead of 'drainage

basin for thermally lifted pollutants', which I am still not convinced it represents, the authors should just stay with 'inverse drainage basin' and describe it as potentially reflecting the probability for thermally lifted pollutants to reach the site. Seems to be the same thing, but it really makes a difference if a confusing name is given or a topographic parameter is just tested. However, then in the conclusion it should also be made clear that the parameter was not a good predictor and, hence, the suggested analogy of inverse drainage flow and flow barriers did not work so well.

*The authors do agree to change the name for DBinv. The following modifications were done:*

- *p.2 line 29 : "To construct the ABL-TopoIndex, we rely on the criteria that the ABL influence will be low if the station is one of the highest points in the mountainous massif, if there is a large altitude difference between the station and the valleys or high plains, if the slopes around the station are steep, and finally if the inverse drainage basin potentially reflecting the source area for thermally lifted pollutants to reach the site is small."*
- *P. 6 line 19: "4) the inverse drainage basin, which potentially reflects the source area for thermally lifted pollutants, is small. "*
- *P. 7 line 33:" Parameter 5 – DBinv: Since the air masses have to be thermally lifted from the valleys and plains towards the summit to influence the station measurements, the size of the inverse drainage basin (DBinv) can be calculated with standard hydrology tools using an inverse topography, where the altitude Z is changed to –Z allowing the summit to become a hole. It potentially represents the region from which pollutants such as aerosols can be thermally lifted without crossing any topographical barrier. DBinv is related to criterion 4 for a large spatial scale (500 km x 500 km). Figures 4d and 5d are examples of the DBinv calculation for BEO and PYR. The ABL influence should increase with increasing size of the inverse drainage basin."*
- *P. 9 line 23: "Figure 4d shows that when the inverse drainage basin is calculated with the inverse topography, BEO is in the center of a large inverse drainage basin that covers most of the plotted domain."*
- *P.21 ine21:" The ABL-TopoIndex is a topographical index based on the hypsometric curve, the slope of the terrain around the station and the inverse drainage basin that potentially reflects the source area for thermally lifted pollutants"*
- *P. 38 line 4: "Figure 4: a) Topography on a 750x750 km$^2$ domain around BEO (Moussala, white dot) in Bulgaria. The main hydrologic flow paths from the station grid cell are given by the cyan lines. The color scale on the left only applies to Fig. 4a. b) hydrographical network, c) hydrologic drainage basins calculated from the real topography, the different drainage basins are defined by various colors and d) "inverse drainage basin" calculated from the inverse topography (DBinv)."*
- *P. 39 line 3:" Figure 6: a) ABL-TopoIndex, b) inverse drainage basin, c) hypsometric percentage of the station elevation, d) hypsometric percentage of the station elevation minus the 50% hypsometry, e) local slope in a circle of 10 km radius centered on the station, f) gradient in elevation as a function of the domain size for some European high altitude stations."*

*To clearly indicate that DBinv is not a good predictor, the following sentence was added to the conclusion (p. 22 line 5):" The inverse drainage basin seems to be the least explanatory parameter in terms of ABL influence and this large scale parameter should either be further*

*evaluated or be combined with a source inventory to increase its relevance for identifying boundary layer influence."*

- Latitude and thermally induced transport: Sure latitude is a good predictor. The differential heating that is necessary to produce thermal lifting obviously depends, among other factors, on latitude. Why else are the authors focusing on summer for mid-latitude sites. The absence of strong differential heating is another factor why the results of ZEP are so poor. There simply is no thermal lifting.

*The authors do agree with this statement and add this information in p. 13 line 6:" ZEP, situated at very low altitude (475 m) and very high latitude (78.9°), also has a very high ABL-TopoIndex value. It was also not included in the correlation analysis since its seasonal and diurnal cycles exhibit different features than the high altitude or middle latitude stations (see Sect. 4.1)."*

- Regression model: I was thinking about generalised additive models (GAM, Wood et al., 2006) and parameter selection like in Jackson et al. (2009). Certainly there is a way to deal with non-normal distributions if necessary. There are even more advanced methods coming up in the age of machine learning (random forests, etc.). But I can see that this goes beyond the scope of the current paper but it could have significantly improved the paper and removed some of the speculative aspects.

*The authors thank the reviewer for these very interesting references. They think however that the context between the cited study and the manuscript presents some substantial differences. The parameters selection for GAM in Jackson et al. (2009) is based on (to a large extent) already known chemical reactions with a good overall agreement achieved between modelled and measured chemical species concentration. Our study is definitely a first step trying to understand the role of the topography in the ABL-influence at high altitude sites and the aerosol parameters used to validate the ABL-TopoIndex do not constitute the sole explanatory variables. Even though the authors do not have any experience with GAM, they expect at least two potential sources of problems:*

> *1) an exponential family distribution has to be specified for the univariate response variable along with a link function (see [https://en.wikipedia.org/wiki/Generalized_additive_model](https://en.wikipedia.org/wiki/Generalized_additive_model)). As already specified in the second response to the referee comments, the only distribution describing the aerosol parameters is the Johnson distribution, which is not part of the exponential family distribution.*

> *2) the number of aerosol time series available for high altitude stations remains sparse (between 15 and 23 for the analyzed parameters) and could be a clear source of large uncertainties.*

*Nonetheless,the approach is very interesting and could be included in future development of the ABL-TopoIndex.*